# PERFORMATIVE POLICY GRADIENT: ASCENT TO OPTIMALITY IN PERFORMATIVE REINFORCEMENT LEARNING

## ABSTRACT

Post-deployment machine learning algorithms often influence the environments they act in, and thus *shift* the underlying dynamics that the standard reinforcement learning (RL) methods ignore. While designing optimal algorithms in this *performative* setting has recently been studied in supervised learning, the RL counterpart remains under-explored. In this paper, we prove the performative counterparts of the performance difference lemma and the policy gradient theorem in RL, and further introduce the **Performative Policy Gradient** algorithm (PePG). PePG is the first policy gradient algorithm designed to account for performativity in RL. Under softmax parametrisation, and also with and without entropy regularisation, we prove that PePG converges to *performatively optimal policies*, i.e. policies that remain optimal under the distribution shifts induced by themselves. Thus, PePG significantly extends the prior works in Performative RL that achieves *performative stability* but not optimality. Furthermore, our empirical analysis on standard performative RL environments validate that PePG outperforms standard policy gradient algorithms and the existing performative RL algorithms aiming for stability.

## 1 INTRODUCTION

Reinforcement Learning (RL) studies the dynamic decision making problems under incomplete information (Sutton & Barto, 1998). Since an RL algorithm tries and optimises an utility function over a sequence of interactions with an unknown environment, RL has emerged as a powerful tool for algorithmic decision making. Specially, in the last decade, RL has underpinned some of the celebrated successes of AI, such as championing Go with AlphaGo (Silver et al., 2014), controlling particle accelerators (St. John et al., 2021), aligning Large Language Models (LLMs) (Bai et al., 2022), reasoning (Havrilla et al.), to name a few. But the existing paradigm of RL assumes that the underlying environment with which the algorithm interacts stays static over time and the goal of the algorithm is to find the utility-maximising, aka optimal policy for choosing actions over time for this specific environment. But *this assumption does not hold universally*.

In this digital age, algorithms are not passive. Their decisions also shape the environment they interact with, inducing distribution shifts. This phenomenon that predictive AI models often trigger actions that influences their own outcomes is termed as *performativity*. In the supervised learning setting, the study of *performative prediction* is pioneered by Perdomo et al. (2020), and then followed by an extensive literature encompassing stochastic optimisation, control, multi-agent RL, games (Izzo et al., 2021; 2022; Miller et al., 2021; Li & Wai, 2022; Narang et al., 2023; Piliouras & Yu, 2023; Góis et al., 2024; Barakat et al., 2025) etc.

There has been several attempts to achieve performative optimality or stability for real-life tasks— recommendation systems (Eilat & Rosenfeld, 2023), measuring the power of firms (Hardt et al., 2022; Mofakhami et al., 2023), healthcare (Zhang et al., 2022) etc. Performativity of algorithms is also omnipresent in practically deployed RL systems. For example, an RL algorithm deployed in a recommender system does not only aim to maximise the user satisfaction but also shifts the preferences of the users in the long-term (Chaney et al., 2018; Mansoury et al., 2020). To clarify the impact of performativity, let us consider an example.

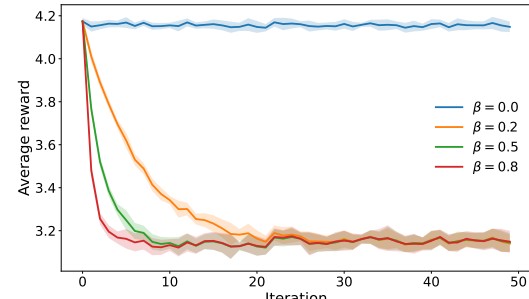

Figure 1: Average reward (over 10 runs) obtained by ERM and Performative Optimal policies across performative strength $\beta$.

**Example 1** (Performative RL in loan approval). *Let us consider a loan approval problem, where an applicant obtains a loan (or get rejected) according to their credit score $x$, and $x$ depends on the capital of the applicant and that of the population. At each time $t$, a loan applicant arrives with a credit score $x_t$ sampled from $\mathcal{N}(\mu_t, \sigma^2)$. The bank chooses whom to give a loan by applying a softmax binary classifier $\pi_\theta : \mathbb{R} \to \{0, 1\}$ on $x$ with threshold parameter $\theta$. This decision has two effects. (a) The bank receives a positive payoff $R$, if the loan*

*applicant who was granted a loan repays, or else, loses by L. Thus, the bank's expected utility for policy $\boldsymbol{\pi}_\theta$ is $U(\theta, \mu) = \mathbb{E}_{x \sim \mathcal{N}(\mu, \sigma^2)} \left[ \boldsymbol{\pi}_{\boldsymbol{\theta}}(x)(\mathbb{P}(repayment|x)R - (1 - \mathbb{P}(repayment|x))L) \right]$. (b) Since the amount of capital both the applicant and the population influence the credit score, we model that the change in the population mean $\mu_{t+1}$ depends on the bank's policy, via a grant rate $\mathbb{E}_{x \sim \mathcal{N}(\mu_t, \sigma_t^2)} \left[ \pi_\theta(x) \right]$. Specifically, $\mu_{t+1} = (1 - \beta)\mu_t + \beta f \left( \mathbb{E}_{x \sim \mathcal{N}(\mu_t, \sigma_t^2)} \left[ \pi_\theta(x) \right] \right)$, where $\beta \in [0, 1]$ is the performative strength and $f : \mathbb{R} \to [-M, M]$. Now, if one ignores the performative nature of this decision making problem, and try to find out the optimal with respect to a static credit distribution, it obtains $\theta^{ERM} \triangleq \arg\max_\theta U(\theta, \mu_0)$. In contrast, if it considers performativity, it obtains $\theta^{Perf} \triangleq \arg\max_\theta U(\theta, \mu^*(\theta))$. In Figure 1, we show that the average reward obtained by both the solutions are significantly different. This demonstrates why performativity is a common phenomenon across algorithmic decision making problems, and how it changes the resulting optimal solution. Further details are in Appendix B.*

These problem scenarios have motivated the study of performative RL. Though Bell et al. (2021) were the first to propose a setting where the transition and reward of an underlying MDP depend non-deterministically on the deployed policy, Mandal et al. (2023) formally introduced *Performative RL*, and its solution concepts, i.e., performatively stable and optimal policies. Performative stable policies do not get affected or changed due to distribution shifts after deployment. Performatively optimal policies yield the highest expected return once deployed in the performative RL environment. Mandal et al. (2023) proposed direct optimization and ascent based techniques that attains performative stability upon repeated retraining. Extending this work, Rank et al. (2024) and Mandal & Radanovic (2024) manage to solve the same problem with delayed retraining for gradually shifting and linear MDPs. However, *there exists no algorithm yet in performative RL that provably converges to the performative optimal policy.*

As we know from the RL literature, the Policy Gradient (PG) type of algorithms that treats policy as a parametric function and updates the parameters through gradient ascent algorithms are efficient and scalable (Williams, 1992; Sutton et al., 1999; Kakade, 2001). Some examples of successful and popular policy gradient methods include TRPO (Schulman et al., 2015), PPO (Schulman et al., 2017), NPG (Kakade, 2001), which are widely used in modern RL applications. Recent theoretical advances also establish finite-sample convergence guarantees and complexity analyses (Agarwal et al., 2021; Yuan et al., 2022) of PG algorithms. Motivated by the simplicity and universality of the PG algorithms, we ask these two questions in the context of performative RL:

> 1. *How to design PG-type algorithms for performative RL environments to achieve optimality?*
> 2. *What are the minimal conditions under which PG-type algorithms converge to the performatively optimal policy*?

**Our contributions** address these questions affirmatively, and showcases the difference of optimality-seeking and stability-seeking algorithms in performative RL.

**I. Algorithm Design:** We propose the first Performative Policy Gradient algorithm, PePG, for performative RL environments. Specifically, we extend the classical vanilla PG and entropy-regularised PG algorithms to Performative RL settings. Though the general algorithm design stays same, we derive a performative policy gradient theorem that shows, evaluation of the gradient involves two novel gradient terms in performative RL – (a) the expected gradient of reward, and (b) the expected gradient of log-transition probabilities times its impact on the expected cumulative return. We leverage this theorem to propose an estimator of the performative policy gradient under any differentiable parametrisation.

**II. Convergence to Performative Optimality.** We further analyse PePG (with and without entropy regularisation) for softmax policies, and softmax Performative Markov Decision Processes (PeMDPs), i.e. the MDPs with softmax transition probabilities and linear rewards with respect to the parameters of the softmax policy. We provide a minimal recipe to prove convergence of PePG using (a) smoothness of the performative value function, and (b) approximate gradient domination lemma for performative policy gradients. This allows us to show that PePG converges to an $\epsilon$-ball around performative optimal policy in $\Omega\left( \frac{|\mathcal{S}||\mathcal{A}|^2}{\epsilon^2(1-\gamma)} \right)$ iterations, where $|\mathcal{S}|$ and $|\mathcal{A}|$ are the number of states and actions, respectively.

Specifically, Mandal et al. (2023) frames the question of using policy gradient to find stable policies as an open problem. The authors further contemplate, as PG functions in the policy space, whether it is possible to converge towards a stable policy. In this paper, we affirmatively solve an extension to this open problem for tabular softmax PeMDPs with softmax policies.

**III. Stability- vs. Optimality-seeking Algorithms in Performative RL.** We further theoretically and numerically contrast the performances of stability-seeking and optimality-seeking algorithms. Theoretically, we derive the performative performance difference lemma that distinguished the effect of policy update in these two types of algorithms. Numerically, we compare the performances of PePG with the state-of-the-art MDRR (Mixed Delayed Repeated Retraining (Rank et al., 2024)) algorithm for finding performatively stable policies in the multi-agent environment proposed by (Mandal et al., 2023). We show that PePG yields significantly higher values functions than MDRR, while MDRR achieves either similar or lower distance from stable state-action distribution than PePG .

## 2 PRELIMINARIES: FROM RL TO PERFORMATIVE RL

Now, we formalise the RL and performative RL problems, and provide the basics of policy gradient algorithms in RL.

### 2.1 RL: INFINITE-HORIZON DISCOUNTED MDPS

In RL, we mostly study Markov Decision Processes (MDPs) defined via the tuple $(\mathcal{S}, \mathcal{A}, \mathbf{P}, r, \gamma)$, where $\mathcal{S} \subseteq \mathbb{R}^d$ is the state space and $\mathcal{A} \subseteq \mathbb{R}^d$ is the action space. Both the spaces are assumed to be compact. At any time step $t \in \mathbb{N}$, an agent plays an action $a_t \in \mathcal{A}$ at a state $s_t \in \mathcal{S}$. It transits the MDP environment to a state $s_{t+1}$ according to a transition kernel $\mathbf{P}(\cdot \mid s_t, a_t) \in \Delta(\mathcal{S})$. The agent further receives a reward $r(s_t, a_t) \in \mathbb{R}$ quantifying the goodness of taking action $a_t$ at $s_t$. The strategy to take an action is represented by a stochastic map, called *policy*, i.e. $\boldsymbol{\pi} : \mathcal{S} \to \Delta(\mathcal{A})$. Given an initial state distribution $\boldsymbol{\rho} \in \Delta(\mathcal{S})$, *the goal is to find the optimal policy $\boldsymbol{\pi}^\star$ that maximises* the expected discounted sum of rewards, i.e., the *value function*: $V_{\boldsymbol{\pi}}(\boldsymbol{\rho}) \triangleq \mathbb{E}_{s_0 \sim \boldsymbol{\rho}, s_{t+1} \sim \mathbb{P}(\cdot \mid s_t, \boldsymbol{\pi}(s_t))} \left[ \sum_{t=0}^{\infty} \gamma^t r(s_t, \boldsymbol{\pi}(s_t)) \right]$, where $\gamma \in (0, 1)$ is called the *discount factor*. $\gamma$ indicates how much a previous reward matters in the next step, and bounds the effective horizon of a policy to $\frac{1}{1-\gamma}$.

---

**Algorithm 1** Vanilla Policy Gradient

---
1: **Input:** Learning rate $\eta > 0$.
2: **Initialize:** Policy parameter $\boldsymbol{\theta}_0(s, a) \forall s \in \mathcal{S}, a \in \mathcal{A}$.
3: **for** $t = 1$ to T **do**
4:     Estimate the gradient $\nabla_{\boldsymbol{\theta}} V^{\boldsymbol{\pi}}(\boldsymbol{\rho}) \mid_{\boldsymbol{\theta} = \boldsymbol{\theta}_t}$
5:     **Gradient ascent step:** $\boldsymbol{\theta}_{t+1} \leftarrow \boldsymbol{\theta}_t + \eta \nabla_{\boldsymbol{\theta}} V^{\boldsymbol{\pi}}(\boldsymbol{\rho}) \mid_{\boldsymbol{\theta} = \boldsymbol{\theta}_t}$
6: **end for**

---

**Policy Gradient (PG) Algorithms.** PG-type algorithms maximise the value function by directly optimising the policy through a gradient over value function (Williams, 1992). To compute the gradient, we choose a parametric family of policies $\boldsymbol{\pi}_{\boldsymbol{\theta}}$ for some $\boldsymbol{\theta} \in \mathbb{R}^d$ (e.g. direct (Agarwal et al., 2021; Wang & Zou, 2022), softmax (Agarwal et al., 2021; Mei et al., 2020), Gaussian (Ciosek & Whiteson, 2020; Ghavamzadeh & Engel, 2006)). Specifically, vanilla PG (Algorithm 1), performs a gradient ascent on the policy parameter at each step $t \in \mathbb{N}$. As the goal is to maximise

$V^{\boldsymbol{\pi}}(\rho)$, we update $\boldsymbol{\theta}$ towards $\nabla_{\boldsymbol{\theta}} V^{\boldsymbol{\pi}}(\rho)$, which is the direction improving the value $V^{\boldsymbol{\pi}}(\rho)$ with a fixed learning rate $\eta > 0$. For vanilla PG, the policy gradient takes the convenient form leading to estimators computable only with policy rollouts.

**Theorem 1** (Policy Gradient Theorem (Sutton et al., 1999))**.** *Fix a differentiable parmeterisation $\theta \mapsto \pi_\theta(a \mid s)$ and an initial distribution $\boldsymbol{\rho}$. Let us define the Q-value function $Q^{\boldsymbol{\pi}_{\boldsymbol{\theta}}}(s, a) \triangleq \mathbb{E}_{s_{t+1} \sim \mathbf{P}_{\boldsymbol{\pi}}(\cdot \mid s_t, \boldsymbol{\pi}(s_t))} \left[ \sum_{t=0}^{\infty} \gamma^t r(s_t, \boldsymbol{\pi}(s_t)) \mid s_0 = s, a_0 = a \right]$, and advantage function $A^{\boldsymbol{\pi}_{\boldsymbol{\theta}}}(s, a) \triangleq Q^{\boldsymbol{\pi}_{\boldsymbol{\theta}}}(s, a) - V^{\boldsymbol{\pi}_{\boldsymbol{\theta}}}(s)$. Then,*

$$\nabla_{\boldsymbol{\theta}} V^{\boldsymbol{\pi}_{\boldsymbol{\theta}}}(\boldsymbol{\rho}) = \frac{1}{1 - \gamma} \mathbb{E}_{\tau \sim \mathbb{P}^{\boldsymbol{\pi}_{\boldsymbol{\theta}}}_{\boldsymbol{\pi}_{\boldsymbol{\theta}}}} \left[ \sum_{t=0}^{\infty} \gamma^t Q^{\boldsymbol{\pi}_{\boldsymbol{\theta}}}(s, a) \nabla_{\boldsymbol{\theta}} \log \boldsymbol{\pi}_{\boldsymbol{\theta}}(a \mid s) \right] = \mathbb{E}_{\tau \sim \mathbb{P}^{\boldsymbol{\pi}_{\boldsymbol{\theta}}}_{\boldsymbol{\pi}_{\boldsymbol{\theta}}}} \left[ \sum_{t=0}^{\infty} \gamma^t A^{\boldsymbol{\pi}_{\boldsymbol{\theta}}}(s, a) \nabla_{\boldsymbol{\theta}} \log \boldsymbol{\pi}_{\boldsymbol{\theta}}(a \mid s) \right].$$

Since the value function is not concave in the policy parameters, achieving optimality with PG has been a challenge. But practical scalability and efficiency of these algorithms has motivated a long-line of work to understand the minimum conditions and parametric forms of policies leading to convergence to the optimal policy (Agarwal et al., 2021; Mei et al., 2020; Wang & Zou, 2022; Yuan et al., 2022). *Our work extends these algorithmic techniques and theoretical insights to performative RL.*

### 2.2 PERFORMATIVE RL: INFINITE-HORIZON DISCOUNTED PEMDPS

Given a policy set $\boldsymbol{\pi} \in \Pi$, we denote the Performative Markov Decision Process (PeMDP) is defined as the set of MDPs $\{\mathcal{M}(\boldsymbol{\pi}) \mid \boldsymbol{\pi} \in \Pi\}$, where each MDP is a tuple $\mathcal{M}(\boldsymbol{\pi}) \triangleq (\mathcal{S}, \mathcal{A}, \mathbf{P}_{\boldsymbol{\pi}}, r_{\boldsymbol{\pi}}, \gamma)$. Note, that the transition kernel and rewards distribution are no more invariant with respect to the policy. They shift with the deployed policy $\boldsymbol{\pi} \in \Delta(\mathcal{A})$ (Mandal et al., 2023; Mandal & Radanovic, 2024). In this setting, the probability of generating a trajectory $\tau_{\boldsymbol{\pi}} \triangleq (s_t, a_t)_{t=0}^{\infty}$ under policy $\boldsymbol{\pi}$ with underlying MDP $\mathcal{M}(\boldsymbol{\pi}')$ is given by[1] $\mathbb{P}^{\boldsymbol{\pi}}_{\boldsymbol{\pi}'}(\tau \mid \boldsymbol{\rho}) \triangleq \rho(s_0) \prod_{t=0}^{\infty} \boldsymbol{\pi}(a_t \mid s_t) \mathbf{P}_{\boldsymbol{\pi}'}(s_{t+1} \mid s_t, a_t)$, where $\boldsymbol{\rho} \in \Delta(\mathcal{S})$ is the initial state distribution. Furthermore, the state-action occupancy measure for deployed policy $\boldsymbol{\pi}$ and environment-inducing policy $\boldsymbol{\pi}'$ is defined as $d^{\boldsymbol{\pi}}_{\boldsymbol{\pi}', \boldsymbol{\rho}} \triangleq (1 - \gamma) \mathbb{E}_{\tau \sim \mathbb{P}^{\boldsymbol{\pi}}_{\boldsymbol{\pi}'}} \left[ \sum_{t=0}^{\infty} \gamma^t \mathbb{1}(s_t = s, a_t = a) \mid s_0 \sim \boldsymbol{\rho} \right]$. Now, we are ready to define the performative expected return, referred as the performative value function that we aim to maximise while solving PeMDP.

**Definition 1** (Performative Value Function)**.** *Given a policy $\boldsymbol{\pi} \in \Pi$ and an initial state distribution $\boldsymbol{\rho} \in \Delta(S)$, the performative value function $V^{\boldsymbol{\pi}}_{\boldsymbol{\pi}}(\boldsymbol{\rho})$ is*

$$V^{\boldsymbol{\pi}}_{\boldsymbol{\pi}}(\boldsymbol{\rho}) \triangleq \mathbb{E}_{\tau \sim \mathbb{P}^{\boldsymbol{\pi}}_{\boldsymbol{\pi}}} \left[ \sum_{t=0}^{\infty} \gamma^t r_{\boldsymbol{\pi}}(s_t, \boldsymbol{\pi}(s_t)) \mid s_0 \sim \boldsymbol{\rho} \right]. \tag{1}$$

---

[1]Hereafter, for relevant quantities, $\boldsymbol{\pi}$ in superscript denotes the deployed policy, and $\boldsymbol{\pi}'$ in the subscript denoted the environment-inducing, i.e. the policy inducing the transition kernel and reward function that the algorithm interacts with.

Equation (2) gives the total expected return that captures the performativity aspect in PeMDPs as the underlying dynamics changes with a deployed policy $\boldsymbol{\pi}(\cdot \mid s)$.

On a similar note, we define the performative Q-value function (or action-value function) of a policy $\boldsymbol{\pi}$ as follows.

**Definition 2** (Performative Q-value). *Given a policy* $\boldsymbol{\pi} \in \Pi$ *and a state-action pair* $(s, a) \in (\mathcal{S}, \mathcal{A})$ *, the performative Q-value function* $Q_{\boldsymbol{\pi}}^{\boldsymbol{\pi}}(s, a)$ *is*

$$Q_{\boldsymbol{\pi}}^{\boldsymbol{\pi}}(s, a) \triangleq \mathbb{E}_{\tau \sim \mathbb{P}_{\boldsymbol{\pi}}^{\boldsymbol{\pi}}} \left[ \sum_{t=0}^{\infty} \gamma^t r_{\boldsymbol{\pi}}(s_t, a_t) \Big| s_0 = s, a_0 = a \right] \quad (2)$$

The Q-value satisfies the following Bellman equation:

$$Q_{\boldsymbol{\pi}}^{\boldsymbol{\pi}}(s, a) = r_{\boldsymbol{\pi}}(s, a) + \gamma \mathbb{E}_{s' \sim \mathbf{P}_{\boldsymbol{\pi}}(\cdot|s,a)} \left[ V_{\boldsymbol{\pi}}^{\boldsymbol{\pi}}(s') \right] \quad (3)$$

Note that, we can maximise performative value function in two ways: (i) considering $\boldsymbol{\pi}$ as both the environment-inducing policy and the policy the RL agent deploys, or (ii) deploying $\boldsymbol{\pi}$ to fix it as the environment-inducing policy and agent plays another policy $\boldsymbol{\pi}'$. At this vantage point, let us introduce the notion of optimality and stability of policies in PeMDPs (Mandal et al., 2023).

**Definition 3** (Performative Optimality). *A policy* $\boldsymbol{\pi}_o^{\star}$ *is performatively optimal if it maximizes the performative value function.*

$$\boldsymbol{\pi}_o^{\star} \in \arg\max_{\boldsymbol{\pi} \in \Delta(\mathcal{A})} V_{\boldsymbol{\pi}}^{\boldsymbol{\pi}}(\boldsymbol{\rho}) . \quad (4)$$

Thus, if we play the policy $\boldsymbol{\pi}$ in the environment induced by policy $\boldsymbol{\pi}$ to maximise the expected return, we land on the performatively optimal policy.

**Definition 4** (Performative Stability). *A policy* $\boldsymbol{\pi}_s^{\star}$ *is performatively stable if there is no gain in performative value function due to deploying any other policy than* $\boldsymbol{\pi}_s^{\star}$ *in the environment induced by* $\boldsymbol{\pi}_s^{\star}$.

$$\boldsymbol{\pi}_s^{\star} \in \arg\max_{\boldsymbol{\pi} \in \Delta(\mathcal{A})} V_{\boldsymbol{\pi}_s^{\star}}^{\boldsymbol{\pi}}(\boldsymbol{\rho}). \quad (5)$$

As noted by Mandal et al. (2023), a performatively optimal policy may not be performatively stable, i.e., $\boldsymbol{\pi}_o^{\star}$ may not be optimal for a changed underlying environment $\mathcal{M}(\boldsymbol{\pi}_o^{\star})$, when it is deployed. Also, in general, the performative value function of $\boldsymbol{\pi}_o^{\star}$ might be equal to or higher than that of $\boldsymbol{\pi}_s^{\star}$. In this paper, *we design PG algorithms computing the performative optimal policy for a given set of MDPs*, and reinstate their differences with performatively stable policies.

The existing literature on PeMDPs (Mandal et al., 2023; Mandal & Radanovic, 2024; Rank et al., 2024; Pollatos et al., 2025; Chen et al., 2024) focused primarily on finding a performatively stable policy, i.e. a $\boldsymbol{\pi}_s^{\star}$ according to Definition 4. In practice, while the notion of stable policies matters for very specific applications, a stable policy may not always suffice. But they might show large sub-optimality gaps, which are often not desired for real-life tasks. *We fill up this gap in literature and propose the first provably converging and computationally efficient PG algorithm for PeMDPs.* Later on, we also empirically show the deficiency of the existing stability finding algorithms if we aim for optimality (Section 5).

**Entropy Regularised PeMDPs.** Entropy regularisation has emerged as a simple but powerful technique in classical RL to design smooth and efficient algorithms with sufficient exploration. Thus, we study another variant of the performative value function that is regularised using discounted entropy (Mei et al., 2020; Neu et al., 2017; Liu et al., 2019; Zhao et al., 2019). In this setting, the original value function in Definition 1 is regularised using the discounted entropy $H_{\boldsymbol{\pi}}(\rho) \triangleq \mathbb{E}_{\tau \sim \mathbb{P}_{\boldsymbol{\pi}}^{\boldsymbol{\pi}}} \left[ -\sum_{t=0}^{\infty} \gamma^t \log \boldsymbol{\pi}(a_t \mid s_t) \right]$. This is equivalent to maximising the expected reward with a shifted reward function $\tilde{r}_{\boldsymbol{\pi}}(\boldsymbol{\pi}(s_t), s_t) = r_{\boldsymbol{\pi}}(\boldsymbol{\pi}(s_t), s_t) - \lambda \log(\boldsymbol{\pi}(a_t \mid s_t))$ for some $\lambda \geq 0$. $\tilde{r}_{\boldsymbol{\pi}}$ is referred as the "soft-reward" in MDP literature (Wang & Uchibe, 2024; Herman et al., 2016; Shi et al., 2019). This allows us to define the soft performative value function.

**Definition 5** (Entropy Regularised (or Soft) Performative Value Function). *Given a policy* $\boldsymbol{\pi} \in \Pi$, *a starting state distribution* $\boldsymbol{\rho} \in \Delta(S)$, *and a regularisation parameter* $\lambda \geq 0$, *the soft performative value function* $\tilde{V}_{\boldsymbol{\pi}}^{\boldsymbol{\pi}}(\boldsymbol{\rho})$ *is*

$$\tilde{V}_{\boldsymbol{\pi}}^{\boldsymbol{\pi}}(\boldsymbol{\rho}) \triangleq \mathbb{E}_{\tau \sim \mathbb{P}_{\boldsymbol{\pi}}^{\boldsymbol{\pi}}} \left[ \sum_{t=0}^{\infty} \gamma^t \left( r_{\boldsymbol{\pi}}(s_t, \boldsymbol{\pi}(s_t)) - \lambda \log \boldsymbol{\pi}(a_t \mid s_t) \right) \mid s_0 \sim \boldsymbol{\rho} \right] = \mathbb{E}_{\tau \sim \mathbb{P}_{\boldsymbol{\pi}}^{\boldsymbol{\pi}}} \left[ \sum_{t=0}^{\infty} \gamma^t \tilde{r}_{\boldsymbol{\pi}}(s_t, \boldsymbol{\pi}(s_t)) \mid s_0 \sim \boldsymbol{\rho} \right] . \quad (6)$$

Since policies belong to the probability simplex, the entropy regularisation naturally lends to smoother and stable PG algorithms. Later, we show that the discounted entropy is a smooth function of the policy parameters for PeMDPs extending the optimization-wise benefits of entropy regularisation to PeMDPs. Additionally, using the notion of soft rewards, we can further define soft performatively optimal and stable policies for entropy regularised PeMDPs. Leveraging it, *we unifiedly design PG algorithms for both the unregularised and the entropy regularised PeMDPs.*

# 3 POLICY GRADIENT ALGORITHMS IN PERFORMATIVE RL

In this section, we first study the impact of policy updates in PeMDPs. Then, we leverage it to derive the performative policy gradient theorem and design Performative Policy Gradient (PePG) algorithm for any differentiable parametric policy class.

## 3.1 IMPACT OF POLICY UPDATES ON PeMDPs

Performance difference lemma has been central in RL to understand the impact of changing policies in terms of value functions (Kakade & Langford, 2002a). It has been also central to analysing and developing PG-type methods (Agarwal et al., 2021; Silver et al., 2014; Kallel et al., 2024). But the existing versions of performance difference cannot handle performativity. Here, we derive the performative version of the performance difference lemma that quantifies the shift in the performative value function due to change the deployed and environment-inducing policies.

**Lemma 1** (Performative Performance Difference Lemma). *The difference in performative value functions induced by $\boldsymbol{\pi}$ and $\boldsymbol{\pi}' \in \Pi$ while starting from the initial state distribution $\boldsymbol{\rho}$ is*

$$V_{\boldsymbol{\pi}}^{\boldsymbol{\pi}}(\boldsymbol{\rho}) - V_{\boldsymbol{\pi}'}^{\boldsymbol{\pi}'}(\boldsymbol{\rho}) = \frac{1}{1-\gamma}\mathbb{E}_{(s,a)\sim\boldsymbol{d}_{\boldsymbol{\pi}',\boldsymbol{\rho}}^{\boldsymbol{\pi}}}[A_{\boldsymbol{\pi}'}^{\boldsymbol{\pi}'}(s,a)]$$
$$+ \frac{1}{1-\gamma}\mathbb{E}_{(s,a)\sim\boldsymbol{d}_{\boldsymbol{\pi}',\boldsymbol{\rho}}^{\boldsymbol{\pi}}}[(r_{\boldsymbol{\pi}}(s,a) - r_{\boldsymbol{\pi}'}(s,a)) + \gamma(\mathbf{P}_{\boldsymbol{\pi}}(\cdot|s,a) - \mathbf{P}_{\boldsymbol{\pi}'}(\cdot|s,a))^{\top}V_{\boldsymbol{\pi}}^{\boldsymbol{\pi}}(\cdot)]. \quad (7)$$

*where $A_{\boldsymbol{\pi}'}^{\boldsymbol{\pi}'}(s,a) \triangleq Q_{\boldsymbol{\pi}'}^{\boldsymbol{\pi}'}(s,a) - V_{\boldsymbol{\pi}'}^{\boldsymbol{\pi}'}(s)$ is the performative advantage function for any state $s \in \mathcal{S}$ and action $a \in \mathcal{A}$.*

The crux of the proof is decomposing the performative value through environment-inducing and deployed policies

$$V_{\boldsymbol{\pi}}^{\boldsymbol{\pi}}(s_0) - V_{\boldsymbol{\pi}'}^{\boldsymbol{\pi}'}(s_0) = \underbrace{V_{\boldsymbol{\pi}}^{\boldsymbol{\pi}}(s_0) - V_{\boldsymbol{\pi}'}^{\boldsymbol{\pi}}(s_0)}_{\text{performative shift term}} + \underbrace{V_{\boldsymbol{\pi}'}^{\boldsymbol{\pi}}(s_0) - V_{\boldsymbol{\pi}'}^{\boldsymbol{\pi}'}(s_0)}_{\text{performance difference term}}.$$

(1) *Connection to Classical RL.* In classical RL, the performance difference lemma yields $V^{\boldsymbol{\pi}}(\boldsymbol{\rho}) - V^{\boldsymbol{\pi}'}(\boldsymbol{\rho}) = \frac{1}{1-\gamma}\mathbb{E}_{(s,a)\sim\boldsymbol{d}_{\rho}^{\boldsymbol{\pi}}}[A^{\boldsymbol{\pi}'}(s,a)]$. The first term in Lemma 1 is equivalent to the classical result in the environment induced by $\boldsymbol{\pi}'$. But due to environment shift, two more terms appear in the performative performance difference incorporating the impacts of reward shifts and transition shifts. (2) *Connection to Performative Stability.* If we ignore the reward and transition shift terms, the performance difference term $V_{\boldsymbol{\pi}'}^{\boldsymbol{\pi}}(s_0) - V_{\boldsymbol{\pi}'}^{\boldsymbol{\pi}'}(s_0)$ quantifies the impact of changing the deployed policy from $\boldsymbol{\pi}'$ to $\boldsymbol{\pi}$ in an environment induced by $\boldsymbol{\pi}'$. Thus, a stability seeking algorithm would like to minimise this term, while an optimality seeking algorithm has to incorporate all of the terms.

Now, we ask: *how much do the new environment shift terms change the performative performance difference?*

For simplicity, we focus on the commonly studied PeMDPs with bounded rewards and gradually shifting environments, i.e. the ones with Lipschitz transitions and rewards with respect to the deployed policies (Rank et al., 2024).

**Assumption 1** (Bounded reward). *We assume that the rewards are bounded in $[-R_{\max}, R_{\max}]$.*

This is the only assumption needed through the paper and is standard in MDP literature (Mei et al., 2020; Li & Yang, 2023).

**Lemma 2** (Bounding Performative Performance Difference for Gradually Shifting Environments). *Let us assume that both rewards and transitions are Lipschitz functions of policy, i.e. $\|r_{\boldsymbol{\pi}} - r_{\boldsymbol{\pi}'}\|_{\infty} \leq L_r \|\boldsymbol{\pi} - \boldsymbol{\pi}'\|_1$ and $\|\mathbf{P}_{\boldsymbol{\pi}} - \mathbf{P}_{\boldsymbol{\pi}'}\|_1 \leq L_{\mathbf{P}} \|\boldsymbol{\pi} - \boldsymbol{\pi}'\|_1$, for some $L_r, L_{\mathbf{P}} \geq 0$. Then, under Assumption 1, the performative shift in the sub-optimality gap of a policy $\boldsymbol{\pi_\theta}$ satisfies*

$$\left| V_{\boldsymbol{\pi}_o^\star}^{\boldsymbol{\pi}_o^\star}(\boldsymbol{\rho}) - V_{\boldsymbol{\pi_\theta}}^{\boldsymbol{\pi_\theta}}(\boldsymbol{\rho}) - \frac{1}{1-\gamma}\mathbb{E}_{(s,a)\sim d_{\boldsymbol{\pi_\theta},\boldsymbol{\rho}}^{\boldsymbol{\pi}_o^\star}}[A_{\boldsymbol{\pi_\theta}}^{\boldsymbol{\pi_\theta}}(s,a)] \right| \leq \frac{2\sqrt{2}}{1-\gamma}(L_r + \frac{\gamma}{1-\gamma}L_{\mathbf{P}}R_{\max})\mathbb{E}_{s_0\sim\boldsymbol{\rho}}D_{\mathrm{H}}\left(\boldsymbol{\pi}_o^\star(\cdot|s_0)\|\boldsymbol{\pi_\theta}(\cdot|s_0)\right).$$

(8)

*where $D_{\mathrm{H}}(\mathbf{x}\|\mathbf{y})$ denotes the Hellinger distance between $\mathbf{x}$ and $\mathbf{y}$.*

*Implication.* Lemma 2 shows novel characterisation of the *extra cost* we pay to adapt to performativity of the environment in terms of Hellinger distance between the true performatively optimal policy $\boldsymbol{\pi}_o^\star$ and any other parametrised policy $\boldsymbol{\pi_\theta}$. This implies that the order of difference between the optimal performative value function and that of any stability-seeking algorithm is $\Theta(\frac{1}{1-\gamma})$. This significantly improves the known order of sub-optimality achieved by existing algorithms. Specifically, Mandal et al. (2023) show that using repeated policy optimisation algorithms converges to a suboptimality gap

---

**Algorithm 2** PePG: **Pe**rformative **P**olicy **G**radient

1: **Input:** Transition Feature Map $\psi(s) \forall s \in \mathcal{S}$, $\xi \in [-R_{\max}, R_{\max}]$ and discount factor $\gamma$.
2: **Initialize:** Initial policy parameters $\theta_0$, initial value function parameters $\phi_0$
3: **for** $k = 1, 2, \dots$ **do**
4:     **Collect trajectories:** $\mathcal{D}_k = \{\tau_i\}_{i=1}^{I}$, where each $\tau_i \triangleq \{(s_{i,t}, a_{i,t}, s_{i,t+1}, r_{i,t})\}_{t=0}^{T-1}$ by playing $\pi_{\theta_k} = \pi(\theta_k)$
5:     Compute returns $R_k \triangleq \{R_{k,i}\}_{i=1}^{I}$, where $R_{k,i} = \{R_{k,i,t}\}_{t=0}^{T-1}$
6:     Compute advantage estimates $\hat{A}_k(\tau_i)$ using value function $\hat{V}_{\phi_k}(\tau_i)$ for each $\tau_i \in \mathcal{D}_k$ (estimate of $V_{\pi_{\theta_k}}^{\pi_{\theta_k}}(\tau_i)$ obtained from fitted value network with parameters $\phi_k$)
7:     **Gradient estimation:** Estimate policy gradient using (12)
8:     **Gradient ascent step:** Update policy parameters using (9)
9:     Fit value function $V_{\phi_{k+1}}$:

$$\phi_{k+1} \leftarrow \arg\min_{\phi} \frac{1}{I \cdot T} \sum_{i=1}^{I} \sum_{t=0}^{T-1} \left( \hat{V}_{\phi_k}(s_t \in \tau_i) - R_{k,i,t} \right)^2$$

10: **end for**

---

$\mathcal{O}\left(\max\{\frac{S^{5/3}A^{1/3}\epsilon^{2/3}}{(1-\gamma)^{14/3}}, \frac{\epsilon S}{(1-\gamma)^4}\}\right)$. Thus, we see an opportunity to improve on the existing works and design algorithms that can achieve suboptimality gap of order $\Theta(\frac{1}{1-\gamma})$.

Additionally, we note that an optimality-seeking algorithm tries to minimise both the advantage function and the effect of the shifts in the environment quantified by the Hellinger distance, i.e., $D_{\mathrm{H}}\left(\pi_o^{\star}(\cdot|s_0) \| \pi_{\theta}(\cdot|s_0)\right)$. While it suffices for a stability-seeking algorithm to minimise the advantage function, and thus, we cannot minimise the RHS of Equation (8) lower than $D_{\mathrm{H}}\left(\pi_o^{\star}(\cdot|s_0) \| \pi_{\theta}(\cdot|s_0)\right)$. Thus, optimality-seeking algorithms can achieve a lower performative performance difference than the stability-seeking algorithms if they also learn and incorporate the performative shifts in the environment.

### 3.2 ALGORITHM DESIGN: PERFORMATIVE POLICY GRADIENT (PePG)

To achieve performative optimality, the goal is to maximise value function at the end of learning process. Gradient ascent is a standard first-order optimisation method to find maxima of a function. Similar to Algorithm 1, the crux of performative policy gradient method lies in the ascent step:

$$\theta_{t+1} \leftarrow \begin{cases} \theta_t + \eta_t \nabla_{\theta} V_{\pi_{\theta}}^{\pi_{\theta}}(\tau) \mid_{\theta=\theta_t} & \text{, for unregularised objective} \\ \theta_t + \eta_t \nabla_{\theta} \tilde{V}_{\pi_{\theta}}^{\pi_{\theta}}(\tau) \mid_{\theta=\theta_t} & \text{, for Entropy-regularised objective.} \end{cases} \quad (9)$$

Given this ascent step, we have to evaluate the gradient at each time step from the rollouts of the present policy. In classical PG, the policy gradient theorem serves this purpose (Williams, 1992; Sutton et al., 1999; Silver et al., 2014). Thus, we derive the performative counterpart of the classic policy gradient theorem.

**Theorem 2** (Performative Policy Gradient Theorem). *The gradient of the performative value function w.r.t $\theta$ is as follows:*

*(a) For the unregularised objective,*

$$\nabla_{\theta} V_{\pi_{\theta}}^{\pi_{\theta}}(\tau) = \mathbb{E}_{\tau \sim \mathbb{P}_{\pi_{\theta}}^{\pi_{\theta}}}\left[ \sum_{t=0}^{\infty} \gamma^t \left( A_{\pi_{\theta}}^{\pi_{\theta}}(s_t, a_t) \left( \nabla_{\theta} \log \pi_{\theta}(a_t \mid s_t) + \nabla_{\theta} \log P_{\pi_{\theta}}(s_{t+1}|s_t, a_t) \right) + \nabla_{\theta} r_{\pi_{\theta}}(s_t, a_t) \right) \right], \quad (10)$$

*(b) For the entropy-regularised objective, we define the soft advantage, soft Q, and soft value functions with respect to the soft rewards $\tilde{r}_{\pi_{\theta}}$ satisfying $\tilde{A}_{\pi_{\theta}}^{\pi_{\theta}}(s, a) = \tilde{Q}_{\pi_{\theta}}^{\pi_{\theta}}(s, a) - \tilde{V}_{\pi_{\theta}}^{\pi_{\theta}}(s)$ that further yields*

$$\nabla_{\theta} \tilde{V}_{\pi_{\theta}}^{\pi_{\theta}}(\tau) = \mathbb{E}_{\tau \sim \mathbb{P}_{\pi_{\theta}}^{\pi_{\theta}}}\left[ \sum_{t=0}^{\infty} \gamma^t \left( \tilde{A}_{\pi_{\theta}}^{\pi_{\theta}}(s_t, a_t) \left( \nabla_{\theta} \log \pi_{\theta}(a_t \mid s_t) + \nabla_{\theta} \log P_{\pi_{\theta}}(s_{t+1}|s_t, a_t) \right) + \nabla_{\theta} \tilde{r}_{\pi_{\theta}}(s_t, a_t | \theta) \right) \right]. \quad (11)$$

**PePG:** To elaborate on the design of PePG (Algorithm 2), we focus only on the REINFORCE update and softmax policy parametrisation. With the appropriate parameter choices, and initialisation of the policy parameter $\theta$ and value function parameter $\phi$, for each episode $k = 1, 2, \dots$, PePG collects $I$ trajectories to calculate return $R^i$ and estimates advantage

function $\hat{A}_k$ (Line 4-6). For a particular trajectory $\tau_i$, the estimated advantage for a given state-action is $\widehat{A^{\pi_{\theta_k}}_{\pi_{\theta_k}}(s^i_t, a^i_t)} = R^i_{t,k} - V_{\phi_k}(s^i)$, where $R^i = \sum_{t=0}^{T-1} \gamma^t r_{\pi_{\theta_k}}(s^i_t, a^i_t)$.

**Gradient Estimation (Line 7).** With the necessary estimates in hand for all the collected $I$ trajectories, PePG computes average gradient estimate over all the trajectories using

$$\widehat{\nabla_{\theta_k} V^{\pi_{\theta_k}}_{\pi_{\theta_k}}} = \frac{1}{I} \sum_{i=1}^{I} \sum_{t=0}^{T} \gamma^t (\widehat{A^{\pi_{\theta_k}}_{\pi_{\theta_k}}(s^i_t, a^i_t)}) \left( \nabla_{\theta_k} \log \pi_{\theta_k}(a^i_t \mid s^i_t) + \nabla_{\theta_k} \log P_{\pi_{\theta_k}}(s^i_{t+1} | s^i_t, a^i_t) \right) + \nabla_{\theta_k} r_{\pi_{\theta_k}}(s^i_t, a^i_t | \theta_k)) \tag{12}$$

where all the individual gradients $\nabla_{\theta_k} \log P_{\pi_{\theta_k}}, \nabla_{\theta_k} r_{\pi_{\theta_k}}$ and $\nabla_{\theta_k} \log \pi_{\theta_k}$ have the closed form expressions for softmax parametrisation according to Equation (35). Further, in Line 8, PePG updates the policy parameter for the next episode using a gradient ascent step leveraging the estimated average gradient over all $I$ trajectories. Specifically, we plug in $\widehat{\nabla_{\theta_k} V^{\pi_{\theta_k}}_{\pi_{\theta_k}}}$ to both the unregularised and entropy-regularised update rules are given in Equation (9). For the next episode, we again run a regression to update the value network plugging in the current estimates and resume the learning process further.

## 4 CONVERGENCE ANALYSIS OF PePG: SOFTMAX POLICIES AND SOFTMAX PeMDPs

For rigorous theoretical analysis of PePG, we restrict ourselves to *softmax policy class*, and *softmax PeMDPs*. We define the softmax PeMDPs as the ones having softmax transition kernesls with feature map $\psi(\cdot) : \mathcal{S} \to \mathbb{R}$, and linear reward functions with respect to the policy parameters, for all state $s \in \mathcal{S}$ and action $a \in \mathcal{A}$. Specifically, the class of softmax PeMDPs is $\{\mathcal{M}(\theta) = \mathcal{M}(\pi_\theta) \mid \theta \in \mathbb{R}^{|\mathcal{S}| \times |\mathcal{A}|}\}$ such that

$$\pi_\theta(a|s) = \frac{e^{\theta_{s,a}}}{\sum_{a'} e^{\theta_{s,a'}}} , \quad \mathbf{P}_{\pi_\theta}(s'|s,a) = \frac{e^{\theta_{s,a}\psi(s')}}{\sum_{s''} e^{\theta_{s,a}\psi(s'')}} , \quad r_\theta(s,a) = \mathcal{P}_{[-R_{\max}, R_{\max}]}[\xi \theta_{s,a}] , \tag{13}$$

where $\psi$ is non-negative and upper bounded by $\psi_{\max} > 0$, and $\xi \in [0, R_{\max}]$ to align with Assumption 1.

Thus, we derive the derivatives of policy, transitions, and rewards as

$$\frac{\partial}{\partial \theta_{s',a'}} \log \pi_\theta(a|s) = \mathbb{1}[s = s', a = a'] - \pi_\theta(a'|s)\mathbb{1}[s = s'],$$

$$\frac{\partial}{\partial \theta_{s',a'}} \log \mathbf{P}_{\pi_\theta}(s''|s,a) = \psi(s'')\mathbb{1}[s = s', a = a'](1 - \mathbf{P}_{\pi_\theta}(s''|s,a)) , \quad \frac{\partial}{\partial \theta_{s',a'}} r_{\pi_\theta}(s,a) = \xi \mathbb{1}[s = s', a = a'] . \tag{14}$$

Given the derivatives, we can now readily estimate the policy gradient and deploy PePG for softmax PeMDPs.

**Convergence Analysis: Challenges and Three Step Analysis.** The main challenge to prove convergence of PePG is that the performative value function is not concave in the paramterisation $\theta$, in general, and also in softmax PeMDPs. The similar issue occurs while proving convergence of PG-type algorithms in classical RL, which has been overcome by leveraging smoothness properties of the value functions and by deriving the local Polyak-Lojasiewicz (PL)-type conditions, known as *gradient domination*, with respect to the policy paramterisation. Leveraging these insights, we devise a three step convergence analysis for PePG.

**Step 1: Smoothness of Performative Value Functions.** First, we prove that the unregularised performative value function is $\mathcal{O}(\frac{|\mathcal{A}|}{(1-\gamma)^2})$ smooth. As we show that the entropy is also a smooth function for softmax PeMDPs, then under proper choice of the regularisation parameter, i.e., $\lambda = \frac{1-\gamma}{1+2\log|\mathcal{A}|}$, entropy regularised performative value function is also $\mathcal{O}(\frac{|\mathcal{A}|}{(1-\gamma)^2})$ smooth. Since gradient ascent/descent methods can work well in smooth functions, we proceed thoroughly.

**Step 2: Gradient Domination for Softmax PeMDPs.** Now, the next step is to relate the performative performance difference with the performative policy gradient. This allows us to connect the per iteration improvement in the performative value function with the performative gradient descent at that step. These are known as PL-type inequalities. For non-concave objectives, PL inequalities guarantee convergence to global maxima by showing that the gradient of the objective at any parameter dominates the sub-optimality w.r.t. that parameter.

**Lemma 3** (Performative Gradient Domination for Softmax PeMDPs)**.** *Let us consider PeMDPs defined in* (13).

*(a) For unregularised value function,*

$$V^{\pi^\star_o}_{\pi^\star_o}(\rho) - V^{\pi_\theta}_{\pi_\theta}(\rho) \leq \sqrt{|\mathcal{S}||\mathcal{A}|} \left\| \frac{d^{\pi^\star_o}_{\pi_\theta, \rho}}{d^{\pi_\theta}_{\pi_\theta, \nu}} \right\|_\infty \|\nabla_\theta V^{\pi_\theta}_{\pi_\theta}(\nu)\|_2 + \frac{R_{\max}}{1-\gamma} \left( 1 + \frac{2\gamma}{1-\gamma} \psi_{\max} \right) . \tag{15}$$

| Algorithms | Regulariser $\lambda$ | Min. #samples | Environment |
|---|---|---|---|
| RPO FS (Mandal et al., 2023) | $\mathcal{O}\left(\frac{|\mathcal{S}|+\gamma|\mathcal{S}|^{5/2}}{(1-\omega)(1-\gamma)^4}\right)$ | $\frac{|\mathcal{A}|^2|\mathcal{S}|^3}{\epsilon^4(1-\gamma)^6\lambda^2}\ln(\#\text{iter})$ | Direct PeMDPs + quadratic-regul. on occupancy $\omega$-dependence between two envs. |
| MDRR (Rank et al., 2024) | $\mathcal{O}\left(\frac{|\mathcal{S}|+\gamma|\mathcal{S}|^{5/2}}{(1-\omega)(1-\gamma)^4}\right)$ | $\frac{|\mathcal{A}|^2|\mathcal{S}|^3}{\epsilon^4(1-\gamma)^6\lambda^2}\ln(\#\text{iter})$ | Direct PeMDPs + quadratic-regul. on occupancy $\omega$-dependence between two envs. |
| PePG (This paper) | $\frac{R_{\max}(1-\gamma)}{1+\log(|\mathcal{A}|)}$ | $\frac{|\mathcal{S}||\mathcal{A}|^2}{\epsilon^2(1-\gamma)^3}$ | softmax PeMDPs + entropy regul. on policy |
| PePG (This paper) | $0$ | $\frac{|\mathcal{S}||\mathcal{A}|}{\epsilon^2}\max\left\{\frac{\gamma R_{\max}|\mathcal{A}|}{(1-\gamma)^3},\frac{\gamma^2}{(1-\gamma)^4}\right\}$ | unregularised softmax PeMDPs |

Table 1: Comparison of theoretical performance of SOTA stability-seeking algorithms against PePG.

*(b) For entropy-regularised value function, $\tilde{V}_{\pi_o^\star}^{\pi_o^\star}(\rho) - \tilde{V}_{\pi_\theta}^{\pi_\theta}(\rho) \leq$*

$$\sqrt{|\mathcal{S}||\mathcal{A}|}\left\|\frac{d_{\pi_\theta,\rho}^{\pi_o^\star}}{d_{\pi_\theta,\nu}^{\pi_\theta}}\right\|_\infty \|\nabla_\theta V_{\pi_\theta}^{\pi_\theta}(\nu)\|_2 + \frac{R_{\max}}{1-\gamma}\left(1+\frac{2\gamma}{1-\gamma}\psi_{\max}\left(1+\frac{\lambda}{R_{\max}}\log|\mathcal{A}|\right)\right) + \frac{\lambda}{1-\gamma}(1+2\log|\mathcal{A}|). \quad (16)$$

**Step 3: Iterative Application of Gradient Domination for Smooth Functions.** Now, we can apply gradient domination along with the classic iterative convergence proof of gradient ascent for smooth functions. The intuition is that since the per-step sub-optimality is dominated by the gradient and the smooth functions are bounded by quadratic envelopes of parameters, applying gradient ascent iteratively would bring the sub-optimality down to small error level after enough iterations. We formalise this in Theorem 3.

**Theorem 3** (Convergence of PePG in softmax PeMDPs). *Let* $\mathrm{Cov} \triangleq \max_{\theta,\nu}\left\|\frac{d_{\pi_\theta,\rho}^{\pi_o^\star}}{d_{\pi_\theta,\nu}^{\pi_\theta}}\right\|_\infty$ *. The gradient ascent algorithm on* $V_{\pi_\theta}^{\pi_\theta}(\rho)$ *(Equation (9)) satisfies, for all distributions* $\rho \in \Delta(\mathcal{S})$.

*(a) in the unregularised case with* $\eta = \Omega(\min\{\frac{(1-\gamma)^2}{\gamma|\mathcal{A}|},\frac{(1-\gamma)^3}{\gamma^2}\})$, $\min_{t<T}\left\{V_{\pi_o^\star}^{\pi_o^\star}(\rho) - V_{\pi_{\theta_t}}^{\pi_{\theta_t}}(\rho)\right\} \leq \epsilon + \mathcal{O}\left(\frac{1}{1-\gamma}\right)$ *when* $T = \Omega\left(\frac{|\mathcal{S}||\mathcal{A}|}{\epsilon^2}\max\left\{\frac{\gamma R_{\max}|\mathcal{A}|}{(1-\gamma)^3},\frac{\gamma^2}{(1-\gamma)^4}\right\}\right).$

*(b) in the entropy regularisation scenario with* $\lambda = \frac{(1-\gamma)}{1+2\log|\mathcal{A}|}$ *and* $\eta = \Omega\left(\frac{(1-\gamma)^2}{\gamma|\mathcal{A}|}\right)$, $\min_{t<T}\left\{\tilde{V}_{\pi_o^\star}^{\pi_o^\star}(\rho) - \tilde{V}_{\pi_\theta^{(t)}}^{\pi_\theta^{(t)}}(\rho)\right\} \leq \epsilon + \mathcal{O}\left(\frac{1}{1-\gamma}\right)$ *when* $T = \Omega\left(\frac{|\mathcal{S}||\mathcal{A}|^2\mathrm{Cov}^2}{\epsilon^2(1-\gamma)^3}\right).$

**Implications.** (1) We observe that PePG converges to an $\epsilon$-optimal policy in $\frac{|\mathcal{S}||\mathcal{A}|^2}{\epsilon^2(1-\gamma)^3}$ iterations. This reduces the sample complexity required for the existing stability-seeking algorithms by at least an order $\frac{|\mathcal{S}|^2}{\epsilon^2(1-\gamma)^3}$, and shows efficiency of using PePG than the algorithms directly optimising the occupancy measures. (2) Additionally, the regularisation parameters needed for the existing algorithms are pretty big and bigger than $\frac{|\mathcal{S}|}{(1-\gamma)^4}$. This is counter-intuitive and does not match the experimental observations. Here, we prove that setting the regularisation parameter to $\frac{1-\gamma}{1+2\log|\mathcal{A}|}$ suffices for proving convergence to optimality. (3) The minimum number of samples required to achieve convergence is proportional to the square of coverage for the softmax PeMDP. This is a ubiquitous quantity dictating convergence of PG-methods in classical RL (Agarwal et al., 2021; Mei et al., 2020), and retraining methods in performative RL (Mandal et al., 2023; Rank et al., 2024). (4) The $\mathcal{O}\left(\frac{1}{1-\gamma}\right)$ suboptimality gap appearing in Theorem 3 is analogous to the effect of using relaxed weak gradient domination result (Yuan et al., 2022, Corollary 3.7). It argues that if the policy gradient in classical MDPs satisfies the relaxed weak gradient domination, i.e., $\epsilon' + \|\nabla_\theta V(\theta)\| \geq 2\sqrt{\mu}(V^* - V(\theta))$ for some $\mu > 0$ and $\epsilon' > 0$, then the corresponding policy gradient method guarantees $\min_{t\in\{0,...,T\}}(V^* - V(\theta_t)) \leq \mathcal{O}(\epsilon) + \mathcal{O}(\epsilon')$ for big enough $T$. Lemma 3 constructs the performative counterpart of this relaxed weak gradient domination property with $\epsilon' = \mathcal{O}\left(\frac{1}{1-\gamma}\right)$. Similarly, (Sahitaj et al., 2025) also supports existence of such a gap empirically for Markov potential games. Thus, this indicates an inherent property of performative policy gradient which has to incorporate gradients of transitions and rewards along with gradients of policies at every step.

## 5 EXPERIMENTAL ANALYSIS

In this section, we empirically compare the performance of PePG in the performative reinforcement learning setting and analyse its behaviour against the state-of-the-art stability-finding methods. [2]

**Performative RL Environment.** We evaluate PePG in the Gridworld test-bed (Mandal et al., 2023), which has become a standard benchmark in performative RL. This environment consists of a grid where two agents $A_1$ (the principal) and $A_2$ (the follower), jointly control an actor navigating from start positions (S) to the goal (G) while avoiding hazards. The environment dynamics are as follows: Agent $A_1$ proposes a control policy for the actor by selecting one of four directional actions. Agent $A_2$ can either accept this action (not intervene) or override it with its own directional choice. *This creates a performative environment for $A_1$, as its effective policy outcomes depend on $A_2$'s responses to its deployed strategy.*

The cost structure follows: visiting blank cells (S) incurs penalty of $-0.01$, goal cells (F) cost $-0.02$, hazard cells (H) impose a severe penalty of $-0.5$, and any intervention by $A_2$ results in an additional cost of $-0.05$ for the intervening agent. The response model also follows that of Mandal et al. (2023), i.e., the agent $A_2$ responds to $A_1$'s policy using a Boltzmann softmax operator. Given $A_1$'s current policy $\boldsymbol{\pi}_1$, we compute the optimal Q-function $Q^{*|\boldsymbol{\pi}_1}$ for each follower agent $A_j$ relative to a perturbed version of the grid world, where each cell types matches $A_1$'s environment with probability $0.7$. We then define an average Q-function over the follower agents and determine the collective response policy via Boltzmann softmax $Q^{*|\pi_1}(s, a) = \frac{1}{n} \sum_{j=2}^{n+1} Q_j^{*|\pi_1}(s, a), \pi_2(a|s) = \frac{\exp(\beta \cdot Q^{*|\pi_1}(s,a))}{\sum_{a'} \exp(\beta \cdot Q^{*|\pi_1}(s,a'))}.$

It is important to note that our experimental setup deliberately uses the immediate response model from the original performative RL framework, rather than the gradually shifting environment introduced by Rank et al. (2024) that assumes slow shifts in the environment. Our choice to use the immediate response model presents a more challenging performative setting where the environment responds instantaneously to policy changes. This allows us to demonstrate that unlike MDRR (Rank et al., 2024), PePG can handle the fundamental performative challenge without requiring environmental assumptions that artificially slows down the feedback loop, thereby highlighting the robustness of the proposed PePG approach.

**Experimental Setup.** We evaluate PePG (with and without entropy regularisation) alongside Mixed Delayed Repeated Retraining (MDRR), which represents the current state-of-the-art in performative reinforcement learning under gradually shifting environments (Rank et al., 2024), and Repeated Policy Optimization with Finite Samples (RPO FS). MDRR has demonstrated significant improvements over traditional repeated retraining methods, by leveraging historical data from multiple deployments, while RPO FS is included as the baseline method from (Mandal et al., 2023) for direct comparison with the original performative RL approach.

All experiments use a $8 \times 8$ grid with $\gamma = 0.9$, exploration parameter $\epsilon = 0.5$ for initial policy construction, one follower agent $A_2$, and 100 trajectory samples per iteration. The algorithms share common parameters of $T = 100$ iterations. For regularization, RPO FS and MDRR use $\lambda = 0.1$ from their original experiments, while entropy-regularized PePG uses $\lambda = 2.0$ (ablation studies for this choice are provided in the appendix). PePG uses learning rate $\eta = 0.1$, MDRR employs memory weight $v = 1.1$ for historical data utilization, delayed round parameter $k = 3$, and FTRL parameters $N = B = 10$, while RPO FS follows the finite-sample optimization from Mandal et al.

**Results and Observations.** Our experimental evaluation across 100 iterations reveals fundamental differences between PePG and MDRR and RPO in the immediate response performative setting. We used shorter training compared to (Rank et al., 2024), as this time-frame sufficiently demonstrates RPO and MDRR's stability convergence and PePG's progression toward optimality.

**I. Results: Optimality:** The left panel reveals a clear performance hierarchy among the four methods. PePG achieves the highest value function performance, with standard PePG reaching approximately $0.1$ and regularized PePG (Reg PePG) reaching $0.05$, both showing consistent improvement from initial values around $-0.15$ and still progressing upward at the end of the 100 iteration window. This steady upward progression highlights PePG's effectiveness in discovering better performative equilibria rather than settling for the first stable solution encountered. RPO FS remains relatively stable around $-0.05$ throughout training, while MDRR stabilizes at the lowest performance level of approximately $-0.2$ and remains flat throughout training.

**II. Results: Comparison of Optimality- and Stability-seeking Algorithms.** The results expose a critical limitation of algorithms designed primarily for stability rather than optimality. MDRR successfully achieves its design goal, with the right panel showing decreasing toward zero in the stability metric $\|d_{t+1} - d_t\|_2$ (the $L_2$ distance between occupancy measures of consecutive policy iterations), indicating policy stabilization. However, this stability comes at the cost of solution quality, as MDRR becomes trapped in a suboptimal point. The method prioritised finding any stable point over finding an optimal

---

[2]Anonymous code repository of PePG implementation is Link. Further ablation studies w.r.t. hyperparameters are in Appendix H.

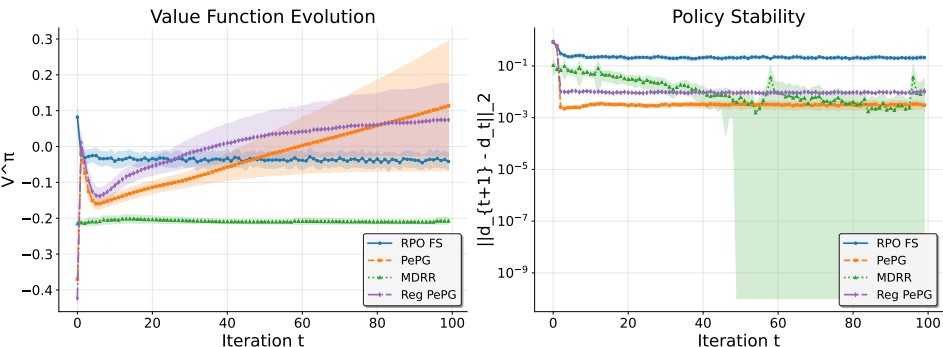

Figure 2: Comparison of evolution in expected average return (both regularised and unregularised) and stability of PePG with SOTA stability-achieving methods. Each algorithm is run for 20 random seeds and 100 iterations.

solution. In contrast, both PePG variants exhibit higher policy variability as they actively explore for better solutions. RPO FS maintains moderate stability around $10^{-1}$ but with limited performance improvement.

## 6  DISCUSSIONS, LIMITATIONS, AND FUTURE WORKS

We study the problem of Performative Reinforcement learning in tabular MDPs (PeMDPs) using softmax parametrised policies with entropy-regularised objective function, where any action taken by the agent cause potential shift in the MDP's underlying reward and transition dynamics. We are the first to develop PG-type algorithm, PePG, that attains performatively optimality against the existing performative stability-seeking algorithms, affirmatively solving an extended open problem in (Mandal et al., 2023). We also derive the novel performative counterpart of classic Performance Difference Lemma and Policy Gradient Theorem that affirmatively captures this performative nature of the environment we act. We provide a sufficient conditions to prove that PePG converges to an $\left(\epsilon + \frac{1}{1-\gamma}\right)$-ball around performative optimal policy in $\Omega\left(\frac{|\mathcal{S}||\mathcal{A}|^2}{\epsilon^2(1-\gamma)^3}\right)$ iterations.

As we develop a PG-type algorithm, it will be interesting to see how much can we reduce variance (Wu et al., 2018; Papini et al., 2018) while achieving optimality. We are still in the tabular setting with finite set of state-actions. A potential future direction would be to scale PePG to continuous state-space with large number of state-actions.

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

# Appendix

## Table of Contents

# A NOTATIONS

| Notation | Description |
|---|---|
| $\mathcal{S}$ | state space |
| $\mathcal{A}$ | action space |
| $\gamma$ | discount factor |
| $\pi_{\boldsymbol{\theta}}$ | policy parametrized by $\boldsymbol{\theta}$ |
| $\Pi(\Theta)$ | policy space |
| $\mathbf{P}_{\boldsymbol{\pi}}$ | transition under the environment induced by policy $\boldsymbol{\pi}$ |
| $r_{\boldsymbol{\pi}}$ | reward under the environment induced by policy $\boldsymbol{\pi}$ |
| $\boldsymbol{\pi}_s^{\star}$ | performatively stable policy |
| $\boldsymbol{\pi}_o^{\star}$ | performatively optimal policy |
| $\mathbf{P}_{\boldsymbol{\pi}_o^{\star}}$ | reward under the environment induced by performatively optimal policy |
| $r_{\boldsymbol{\pi}_o^{\star}}$ | reward under the environment induced by performatively optimal policy |
| $d_{\boldsymbol{\pi}_o^{\star}}^{\boldsymbol{\pi}_o^{\star}}$ | state-action occupancy of optimal policy |
| $V_{\boldsymbol{\pi}_o^{\star}}^{\boldsymbol{\pi}_o^{\star}}$ | value function of optimal policy |
| $d_{\boldsymbol{\pi}_2}^{\boldsymbol{\pi}_1}$ | state-action occupancy of playing policy $\boldsymbol{\pi}_2$ in the environment induced by policy $\boldsymbol{\pi}_1$ |
| $V_{\boldsymbol{\pi}_1}^{\boldsymbol{\pi}_2}$ | value function for playing policy $\boldsymbol{\pi}_2$ in the environment induced by policy $\boldsymbol{\pi}_1$ |
| $Q_{\boldsymbol{\pi}_1}^{\boldsymbol{\pi}_2}$ | Q-value function for playing policy $\boldsymbol{\pi}_2$ in the environment induced by policy $\boldsymbol{\pi}_1$ |
| $A_{\boldsymbol{\pi}_1}^{\boldsymbol{\pi}_2}$ | advantage function for playing policy $\boldsymbol{\pi}_2$ in the environment induced by policy $\boldsymbol{\pi}_1$ |
| $\Delta_K$ | $K$-dimensional simplex |
| $\boldsymbol{\rho}$ | Initial state distribution $\in \Delta_{\mathcal{S}}$ |

# B DETAILS OF THE TOY EXAMPLE: LOAN APPROVEMENT PROBLEM

*Environment.* We consider a population of loan applicants represented by a scalar feature $x \in \mathbb{R}$, distributed as $x \sim \mathcal{N}(\mu, \sigma^2)$, where $\mu$ is the population mean and $\sigma > 0$ is fixed.

*Bank's Policy.* The bank chooses a *threshold policy* parameterized by $\theta \in \mathbb{R}$. A loan is granted to an applicant $x$ if $x \geq \theta$. To smooth analysis, we use a differentiable approximation: $\pi_\theta(x) = \sigma(k(x - \theta))$, where $\sigma(z) = \frac{1}{1+e^{-z}}$ is the logistic sigmoid and $k > 0$ controls smoothness.

*Rewards.* If a loan is granted to applicant $x$, the bank receives a random payoff:

$$r(x) = \begin{cases} +R & \text{if applicant repays,} \\ -L & \text{if applicant defaults,} \end{cases}$$

with repayment probability $\mathbb{P}(\text{repay} \mid x) = \sigma(\gamma x - c)$, where $\gamma > 0$ controls sensitivity and $c$ is a calibration constant. The expected reward from granting to $x$ is

$$u(x) = \sigma(\gamma x - c) \cdot R - \big(1 - \sigma(\gamma x - c)\big) \cdot L.$$

*Expected Utility.* Given distribution $x \sim \mathcal{N}(\mu, \sigma^2)$, the bank's expected utility for policy $\theta$ is

$$U(\theta, \mu) = \mathbb{E}_{x \sim \mathcal{N}(\mu, \sigma^2)}\big[\pi_\theta(x) \cdot u(x)\big].$$

*Performative Feedback.* The population mean $\mu$ depends on the bank's policy, via the grant rate: $g(\theta, \mu) = \mathbb{E}_{x \sim \mathcal{N}(\mu, \sigma^2)}\big[\pi_\theta(x)\big]$.

We assume a bounded performative update rule: $\mu_{t+1} = (1 - \beta)\mu_t + \beta \cdot f\big(g(\theta, \mu_t)\big)$, where $\beta \in [0, 1]$ is the performative strength and $f(g) \in [-M, M]$ maps the grant rate to a feasible population mean.

At equilibrium, the induced feature distribution satisfies the fixed point condition:

$$\mu^*(\theta) = (1 - \beta)\mu^*(\theta) + \beta f\big(g(\theta, \mu^*(\theta))\big).$$

*Optimization Problems.* **ERM Optimum.** Ignoring performative effects (i.e. assuming $\mu = \mu_0$ is fixed), the ERM-optimal policy solves

$$\theta^{\text{ERM}} = \arg\max_\theta U(\theta, \mu_0).$$

**Performative Optimum.** Accounting for performative feedback, the performative-optimal policy solves

$$\theta^{\text{Perf}} = \arg\max_\theta U\big(\theta, \mu^*(\theta)\big).$$

*Learning via Reinforcement Learning*

An RL agent plays policies $\theta_t$ sequentially. At each round $t$:

1. Sample $x \sim \mathcal{N}(\mu_t, \sigma^2)$.
2. Grant loan with probability $\pi_{\theta_t}(x)$.
3. Observe reward $r_t$.
4. Update $\theta_{t+1}$ using policy gradient (REINFORCE).
5. Update population mean via performative dynamics:

$$\mu_{t+1} = (1 - \beta)\mu_t + \beta f\big(g(\theta_t, \mu_t)\big).$$

## C  DETAILED RELATED WORKS

**Performative Prediction.** The study of performative prediction started with the pioneering work of (Perdomo et al., 2020), where they leveraged repeated retraining with the aim to converge towards a performatively stable point. We see extension of this work trying to achieve performative optimality (Izzo et al., 2021; 2022; Miller et al., 2021). This further opened a plethora of works in various other domains such as Multi-agent systems (Narang et al., 2023; Li et al., 2022; Piliouras & Yu, 2023), control systems (Cai et al., 2024; Barakat et al., 2025), stochastic optimisation (Li & Wai, 2022; Mendler-Dünner et al., 2020), games (Wang et al., 2023; Góis et al., 2024) etc. There has been several attempt of achieve performative optimality or stability for real-life tasks like recommendation (Eilat & Rosenfeld, 2023), to measure the power of firms (Hardt et al., 2022; Mofakhami et al., 2023), in healthcare (Zhang et al., 2022) etc. Another interesting setting is the *stateful* performative prediction i.e. prediction under gradual shifts in the distribution (Brown et al., 2022; Izzo et al., 2022; Ray et al., 2022), that paved the way for incorporating performative prediction in Reinforcement Learning.

**Performative Reinforcement Learning.** Bell et al. (2021) were the first to propose a setting where the transition and reward of an underlying MDP depend non-deterministically on the deployed policy, thus capturing the essence of performativity to some extent. However, Mandal et al. (2023) can be considered the pioneer in introducing the notion of "*Performative Reinforcement Learning*" and its solution concepts, performatively stable and optimal policy. They propose direct optimization and ascent based techniques which manage to attain performative stability upon repeated retraining. Extensions to this work, Rank et al. (2024) and Mandal & Radanovic (2024) manage to solve the same problem with delayed retraining for linear MDPs. However, there exists no literature that proposes a performative RL algorithm that converges to the performative optimal policy.

Specifically, Mandal et al. (2023) frames the question of using policy gradient to find stable policies as an open problem. The authors further contemplate, as PG functions in the policy space, whether it is possible to converge towards a stable policy. Thus, in this paper, we affirmatively solve an extension (rather a harder problem) of this open problem for tabular MDPs with softmax policies.

**Policy Gradient Algorithms.** Policy gradient algorithms build a central paradigm in reinforcement learning, directly optimizing parametrised policies by estimating the gradient of expected return. The foundational policy gradient theorem (Sutton et al., 1999) established an expression for this gradient in terms of the score and action-value function, while Williams (1992) introduced the REINFORCE algorithm, providing an unbiased likelihood-ratio estimator. Convergence properties of stochastic gradient ascent in policy space were analysed in these early works. Subsequently, Konda & Tsitsiklis (2000) formalized actor–critic methods via two-timescale stochastic approximation, and Kakade (2002) proposed the natural policy gradient, leveraging the Fisher information geometry to accelerate learning. Extensions to trust region methods (Schulman et al., 2015), proximal policy optimization (Schulman et al., 2017), and entropy-regularized objectives (Mnih et al., 2016) have made policy gradient methods widely practical in high-dimensional settings. Recent theoretical advances provide finite-sample convergence guarantees and complexity analyses (Agarwal et al., 2021; Yuan et al., 2022), as well as robustness to distributional shift and adversarial perturbations (Zhang et al., 2020; Xu et al., 2020). Collectively, this body of work establishes policy gradient methods as both practically effective and theoretically grounded method for solving MDP.

## D IMPACT OF POLICY UPDATES ON PeMDPs (SECTION 3.1)

**Lemma 1** (Peformative Performance Difference Lemma). *The difference in performative value functions induced by $\boldsymbol{\pi}$ and $\boldsymbol{\pi}' \in \Pi$ while starting from the initial state distribution $\boldsymbol{\rho}$ is*

$$(1) \quad V_{\boldsymbol{\pi}}^{\boldsymbol{\pi}}(\boldsymbol{\rho}) - V_{\boldsymbol{\pi}'}^{\boldsymbol{\pi}'}(\boldsymbol{\rho}) = \frac{1}{1-\gamma}\mathbb{E}_{(s,a)\sim \boldsymbol{d}_{\boldsymbol{\pi}',\rho}^{\boldsymbol{\pi}}}[A_{\boldsymbol{\pi}'}^{\boldsymbol{\pi}'}(s,a)]$$

$$+ \frac{1}{1-\gamma}\mathbb{E}_{(s,a)\sim \boldsymbol{d}_{\boldsymbol{\pi}',\rho}^{\boldsymbol{\pi}}}[(r_{\boldsymbol{\pi}}(s,a) - r_{\boldsymbol{\pi}'}(s,a)) + \gamma(\mathbf{P}_{\boldsymbol{\pi}}(\cdot|s,a) - \mathbf{P}_{\boldsymbol{\pi}'}(\cdot|s,a))^{\top} V_{\boldsymbol{\pi}}^{\boldsymbol{\pi}}(\cdot)]. \quad (17)$$

*where $A_{\boldsymbol{\pi}'}^{\boldsymbol{\pi}'}(s,a) \triangleq Q_{\boldsymbol{\pi}'}^{\boldsymbol{\pi}'}(s,a) - V_{\boldsymbol{\pi}'}^{\boldsymbol{\pi}'}(s)$ is the performative advantage function for any state $s \in \mathcal{S}$ and action $a \in \mathcal{A}$.*

$$(2) \quad V_{\boldsymbol{\pi}}^{\boldsymbol{\pi}}(\rho) - V_{\boldsymbol{\pi}'}^{\boldsymbol{\pi}'}(\rho) = \frac{1}{1-\gamma}\mathbb{E}_{(s,a)\sim \boldsymbol{d}_{\boldsymbol{\pi}',\rho}^{\boldsymbol{\pi}}}\left[A_{\boldsymbol{\pi}'}^{\boldsymbol{\pi}'}(s,a)\right]$$

$$+ \frac{1}{1-\gamma}\mathbb{E}_{(s,a)\sim \boldsymbol{d}_{\boldsymbol{\pi},\rho}^{\boldsymbol{\pi}}}\left[(r_{\boldsymbol{\pi}}(s,a) - r_{\boldsymbol{\pi}'}(s,a)) + \gamma(\mathbf{P}_{\boldsymbol{\pi}}(\cdot|s,a) - \mathbf{P}_{\boldsymbol{\pi}'}(\cdot|s,a))^{\top} V_{\boldsymbol{\pi}'}^{\boldsymbol{\pi}}(\cdot)\right]. \quad (18)$$

*where $A_{\boldsymbol{\pi}'}^{\boldsymbol{\pi}'}(s,a) \triangleq Q_{\boldsymbol{\pi}'}^{\boldsymbol{\pi}'}(s,a) - V_{\boldsymbol{\pi}'}^{\boldsymbol{\pi}'}(s)$ is the performative advantage function for any state $s \in \mathcal{S}$ and action $a \in \mathcal{A}$.*

$$(3) \quad V_{\boldsymbol{\pi}}^{\boldsymbol{\pi}}(\rho) - V_{\boldsymbol{\pi}'}^{\boldsymbol{\pi}'}(\rho) = \frac{1}{1-\gamma}\mathbb{E}_{(s,a)\sim \boldsymbol{d}_{\boldsymbol{\pi},\rho}^{\boldsymbol{\pi}}}\left[A_{\boldsymbol{\pi}}^{\boldsymbol{\pi}'}(s,a)\right] \quad (19)$$

$$+ \frac{1}{1-\gamma}\mathbb{E}_{(s,a)\sim \boldsymbol{d}_{\boldsymbol{\pi}',\rho}^{\boldsymbol{\pi}'}}\left[(r_{\boldsymbol{\pi}}(s,a) - r_{\boldsymbol{\pi}'}(s,a)) + \gamma(\mathbf{P}_{\boldsymbol{\pi}}(\cdot|s,a) - \mathbf{P}_{\boldsymbol{\pi}'}(\cdot|s,a))^{\top} V_{\boldsymbol{\pi}}^{\boldsymbol{\pi}'}(\cdot)\right]. \quad (20)$$

*where $A_{\boldsymbol{\pi}}^{\boldsymbol{\pi}'}(s,a) \triangleq Q_{\boldsymbol{\pi}}^{\boldsymbol{\pi}'}(s,a) - V_{\boldsymbol{\pi}}^{\boldsymbol{\pi}'}(s)$ is the performative advantage function for any state $s \in \mathcal{S}$ and action $a \in \mathcal{A}$.*

We only use the first version of this lemma in the main draft, and also hereafter, for the proofs.

*Proof of Lemma 1.* We do this proof in two steps. First step involves a decomposition of the difference in value function into two terms : (i) difference in value function after deploying the same policy while agent plays two different policies i.e. the difference that explains stability of the deployed policy, and (ii) difference in value function for deploying two different policies i.e. performance difference for changing the deployed policy. While the second term can be bounded using classic performance difference lemma, in the next and final step, we control the stability inducing term (i).

*Part(1)* – **Step 1: Decomposition.** We start by decomposing the performative performance difference to get a stability and a performance difference terms separately.

$$V_{\boldsymbol{\pi}}^{\boldsymbol{\pi}}(s_0) - V_{\boldsymbol{\pi}'}^{\boldsymbol{\pi}'}(s_0) = \underbrace{V_{\boldsymbol{\pi}}^{\boldsymbol{\pi}}(s_0) - V_{\boldsymbol{\pi}'}^{\boldsymbol{\pi}}(s_0)}_{\text{performative shift term}} + \underbrace{V_{\boldsymbol{\pi}'}^{\boldsymbol{\pi}}(s_0) - V_{\boldsymbol{\pi}'}^{\boldsymbol{\pi}'}(s_0)}_{\text{performance difference term}}$$

$$= V_{\boldsymbol{\pi}}^{\boldsymbol{\pi}}(s_0) - V_{\boldsymbol{\pi}'}^{\boldsymbol{\pi}}(s_0) + \frac{1}{1-\gamma}\mathbb{E}_{(s,a)\sim d_{\boldsymbol{\pi}'}^{\boldsymbol{\pi}}(\cdot|s_0)}[A_{\boldsymbol{\pi}'}^{\boldsymbol{\pi}'}(s,a)] \quad (21)$$

The last equality is a consequence of the classical performance difference lemma (Kakade & Langford, 2002b).

**Step 2: Controlling the performative shift term.** First, let us define $\mathbf{P}_{\boldsymbol{\pi}}^{\boldsymbol{\pi}}(s',s) \triangleq \sum_{a\in\mathcal{A}} \mathbf{P}_{\boldsymbol{\pi}}(s'|s,a)\boldsymbol{\pi}(a|s)$, and $\langle \mathbf{P}_{\boldsymbol{\pi}}^{\boldsymbol{\pi}}(\cdot,s_0), V_{\boldsymbol{\pi}}^{\boldsymbol{\pi}}(\cdot)\rangle \triangleq \sum_{s\in\mathcal{S}} V_{\boldsymbol{\pi}}^{\boldsymbol{\pi}}(s)\mathbf{P}_{\boldsymbol{\pi}}^{\boldsymbol{\pi}}(s,s_0)$.

We first observe that

$$V_{\boldsymbol{\pi}}^{\boldsymbol{\pi}}(s_0) - V_{\boldsymbol{\pi}'}^{\boldsymbol{\pi}}(s_0) = \mathbb{E}_{a\sim\boldsymbol{\pi}(\cdot|s_0)}\left[r_{\boldsymbol{\pi}}(s_0,a) - r_{\boldsymbol{\pi}'}(s_0,a)\right] + \gamma\mathbb{E}_{s\sim\mathbf{P}_{\boldsymbol{\pi}}^{\boldsymbol{\pi}}(\cdot,s_0)}[V_{\boldsymbol{\pi}}^{\boldsymbol{\pi}}(s)] - \gamma\mathbb{E}_{s\sim\mathbf{P}_{\boldsymbol{\pi}'}^{\boldsymbol{\pi}}(\cdot,s_0)}[V_{\boldsymbol{\pi}'}^{\boldsymbol{\pi}}(s)]$$

$$= \mathbb{E}_{a\sim\boldsymbol{\pi}(\cdot|s_0)}\left[r_{\boldsymbol{\pi}}(s_0,a) - r_{\boldsymbol{\pi}'}(s_0,a)\right]$$

$$+ \gamma\sum_s\left(\mathbf{P}_{\boldsymbol{\pi}}^{\boldsymbol{\pi}}(s,s_0) - \mathbf{P}_{\boldsymbol{\pi}'}^{\boldsymbol{\pi}}(s,s_0)\right)V_{\boldsymbol{\pi}}^{\boldsymbol{\pi}}(s) + \gamma\sum_s\mathbf{P}_{\boldsymbol{\pi}'}^{\boldsymbol{\pi}}(s,s_0)\left(V_{\boldsymbol{\pi}}^{\boldsymbol{\pi}}(s) - V_{\boldsymbol{\pi}'}^{\boldsymbol{\pi}}(s)\right)$$

$$= \mathbb{E}_{(s,a)\sim d_{\boldsymbol{\pi}'}^{\boldsymbol{\pi}}(\cdot|s_0)}\left[r_{\boldsymbol{\pi}}(s,a) - r_{\boldsymbol{\pi}'}(s,a) + \gamma(\mathbf{P}_{\boldsymbol{\pi}}(\cdot|s,a) - \mathbf{P}_{\boldsymbol{\pi}'}(\cdot|s,a))^{\top} V_{\boldsymbol{\pi}}^{\boldsymbol{\pi}}(\cdot)\right]$$

The last equality is obtained by recurring the preceding step iteratively.

**Combining** steps 1 and 2 and taking expectation over $s_0 \sim \rho$, we get

$$V_{\boldsymbol{\pi}}^{\boldsymbol{\pi}}(\rho) - V_{\boldsymbol{\pi}'}^{\boldsymbol{\pi}'}(\rho) = \frac{1}{1-\gamma} \mathbb{E}_{(s,a) \sim \boldsymbol{d}_{\boldsymbol{\pi}',\rho}^{\boldsymbol{\pi}}} \Big[ A_{\boldsymbol{\pi}'}^{\boldsymbol{\pi}'}(s,a) + (r_{\boldsymbol{\pi}}(s,a) - r_{\boldsymbol{\pi}'}(s,a)) + \gamma(\mathbf{P}_{\boldsymbol{\pi}}(\cdot|s,a) - \mathbf{P}_{\boldsymbol{\pi}'}(\cdot|s,a))^{\top} V_{\boldsymbol{\pi}}^{\boldsymbol{\pi}}(\cdot) \Big].$$

*Part(2)* – The second equality is obtained by changing the Step 2 as follows:

$$V_{\boldsymbol{\pi}}^{\boldsymbol{\pi}}(s_0) - V_{\boldsymbol{\pi}'}^{\boldsymbol{\pi}}(s_0) = \mathbb{E}_{a \sim \boldsymbol{\pi}(\cdot|s_0)} \Big[ r_{\boldsymbol{\pi}}(s_0,a) - r_{\boldsymbol{\pi}'}(s_0,a) \Big] + \gamma \mathbb{E}_{s \sim \mathbf{P}_{\boldsymbol{\pi}}^{\boldsymbol{\pi}}(\cdot,s_0)}[V_{\boldsymbol{\pi}}^{\boldsymbol{\pi}}(s)] - \gamma \mathbb{E}_{s \sim \mathbf{P}_{\boldsymbol{\pi}'}^{\boldsymbol{\pi}}(\cdot,s_0)}[V_{\boldsymbol{\pi}'}^{\boldsymbol{\pi}}(s)]$$

$$= \mathbb{E}_{a \sim \boldsymbol{\pi}(\cdot|s_0)} \Big[ r_{\boldsymbol{\pi}}(s_0,a) - r_{\boldsymbol{\pi}'}(s_0,a) \Big]$$
$$+ \gamma \sum_s \left( \mathbf{P}_{\boldsymbol{\pi}}^{\boldsymbol{\pi}}(s,s_0) - \mathbf{P}_{\boldsymbol{\pi}'}^{\boldsymbol{\pi}}(s,s_0) \right) V_{\boldsymbol{\pi}}^{\boldsymbol{\pi}}(s) + \gamma \sum_s \mathbf{P}_{\boldsymbol{\pi}}^{\boldsymbol{\pi}}(s,s_0) \left( V_{\boldsymbol{\pi}}^{\boldsymbol{\pi}}(s) - V_{\boldsymbol{\pi}'}^{\boldsymbol{\pi}}(s) \right)$$

$$= \frac{1}{1-\gamma} \mathbb{E}_{(s,a) \sim \boldsymbol{d}_{\boldsymbol{\pi}',\rho}^{\boldsymbol{\pi}}} \Big[ A_{\boldsymbol{\pi}'}^{\boldsymbol{\pi}'}(s,a) \Big]$$
$$+ \frac{1}{1-\gamma} \mathbb{E}_{(s,a) \sim \boldsymbol{d}_{\boldsymbol{\pi},\rho}^{\boldsymbol{\pi}}} \Big[ (r_{\boldsymbol{\pi}}(s,a) - r_{\boldsymbol{\pi}'}(s,a)) + \gamma(\mathbf{P}_{\boldsymbol{\pi}}(\cdot|s,a) - \mathbf{P}_{\boldsymbol{\pi}'}(\cdot|s,a))^{\top} V_{\boldsymbol{\pi}'}^{\boldsymbol{\pi}}(\cdot) \Big].$$

The last equality is obtained by recurring the preceding step iteratively.

*Part(3)* – The third equality is obtained through the following steps.

$$V_{\boldsymbol{\pi}}^{\boldsymbol{\pi}}(\rho) - V_{\boldsymbol{\pi}'}^{\boldsymbol{\pi}'}(\rho) = V_{\boldsymbol{\pi}}^{\boldsymbol{\pi}}(s_0) - V_{\boldsymbol{\pi}'}^{\boldsymbol{\pi}'}(s_0) + V_{\boldsymbol{\pi}}^{\boldsymbol{\pi}'}(s_0) - V_{\boldsymbol{\pi}'}^{\boldsymbol{\pi}'}(s_0)$$

$$= \frac{1}{1-\gamma} \mathbb{E}_{(s,a) \sim d_{\boldsymbol{\pi}}^{\boldsymbol{\pi}}(\cdot|s_0)}[A_{\boldsymbol{\pi}}^{\boldsymbol{\pi}'}(s,a)] + V_{\boldsymbol{\pi}}^{\boldsymbol{\pi}'}(s_0) - V_{\boldsymbol{\pi}'}^{\boldsymbol{\pi}'}(s_0)$$

$$= \frac{1}{1-\gamma} \mathbb{E}_{(s,a) \sim d_{\boldsymbol{\pi}}^{\boldsymbol{\pi}}(\cdot|s_0)}[A_{\boldsymbol{\pi}}^{\boldsymbol{\pi}'}(s,a)] + \mathbb{E}_{a \sim \boldsymbol{\pi}'(\cdot|s_0)} \Big[ r_{\boldsymbol{\pi}}(s_0,a) - r_{\boldsymbol{\pi}'}(s_0,a) \Big]$$

$$+ \gamma \sum_s \left( \mathbf{P}_{\boldsymbol{\pi}}^{\boldsymbol{\pi}'}(s,s_0) - \mathbf{P}_{\boldsymbol{\pi}'}^{\boldsymbol{\pi}'}(s,s_0) \right) V_{\boldsymbol{\pi}}^{\boldsymbol{\pi}'}(s) + \gamma \sum_s \mathbf{P}_{\boldsymbol{\pi}'}^{\boldsymbol{\pi}'}(s,s_0) \left( V_{\boldsymbol{\pi}}^{\boldsymbol{\pi}'}(s) - V_{\boldsymbol{\pi}'}^{\boldsymbol{\pi}'}(s) \right)$$

$$= \frac{1}{1-\gamma} \mathbb{E}_{(s,a) \sim \boldsymbol{d}_{\boldsymbol{\pi},\rho}^{\boldsymbol{\pi}}} \Big[ A_{\boldsymbol{\pi}}^{\boldsymbol{\pi}'}(s,a) \Big]$$
$$+ \frac{1}{1-\gamma} \mathbb{E}_{(s,a) \sim \boldsymbol{d}_{\boldsymbol{\pi}',\rho}^{\boldsymbol{\pi}}} \Big[ (r_{\boldsymbol{\pi}}(s,a) - r_{\boldsymbol{\pi}'}(s,a)) + \gamma(\mathbf{P}_{\boldsymbol{\pi}}(\cdot|s,a) - \mathbf{P}_{\boldsymbol{\pi}'}(\cdot|s,a))^{\top} V_{\boldsymbol{\pi}}^{\boldsymbol{\pi}'}(\cdot) \Big].$$

$\square$

**Lemma 2** (Bounding Performative Performance Difference for Gradually Shifting Environments). *Let us assume that both rewards and transitions are Lipschitz functions of policy, i.e.* $\|r_{\boldsymbol{\pi}} - r_{\boldsymbol{\pi}'}\| \le L_r \|\boldsymbol{\pi} - \boldsymbol{\pi}'\|$ *and* $\|\mathbf{P}_{\boldsymbol{\pi}} - \mathbf{P}_{\boldsymbol{\pi}'}\| \le L_{\mathbf{P}} \|\boldsymbol{\pi} - \boldsymbol{\pi}'\|$, *for some* $L_r, L_{\mathbf{P}} \ge 0$. *Then, under Assumption 1, the performative shift in the sub-optimality gap of a policy* $\boldsymbol{\pi_\theta}$ *satisfies*

$$\left| V_{\boldsymbol{\pi}_o^\star}^{\boldsymbol{\pi}_o^\star}(\boldsymbol{\rho}) - V_{\boldsymbol{\pi_\theta}}^{\boldsymbol{\pi_\theta}}(\boldsymbol{\rho}) - \frac{1}{1-\gamma} \mathbb{E}_{(s,a) \sim \boldsymbol{d}_{\boldsymbol{\pi_\theta},\rho}^{\boldsymbol{\pi}_o^\star}} [A_{\boldsymbol{\pi_\theta}}^{\boldsymbol{\pi_\theta}}(s,a)] \right| \le \frac{2\sqrt{2}}{1-\gamma}(L_r + \frac{\gamma}{1-\gamma} L_{\mathbf{P}} R_{\max}) \mathbb{E}_{s_0 \sim \boldsymbol{\rho}} D_{\mathrm{H}}\left(\boldsymbol{\pi}_o^\star(\cdot|s_0) \| \boldsymbol{\pi_\theta}(\cdot|s_0)\right).$$

$$(22)$$

*where* $D_{\mathrm{H}}(\mathbf{x} \| \mathbf{y})$ *denotes the Hellinger distance between* $\mathbf{x}$ *and* $\mathbf{y}$.

*Proof of Lemma 2.* We do this proof in three steps. We start from the final expression in Lemma 1, then in step 2 we impose bounds on reward and transition differences leveraging the Lipschitz assumption. Lastly, we bound the policy difference in first order norm using relation between Total Variation (TV) and Hellinger distance.

**Step 1:** From Lemma 1, we get

$$V_{\boldsymbol{\pi}_o^\star}^{\boldsymbol{\pi}_o^\star}(s_0) - V_{\boldsymbol{\pi_\theta}}^{\boldsymbol{\pi_\theta}}(s_0) = \frac{1}{1-\gamma} \mathbb{E}_{(s,a) \sim \boldsymbol{d}_{\boldsymbol{\pi_\theta},\rho}^{\boldsymbol{\pi}_o^\star}} \Big[ A_{\boldsymbol{\pi_\theta}}^{\boldsymbol{\pi_\theta}}(s,a) + (r_{\boldsymbol{\pi}_o^\star}(s,a) - r_{\boldsymbol{\pi_\theta}}(s,a)) + \gamma(\mathbf{P}_{\boldsymbol{\pi}_o^\star}(\cdot|s,a) - \mathbf{P}_{\boldsymbol{\pi_\theta}}(\cdot|s,a))^{\top} V_{\boldsymbol{\pi}_o^\star}^{\boldsymbol{\pi}_o^\star}(\cdot) \Big] \bigg|.$$

Thus,

$$\left| V_{\boldsymbol{\pi}_o^\star}^{\boldsymbol{\pi}_o^\star}(\boldsymbol{\rho}) - V_{\boldsymbol{\pi_\theta}}^{\boldsymbol{\pi_\theta}}(\boldsymbol{\rho}) - \frac{1}{1-\gamma} \mathbb{E}_{(s,a) \sim d_{\boldsymbol{\pi_\theta},\rho}^{\boldsymbol{\pi}_o^\star}} [A_{\boldsymbol{\pi_\theta}}^{\boldsymbol{\pi_\theta}}(s,a)] \right|$$

$$= \frac{1}{1-\gamma} \left| \mathbb{E}_{(s,a)\sim d_{\pi_\theta,\rho}^{\pi_o^\star}} (r_{\pi_o^\star}(s,a) - r_{\pi_\theta}(s,a)) + \gamma(\mathbf{P}_{\pi_o^\star}(\cdot|s,a) - \mathbf{P}_{\pi_\theta}(\cdot|s,a))^\top V_{\pi_o^\star}^{\pi_o^\star}(\cdot) \right| \tag{23}$$

**Step 2:** Using Jensen's inequality together with the fact that $d_{\pi_\theta,\rho}^{\pi_o^\star}(s,a|s_0) \leq 1$, for rewards, we get

$$\left| \mathbb{E}_{(s,a)\sim d_{\pi_\theta,\rho}^{\pi_o^\star}(\cdot,\cdot|s_0)} \left[ r_{\pi_o^\star}(s,a) - r_{\pi_\theta}(s,a) \right] \right| \leq \mathbb{E}_{(s,a)\sim d_{\pi_\theta,\rho}^{\pi_o^\star}(\cdot,\cdot|s_0)} \left| r_{\pi_o^\star}(s,a) - r_{\pi_\theta}(s,a) \right| \leq \|r_{\pi_o^\star} - r_{\pi_\theta}\|_1$$

Similarly for transitions, we get

$$\left| \mathbb{E}_{(s,a)\sim d_{\pi_\theta,\rho}^{\pi_o^\star}(\cdot,\cdot|s_0)} \left[ (\mathbf{P}_{\pi_o^\star} - \mathbf{P}_{\pi_\theta})^\top V_\pi^\pi \right] \right| \leq \mathbb{E}_{(s,a)\sim d_{\pi_\theta,\rho}^{\pi_o^\star}(\cdot,\cdot|s_0)} \left| (\mathbf{P}_{\pi_o^\star} - \mathbf{P}_{\pi_\theta})^\top V_\pi^\pi \right|$$

$$\overset{(a)}{\leq} \mathbb{E}_{(s,a)\sim d_{\pi_\theta,\rho}^{\pi_o^\star}(\cdot,\cdot|s_0)} \left[ \|\mathbf{P}_{\pi_o^\star} - \mathbf{P}_{\pi_\theta}\|_1 \cdot \|V_{\pi_o^\star}^{\pi_o^\star}\|_\infty \right]$$

$$= \|\mathbf{P}_{\pi_o^\star} - \mathbf{P}_{\pi_\theta}\|_1 \cdot \|V_{\pi_o^\star}^{\pi_o^\star}\|_\infty ,$$

(a) holds due to Hölder's inequality.

Now, leveraging the triangle inequality and Lipschitzness assumption on reward and transitions, we further get

$$\left| \mathbb{E}_{(s,a)\sim d_{\pi_\theta,\rho}^{\pi_o^\star}(\cdot,\cdot|s_0)} \left[ r_{\pi_o^\star}(s,a) - r_{\pi_\theta}(s,a) + \gamma(\mathbf{P}_{\pi_o^\star} - \mathbf{P}_{\pi_\theta})^\top V_\pi^\pi \right] \right| \leq L_r \|\pi_o^\star - \pi_\theta\|_1 + \gamma L_\mathbf{P} \left\| V_{\pi_o^\star}^{\pi_o^\star} \right\|_\infty \|\pi_o^\star - \pi_\theta\|_1$$

Finally, due to Assumption 1, we get $\left\| \mathbf{V}_{\pi_o^\star}^{\pi_o^\star} \right\|_\infty \leq \frac{R_{\max}}{1-\gamma}$, and thus,

$$\left| \mathbb{E}_{(s,a)\sim d_{\pi_\theta,\rho}^{\pi_o^\star}(\cdot,\cdot|s_0)} \left[ r_{\pi_o^\star}(s,a) - r_{\pi_\theta}(s,a) + \gamma(\mathbf{P}_{\pi_o^\star} - \mathbf{P}_{\pi_\theta})^\top V_{\pi_o^\star}^{\pi_o^\star} \right] \right| \leq L_r \|\pi_o^\star - \pi_\theta\|_1 + \frac{\gamma}{1-\gamma} L_\mathbf{P} R_{\max} \|\pi_o^\star - \pi_\theta\|_1$$

**Step 3:** We know $\|\pi_o^\star - \pi_\theta\|_1 = 2\mathrm{TV}(\pi_o^\star \| \pi_\theta) \leq 2\sqrt{2} D_\mathrm{H}(\pi_o^\star \| \pi_\theta)$. Thus,

$$\left| \mathbb{E}_{(s,a)\sim d_{\pi_\theta,\rho}^{\pi_o^\star}(\cdot,\cdot|s_0)} \left[ r_{\pi_o^\star}(s,a) - r_{\pi_\theta}(s,a) + \gamma(\mathbf{P}_{\pi_o^\star} - \mathbf{P}_{\pi_\theta})^\top V_{\pi_o^\star}^{\pi_o^\star} \right] \right|$$

$$\leq 2\sqrt{2} \left( L_r + \frac{\gamma}{1-\gamma} L_\mathbf{P} R_{\max} \right) D_\mathrm{H}(\pi_o^\star(\cdot \mid s_0) \| \pi_\theta(\cdot \mid s_0)) \tag{24}$$

We conclude this proof by putting the upper bound in Equation (24) in Equation (23) and taking expectation over $s_0 \sim \rho$ to get the desired expression.

$\square$

## E  SMOOTHNESS OF PERFORMATIVE VALUE FUNCTION AND ENTROPY REGULARISER

**Lemma 4** (Performative Smoothness Lemma). *Let $\pi_\alpha \triangleq \pi_{\theta+\alpha u}$, and let $V_\alpha^\alpha(s_0)$ be the corresponding value at a fixed state $s_0$, i.e., $V_\alpha^\alpha(s_0) \triangleq V_{\pi_\alpha}^{\pi_\alpha}(s_0)$. If the following conditions hold true,*

$$\sum_{a\in\mathcal{A}} \left| \frac{\mathrm{d}\pi_\alpha(a \mid s_0)}{\mathrm{d}\alpha} \right|_{\alpha=0} \right| \leq C_1, \quad \sum_{a\in\mathcal{A}} \left| \frac{\mathrm{d}^2\pi_\alpha(a \mid s_0)}{\mathrm{d}\alpha^2} \right|_{\alpha=0} \right| \leq C_2, \sum_{s\in\mathcal{S}} \left| \frac{\mathrm{d}\mathbf{P}_\alpha(s \mid s_0, a_0)}{\mathrm{d}\alpha} \right|_{\alpha=0} \right| \leq T_1,$$

$$\sum_{s\in\mathcal{S}} \left| \frac{\mathrm{d}^2\mathbf{P}_\alpha(s \mid s_0, a_0)}{\mathrm{d}\alpha^2} \right|_{\alpha=0} \right| \leq T_2, \sum_{a\in\mathcal{A}} \left| \frac{\mathrm{d}r_\alpha(s_0, a)}{\mathrm{d}\alpha} \right|_{\alpha=0} \right| \leq R_1, \quad \sum_{a\in\mathcal{A}} \left| \frac{\mathrm{d}^2r_\alpha(s_0, a)}{\mathrm{d}\alpha^2} \right|_{\alpha=0} \right| \leq R_2,$$

*we get*

$$\max_{\|u\|_2=1} \left\| \frac{\mathrm{d}^2V_\alpha^\alpha(s_0)}{\mathrm{d}\alpha^2} \right|_{\alpha=0} \right\| \leq \frac{C_2}{1-\gamma} + 2C_1\beta_1 + C_2\beta_2 \triangleq L,$$

*where $\beta_1 = \frac{\gamma}{(1-\gamma)^2}(C_1 + T_1) + \frac{R_1}{1-\gamma}$ and $\beta_2 = \frac{2\gamma^2}{(1-\gamma)^3}(C_1 + T_1)^2 + \frac{\gamma}{(1-\gamma)^2}(C_2 + 2C_1T_1 + T_2) + \frac{2\gamma R_1}{(1-\gamma)^2}(C_2 + 2C_1T_1 + T_2) + \frac{R_2}{1-\gamma} + \frac{\gamma C_1 R_1}{(1-\gamma)^2}$.*

*Proof.* **Step 1:** To prove the second order smoothness of the value function we start by taking its second derivative. Consider the expected return under policy $\pi_\alpha$:

$$V_\alpha^\alpha(s_0) = \sum_a \pi_\alpha(a \mid s_0)Q_\alpha^\alpha(s_0, a)$$

Differentiating twice with respect to $\alpha$, we obtain:

$$\frac{\mathrm{d}^2V_\alpha^\alpha(s_0)}{\mathrm{d}\alpha^2} = \sum_a \frac{\mathrm{d}^2\pi_\alpha(a \mid s_0)}{\mathrm{d}\alpha^2}Q_\alpha^\alpha(s_0, a) + 2\sum_a \frac{\mathrm{d}\pi_\alpha(a \mid s_0)}{\mathrm{d}\alpha} \frac{\mathrm{d}Q_\alpha^\alpha(s_0, a)}{\mathrm{d}\alpha} + \sum_a \pi_\alpha(a \mid s_0)\frac{\mathrm{d}^2Q_\alpha^\alpha(s_0, a)}{\mathrm{d}\alpha^2}$$

$Q_\alpha^\alpha(s_0, a_0)$ is the Q-function corresponding to the policy $\pi_\alpha$ at state $s_0$ and action $a_0$. Observe that $Q_\alpha^\alpha(s_0, a_0)$ can further be written as:

$$Q_\alpha^\alpha(s_0, a_0) = e_{(s_0,a_0)}^\top(I - \gamma\tilde{\mathbf{P}}(\alpha))^{-1}r_\alpha = e_{(s_0,a_0)}^\top M(\alpha)r_\alpha$$

where $M(\alpha) \triangleq (I - \gamma\mathbf{P}(\alpha))^{-1}$ and $\tilde{\mathbf{P}}(\alpha)$ is the state-action transition matrix under policy $\pi_\alpha$, defined as:

$$[\tilde{\mathbf{P}}(\alpha)](s', a' \mid s, a) \triangleq \pi_\alpha(a' \mid s')\mathbf{P}_\alpha(s' \mid s, a)$$

Differentiating $Q_\alpha^\alpha(s, a)$ with respect to $\alpha$ gives:

$$\frac{\mathrm{d}Q_\alpha^\alpha(s_0, a_0)}{\mathrm{d}\alpha} = \gamma e_{(s_0,a_0)}^\top M(\alpha)\frac{\mathrm{d}\tilde{\mathbf{P}}(\alpha)}{\mathrm{d}\alpha}M(\alpha)r_\alpha + e_{(s_0,a_0)}^\top M(\alpha)\frac{\mathrm{d}r_\alpha}{\mathrm{d}\alpha}$$

And correspondingly,

$$\frac{\mathrm{d}^2Q_\alpha^\alpha(s_0, a_0)}{\mathrm{d}\alpha^2} = 2\gamma^2 e_{(s_0,a_0)}^\top M(\alpha)\frac{\mathrm{d}\tilde{\mathbf{P}}(\alpha)}{\mathrm{d}\alpha}M(\alpha)\frac{\mathrm{d}\tilde{\mathbf{P}}(\alpha)}{\mathrm{d}\alpha}M(\alpha)r_\alpha + \gamma e_{(s_0,a_0)}^\top M(\alpha)\frac{\mathrm{d}^2\tilde{\mathbf{P}}(\alpha)}{\mathrm{d}\alpha^2}M(\alpha)r_\alpha$$

$$+ \gamma e_{(s_0,a_0)}^\top M(\alpha)\frac{\mathrm{d}\tilde{\mathbf{P}}(\alpha)}{\mathrm{d}\alpha}M(\alpha)\frac{\mathrm{d}r_\alpha}{\mathrm{d}\alpha} + e_{(s_0,a_0)}^\top M(\alpha)\frac{\mathrm{d}^2r_\alpha}{\mathrm{d}\alpha^2}$$

$$+ \gamma e_{(s_0,a_0)}^\top M(\alpha)\frac{\mathrm{d}\tilde{\mathbf{P}}(\alpha)}{\mathrm{d}\alpha}M(\alpha)\frac{\mathrm{d}r_\alpha}{\mathrm{d}\alpha} \tag{25}$$

**Step 2:** Now we need to find the derivative of $\tilde{\mathbf{P}}(\alpha)$ w.r.t $\alpha$ in order to substitute in (25). Hence, we can differentiate $\tilde{\mathbf{P}}(\alpha)$ with respect to $\alpha$ to obtain:

$$\frac{\mathrm{d}\tilde{\mathbf{P}}(\alpha)}{\mathrm{d}\alpha} \bigg|_{\alpha=0} (s', a' \mid s, a) = \frac{\mathrm{d}\pi_\alpha(a' \mid s')}{\mathrm{d}\alpha} \bigg|_{\alpha=0} \mathbf{P}_\alpha(s' \mid s, a) + \frac{\mathrm{d}\mathbf{P}_\alpha(s' \mid s, a)}{\mathrm{d}\alpha} \bigg|_{\alpha=0} \pi_\alpha(a' \mid s')$$

Now, for an arbitrary vector $\mathbf{x}$, we have:

$$\left[\left.\frac{\mathrm{d}\tilde{\mathbf{P}}(\alpha)}{\mathrm{d}\alpha}\right|_{\alpha=0}\mathbf{x}\right]_{(s,a)} = \sum_{s',a'} \left.\frac{\mathrm{d}\boldsymbol{\pi}_\alpha(a'\mid s')}{\mathrm{d}\alpha}\right|_{\alpha=0}\mathbf{P}_\alpha(s'\mid s,a)\mathbf{x}_{s',a'} + \sum_{s',a'} \left.\frac{\mathrm{d}\mathbf{P}_\alpha(s'\mid s,a)}{\mathrm{d}\alpha}\right|_{\alpha=0}\boldsymbol{\pi}_\alpha(a'\mid s')\mathbf{x}_{s',a'}$$

Taking the maximum over unit vectors $\mathbf{u}$ in $\ell_2$-norm:

$$\max_{\|\mathbf{u}\|_2=1}\left\|\left.\frac{\mathrm{d}\tilde{\mathbf{P}}(\alpha)}{\mathrm{d}\alpha}\right|_{\alpha=0}\mathbf{x}\right\|_\infty \leq \max_{\|\mathbf{u}\|_2=1}\left|\sum_{s',a'}\left.\frac{\mathrm{d}\boldsymbol{\pi}_\alpha(a'\mid s')}{\mathrm{d}\alpha}\right|_{\alpha=0}\mathbf{P}_\alpha(s'\mid s,a)\mathbf{x}_{s',a'}\right|$$

$$+ \max_{\|\mathbf{u}\|_2=1}\left|\sum_{s',a'}\left.\frac{\mathrm{d}\mathbf{P}_\alpha(s'\mid s,a)}{\mathrm{d}\alpha}\right|_{\alpha=0}\boldsymbol{\pi}_\alpha(a'\mid s')\mathbf{x}_{s',a'}\right|$$

$$\leq \max_{s,a}\sum_{s'}\mathbf{P}_\alpha(s'\mid s,a)\sum_{a'}\left|\left.\frac{\mathrm{d}\boldsymbol{\pi}_\alpha(a'\mid s')}{\mathrm{d}\alpha}\right|_{\alpha=0}\right|\cdot\|\mathbf{x}\|_\infty$$

$$+ \max_{s,a}\sum_{a'}\boldsymbol{\pi}_\alpha(a'\mid s')\sum_{s'}\left|\left.\frac{\mathrm{d}\mathbf{P}_\alpha(s'\mid s,a)}{\mathrm{d}\alpha}\right|_{\alpha=0}\right|\cdot\|\mathbf{x}\|_\infty$$

$$\leq \max_{s,a}\sum_{s'}\mathbf{P}_\alpha(s'\mid s,a)\|\mathbf{x}\|_\infty C_1 + \max_{s,a}\sum_{a'}\boldsymbol{\pi}(a'\mid s')\|\mathbf{x}\|_\infty T_1$$

$$\leq C_1\|\mathbf{x}\|_\infty + T_1\|\mathbf{x}\|_\infty = (C_1+T_1)\|\mathbf{x}\|_\infty$$

By the definition of the $\ell_\infty$-norm, we conclude:

$$\max_{\|\mathbf{u}\|_2=1}\left\|\left.\frac{\mathrm{d}\mathbf{P}_\alpha}{\mathrm{d}\alpha}\right|_{\alpha=0}\mathbf{x}\right\|_\infty \leq (C_1+T_1)\|\mathbf{x}\|_\infty \tag{26}$$

Similarly, differentiating $\tilde{\mathbf{P}}(\alpha)$ twice w.r.t. $\alpha$, we get

$$\left[\left.\frac{\mathrm{d}^2\tilde{\mathbf{P}}(\alpha)}{\mathrm{d}\alpha^2}\right|_{\alpha=0}\right]_{(s,a)\to(s',a')} = \left.\frac{\mathrm{d}^2\boldsymbol{\pi}_\alpha(a'\mid s')}{(\mathrm{d}\alpha)^2}\right|_{\alpha=0}\mathbf{P}_\alpha(s'\mid s,a) + \left.\frac{\mathrm{d}^2\mathbf{P}_\alpha(s'\mid s,a)}{\mathrm{d}\alpha^2}\right|_{\alpha=0}\boldsymbol{\pi}_\alpha(a'\mid s')$$

$$+ 2\left.\frac{\mathrm{d}\boldsymbol{\pi}_\alpha(a'\mid s')}{\mathrm{d}\alpha}\right|_{\alpha=0}\left.\frac{\mathrm{d}\mathbf{P}_\alpha(s'\mid s,a)}{\mathrm{d}\alpha}\right|_{\alpha=0}$$

Hence, we can consider the following norm bound:

$$\max_{\|\mathbf{u}\|_2=1}\left\|\left.\frac{\mathrm{d}^2\tilde{\mathbf{P}}(\alpha)}{\mathrm{d}\alpha^2}\right|_{\alpha=0}\mathbf{x}\right\|_\infty \leq C_2\|\mathbf{x}\|_\infty + 2C_1T_1\|\mathbf{x}\|_\infty + T_2\|\mathbf{x}\|_\infty = (C_2+2C_1T_1+T_2)\|\mathbf{x}\|_\infty \tag{27}$$

**Step 3:** Now we need to put the pieces back together in order to calculate the second derivative of $V_\alpha^\alpha$ w.r.t $\alpha$. Let us recall $M(\alpha)$. Using the power series expansion of the matrix inverse, we can write $M(\alpha)$ as:

$$M(\alpha) = (I - \gamma\tilde{\mathbf{P}}(\alpha))^{-1} = \sum_{n=0}^\infty \gamma^n\tilde{\mathbf{P}}(\alpha)^n$$

which implies that $M(\alpha) \geq 0$ (component-wise), and

$$M(\alpha)\mathbf{1} = \frac{1}{1-\gamma}\mathbf{1},$$

i.e., each row of $M(\alpha)$ is positive and sums to $\frac{1}{1-\gamma}$.

This implies:

$$\max_{\|u\|_2=1}\|M(\alpha)\mathbf{x}\|_\infty \leq \frac{1}{1-\gamma}\|\mathbf{x}\|_\infty.$$

This gives, using the expressions for $\frac{\mathrm{d}^2 Q_\alpha^\alpha(s_0,a_0)}{\mathrm{d}\alpha^2}$ and $\frac{\mathrm{d}Q_\alpha^\alpha(s_0,a_0)}{\mathrm{d}\alpha}$, an upper bound on their magnitudes based on $\|\mathbf{x}\|_\infty$ and constants arising from bounds on the derivatives of $\tilde{\mathbf{P}}(\alpha)$ and $r_\alpha$.

$$\max_{\|\mathbf{u}\|_2=1} \left\| \frac{\mathrm{d}^2 Q_\alpha^\alpha(s_0,a_0)}{\mathrm{d}\alpha^2} \right\|_\infty$$

$$\leq 2\gamma^2 \left\| M(\alpha)\frac{\mathrm{d}\tilde{\mathbf{P}}(\alpha)}{\mathrm{d}\alpha} M(\alpha)\frac{\mathrm{d}\tilde{\mathbf{P}}(\alpha)}{\mathrm{d}\alpha} M(\alpha)r_\alpha \right\|_\infty + \gamma \left\| M(\alpha)\frac{\mathrm{d}^2\tilde{\mathbf{P}}(\alpha)}{\mathrm{d}\alpha^2} M(\alpha)r_\alpha \right\|_\infty$$

$$+ \gamma \left\| M(\alpha)\frac{\mathrm{d}^2\tilde{\mathbf{P}}(\alpha)}{\mathrm{d}\alpha^2} M(\alpha)\frac{\mathrm{d}r_\alpha}{\mathrm{d}\alpha} \right\|_\infty + \left\| M(\alpha)\frac{\mathrm{d}^2 r_\alpha}{\mathrm{d}\alpha^2} \right\|_\infty + 2\gamma \left\| M(\alpha)\frac{\mathrm{d}\tilde{\mathbf{P}}(\alpha)}{\mathrm{d}\alpha} M(\alpha)\frac{\mathrm{d}r_\alpha}{\mathrm{d}\alpha} \right\|_\infty$$

Bounding using known bounds on transitions and rewards:

$$\max_{\|\mathbf{u}\|_2=1} \left\| \frac{\mathrm{d}^2 Q_\alpha^\alpha(s_0,a_0)}{\mathrm{d}\alpha^2} \right\|_\infty \leq \frac{2\gamma^2}{(1-\gamma)^3}(C_1+T_1)^2 + \frac{\gamma}{(1-\gamma)^2}(C_2+2C_1T_1+T_2)$$

$$+ \frac{2\gamma R_1}{(1-\gamma)^2}(C_2+2C_1T_1+T_2) + \frac{R_2}{1-\gamma} + \frac{\gamma C_1 R_1}{(1-\gamma)^2} = \beta_2$$

Corresponding bound on the first derivative is:

$$\max_{\|\mathbf{u}\|_2=1} \left\| \frac{\mathrm{d}Q_\alpha^\alpha(s_0,a_0)}{\mathrm{d}\alpha} \right\|_\infty \leq \gamma \left\| M(\alpha)\frac{\mathrm{d}\tilde{\mathbf{P}}(\alpha)}{\mathrm{d}\alpha} M(\alpha)\frac{\mathrm{d}r_\alpha}{\mathrm{d}\alpha} \right\|_\infty + \left\| M(\alpha)\frac{\mathrm{d}r_\alpha}{\mathrm{d}\alpha} \right\|_\infty$$

$$\leq \frac{\gamma}{(1-\gamma)^2}(C_1+T_1) + \frac{R_1}{1-\gamma} = \beta_1$$

**Step 4:** Finally, putting all the bounds together to evaluate the upper bound of the desired quantity, we get,

$$\max_{\|\mathbf{u}\|_2=1} \left\| \frac{\mathrm{d}^2 V_\alpha^\alpha(s_0)}{\mathrm{d}\alpha^2} \right\|_\infty \leq \frac{C_2}{1-\gamma} + 2C_1\beta_1 + \beta_2 \tag{28}$$

$\square$

**Corollary 1.** *For softmax PeMDPs, we characterise*

$$C_1 = 2, \quad C_2 = 6, \quad T_1 = L_\mathbf{P} = \max_s |\psi(s)| \triangleq \psi_{\max}, \quad T_2 = \max_s |\psi(s)|^2, \quad R_1 = L_r|\mathcal{A}| = \xi|\mathcal{A}|, \quad R_2 = 0$$

*Thus,*

$$\max_{\|u\|_2=1} \left\| \frac{\mathrm{d}^2 V_\alpha^\alpha(s_0)}{\mathrm{d}\alpha^2} \Big|_{\alpha=0} \right\| \leq \mathcal{O}\left( \max\left\{ \frac{\gamma R_{\max}|\mathcal{A}|}{(1-\gamma)^2}, \frac{\gamma^2}{(1-\gamma)^3} \right\} \right) \triangleq \mathcal{O}(L). \tag{29}$$

*Proof.* We use the expressions already found in (35) to state the following:

$$\sum_{a\in\mathcal{A}} \left| \frac{\mathrm{d}}{\mathrm{d}\alpha} \boldsymbol{\pi}_{\boldsymbol{\theta}+\alpha\mathbf{u}}(a\mid s) \Big|_{\alpha=0} \right| \leq \sum_{a\in\mathcal{A}} \boldsymbol{\pi}_{\boldsymbol{\theta}}(a\mid s) \left| \mathbf{u}_s^\top (\mathbf{e}_a - \boldsymbol{\pi}(\cdot\mid s)) \right| \leq \max_{a\in\mathcal{A}} \left( \mathbf{u}_s^\top \mathbf{e}_a + \mathbf{u}_s^\top \boldsymbol{\pi}(\cdot\mid s) \right) \leq 2.$$

Similarly, differentiating once again w.r.t. $\alpha$, we get

$$\sum_{a\in\mathcal{A}} \left| \frac{\mathrm{d}^2}{\mathrm{d}\alpha^2} \boldsymbol{\pi}_{\boldsymbol{\theta}+\alpha\mathbf{u}}(a\mid s) \Big|_{\alpha=0} \right| \leq \max_{a\in\mathcal{A}} \left( \mathbf{u}_s^\top \mathbf{e}_a \mathbf{e}_a^\top \mathbf{u}_s + \mathbf{u}_s^\top \mathbf{e}_a \boldsymbol{\pi}(\cdot\mid s)^\top \mathbf{u}_s + \mathbf{u}_s^\top \boldsymbol{\pi}(\cdot\mid s)\mathbf{e}_a^\top \mathbf{u}_s \right)$$

$$+ 2\, \mathbf{u}_s^\top \boldsymbol{\pi}(\cdot \mid s) \boldsymbol{\pi}(\cdot \mid s)^\top \mathbf{u}_s + \mathbf{u}_s^\top \mathrm{diag}(\boldsymbol{\pi}(\cdot \mid s)) \mathbf{u}_s \Big) \le 6.$$

And hence for transition we get,

$$\sum_{s' \in \mathcal{S}} \left| \frac{\mathrm{d}}{\mathrm{d}\alpha} \mathbf{P}_{\boldsymbol{\pi}_{\boldsymbol{\theta} + \alpha \mathbf{u}}}(a \mid s) \Big|_{\alpha=0} \right| \le \sum_{s' \in \mathcal{S}} |\psi(s')| \mathbf{P}_{\boldsymbol{\pi}_{\boldsymbol{\theta}}}(s' \mid s, a) \left| \mathbf{u}_{s,a}(1 - \mathbf{P}_{\boldsymbol{\pi}_{\boldsymbol{\theta}}}(\cdot \mid s, a)) \right| \le |\mathbf{u}_{s,a}| \max_s |\psi(s)| \le \max_s |\psi(s)|$$

And similarly, it can be shown that:

$$\sum_{a \in \mathcal{A}} \left| \frac{\mathrm{d}^2}{\mathrm{d}\alpha^2} \mathbf{P}_{\boldsymbol{\pi}_{\boldsymbol{\theta} + \alpha \mathbf{u}}}(a \mid s) \Big|_{\alpha=0} \right| \le |\mathbf{u}_{s,a}|^2 \max_s |\psi(s)|^2 \le \max_s |\psi(s)|^2$$

Similarly for rewards we get:

$$\sum_{a \in \mathcal{A}} \left| \frac{\mathrm{d}}{\mathrm{d}\alpha} r_{\boldsymbol{\pi}_{\boldsymbol{\theta} + \alpha \mathbf{u}}}(a \mid s) \Big|_{\alpha=0} \right| \le \xi |\mathcal{A}| \quad , \quad \sum_{a \in \mathcal{A}} \left| \frac{\mathrm{d}^2}{\mathrm{d}\alpha^2} r_{\boldsymbol{\pi}_{\boldsymbol{\theta} + \alpha \mathbf{u}}}(a \mid s) \Big|_{\alpha=0} \right| = 0$$

Hence, we can use the following choice of constants for softmax parametrization,

$$C_1 = 2 \quad , \quad C_2 = 6$$
$$T_1 = L_{\mathbf{P}} = \max_s |\psi(s)| \quad , \quad T_2 = \max_s |\psi(s)|^2$$
$$R_1 = L_r |\mathcal{A}| = \xi |\mathcal{A}| \quad , \quad R_2 = 0$$

to get the desired order of $\max_{\|u\|_2=1} \left\| \frac{\mathrm{d}^2 V_\alpha^\alpha(s_0)}{\mathrm{d}\alpha^2} \Big|_{\alpha=0} \right\|$.

$\square$

**Lemma 5** (Smoothness of Entropy Regularizer). *Define the discounted entropy regularizer as:*

$$\mathcal{H}_{\boldsymbol{\pi}_{\boldsymbol{\theta}_\alpha}}^{\boldsymbol{\pi}_{\boldsymbol{\theta}_\alpha}}(s) = \mathbb{E}_{\tau \sim \mathbf{P}_{\boldsymbol{\pi}}^{\boldsymbol{\pi}}} \left[ \sum_{t=0}^{\infty} -\gamma^t \log \boldsymbol{\pi}_{\boldsymbol{\theta}_\alpha}(a_t \mid s_t) \right]$$

*Under the same assumptions as 4, the following holds:*

$$\max_{\|u\|_2=1} \left\| \frac{\partial^2 \mathcal{H}_{\boldsymbol{\pi}_{\boldsymbol{\theta}_\alpha}}^{\boldsymbol{\pi}_{\boldsymbol{\theta}_\alpha}}(s)}{\partial \alpha^2} \Big|_{\alpha=0} \right\|_\infty \le \beta_\lambda$$

*where*

$$\beta_\lambda = 2\gamma^2 \frac{3(1 + \log |\mathcal{A}|)}{1 - \gamma} + \gamma \frac{2 \log |\mathcal{A}|}{(1 - \gamma)^2} (C_1 + T_1) + 2\gamma \frac{\log |\mathcal{A}|}{(1 - \gamma)^2} (C_2 + 2C_1 T_1 + T_2) + \frac{\log |\mathcal{A}|}{(1 - \gamma)^3} (C_1 + T_1)^2.$$

*Proof.* **Step 1:** Define the state-wise entropy term:

$$h_{\boldsymbol{\theta}_\alpha}(s) = -\sum_a \boldsymbol{\pi}_{\boldsymbol{\theta}_\alpha}(a \mid s) \log \boldsymbol{\pi}_{\boldsymbol{\theta}_\alpha}(a \mid s).$$

From Mei et al. (2020) (Lemma 7) we report that,

$$\left\| \frac{\partial h_{\boldsymbol{\theta}_\alpha}}{\partial \alpha} \right\|_\infty \le 2 \cdot \log |\mathcal{A}| \cdot \|u\|_2, \qquad \left\| \frac{\partial^2 h_{\boldsymbol{\theta}_\alpha}}{\partial \alpha^2} \right\|_\infty \le 3 \cdot (1 + \log |\mathcal{A}|) \cdot \|\mathbf{u}\|_2^2. \tag{30}$$

Additionally, Mei et al. (2020) also presents a second result expressing the second derivative of the entropy w.r.t $\alpha$,

$$\frac{\partial^2 \mathcal{H}_{\boldsymbol{\pi}_{\boldsymbol{\theta}_\alpha}}^{\boldsymbol{\pi}_{\boldsymbol{\theta}_\alpha}}(s)}{\partial \alpha^2} = 2\gamma^2 \, \mathbf{e}_s^\top M(\alpha) \frac{\partial \mathbf{P}(\alpha)}{\partial \alpha} M(\alpha) \frac{\partial \mathbf{P}(\alpha)}{\partial \alpha} M(\alpha) h_{\boldsymbol{\theta}_\alpha}$$

$$+ \gamma \, \mathbf{e}_s^\top M(\alpha) \frac{\partial^2 \mathbf{P}(\alpha)}{\partial \alpha^2} M(\alpha) h_{\boldsymbol{\theta}_\alpha} + 2\gamma \, \mathbf{e}_s^\top M(\alpha) \frac{\partial \mathbf{P}(\alpha)}{\partial \alpha} M(\alpha) \frac{\partial h_{\boldsymbol{\theta}_\alpha}}{\partial \alpha} + \mathbf{e}_s^\top M(\alpha) \frac{\partial^2 h_{\boldsymbol{\theta}_\alpha}}{\partial \alpha^2}.$$

**Step 2:** Now we proceed with bounding the absolute value of each term which will contribute towards bounding the overall second derivative of the regulariser.

For the last term,

$$\left| \mathbf{e}_s^\top M(\alpha) \frac{\partial^2 h_{\boldsymbol{\theta}_\alpha}}{\partial \alpha^2} \Big|_{\alpha=0} \right| \leq \|\mathbf{e}_s^\top\|_1 \cdot \left\| M(\alpha) \frac{\partial^2 h_{\boldsymbol{\theta}_\alpha}}{\partial \alpha^2} \Big|_{\alpha=0} \right\|_\infty$$

$$\leq \frac{1}{1-\gamma} \cdot \left\| \frac{\partial^2 h_{\boldsymbol{\theta}_\alpha}}{\partial \alpha^2} \Big|_{\alpha=0} \right\|_\infty$$

$$\leq \frac{3 \cdot (1 + \log |\mathcal{A}|)}{1-\gamma} \cdot \|\mathbf{u}\|_2^2.$$

For the second last term,

$$\left| \mathbf{e}_s^\top M(\alpha) \frac{\partial \mathbf{P}(\alpha)}{\partial \alpha} M(\alpha) \frac{\partial h_{\boldsymbol{\theta}_\alpha}}{\partial \alpha} \Big|_{\alpha=0} \right| \leq \left\| M(\alpha) \frac{\partial \mathbf{P}(\alpha)}{\partial \alpha} M(\alpha) \frac{\partial h_{\boldsymbol{\theta}_\alpha}}{\partial \alpha} \Big|_{\alpha=0} \right\|_\infty$$

$$\leq \frac{1}{1-\gamma} \cdot \left\| \frac{\partial \mathbf{P}(\alpha)}{\partial \alpha} M(\alpha) \frac{\partial h_{\boldsymbol{\theta}_\alpha}}{\partial \alpha} \Big|_{\alpha=0} \right\|_\infty$$

$$\leq \frac{(C_1 + T_1) \cdot \|u\|_2}{1-\gamma} \cdot \left\| M(\alpha) \frac{\partial h_{\boldsymbol{\theta}_\alpha}}{\partial \alpha} \Big|_{\alpha=0} \right\|_\infty$$

$$\leq \frac{(C_1 + T_1) \cdot \|\mathbf{u}\|_2}{(1-\gamma)^2} \cdot \left\| \frac{\partial h_{\boldsymbol{\theta}_\alpha}}{\partial \alpha} \Big|_{\alpha=0} \right\|_\infty$$

$$\leq \frac{2 \cdot \log |\mathcal{A}|}{(1-\gamma)^2} (C_1 + T_1) \cdot \|\mathbf{u}\|_2^2.$$

For the second term,

$$\left| \mathbf{e}_s^\top M(\alpha) \frac{\partial^2 \mathbf{P}(\alpha)}{\partial \alpha^2} M(\alpha) h_{\boldsymbol{\theta}_\alpha} \Big|_{\alpha=0} \right| \leq \left\| M(\alpha) \frac{\partial^2 \mathbf{P}(\alpha)}{\partial \alpha^2} M(\alpha) h_{\boldsymbol{\theta}_\alpha} \Big|_{\alpha=0} \right\|_\infty$$

$$\leq \frac{1}{1-\gamma} \cdot \left\| \frac{\partial^2 \mathbf{P}(\alpha)}{\partial \alpha^2} M(\alpha) h_{\boldsymbol{\theta}_\alpha} \Big|_{\alpha=0} \right\|_\infty$$

$$\leq \frac{\|\mathbf{u}\|_2^2}{1-\gamma} \cdot \left\| M(\alpha) h_{\boldsymbol{\theta}_\alpha} \Big|_{\alpha=0} \right\|_\infty (C_2 + 2C_1 T_1 + T_2)$$

$$\leq \frac{\|\mathbf{u}\|_2^2}{(1-\gamma)^2} \cdot \left\| h_{\boldsymbol{\theta}_\alpha} \Big|_{\alpha=0} \right\|_\infty (C_2 + 2C_1 T_1 + T_2)$$

$$\leq \frac{\log |\mathcal{A}|}{(1-\gamma)^2} (C_2 + 2C_1 T_1 + T_2) \cdot \|\mathbf{u}\|_2^2.$$

For the first term,

$$\left| \mathbf{e}_s^\top M(\alpha) \frac{\partial \mathbf{P}(\alpha)}{\partial \alpha} M(\alpha) \frac{\partial \mathbf{P}(\alpha)}{\partial \alpha} M(\alpha) h_{\boldsymbol{\theta}_\alpha} \Big|_{\alpha=0} \right| \leq \left\| M(\alpha) \frac{\partial \mathbf{P}(\alpha)}{\partial \alpha} M(\alpha) \frac{\partial \mathbf{P}(\alpha)}{\partial \alpha} M(\alpha) h_{\boldsymbol{\theta}_\alpha} \Big|_{\alpha=0} \right\|_\infty$$

$$\leq \frac{1}{1-\gamma} \cdot \|\mathbf{u}\|_2 \cdot \frac{1}{1-\gamma} \cdot \|\mathbf{u}\|_2 \cdot \frac{1}{1-\gamma} \cdot \log |\mathcal{A}| \cdot (C_1 + T_1)^2$$

$$= \frac{\log |\mathcal{A}|}{(1-\gamma)^3} (C_1 + T_1)^2 \cdot \|\mathbf{u}\|_2^2.$$

**Step 3:** Now combining all the above equations, we get the final expression,

$$\max_{\|\mathbf{u}\|_2=1} \left\| \frac{\partial^2 \mathcal{H}_{\pi_{\theta_\alpha}}^{\pi_{\theta_\alpha}}(s)}{\partial \alpha^2} \bigg|_{\alpha=0} \right\|_\infty \le \beta_\lambda$$

where

$$\beta_\lambda = 2\gamma^2 \cdot \frac{3 \cdot (1 + \log|\mathcal{A}|)}{1-\gamma} + \gamma \cdot \frac{2 \cdot \log|\mathcal{A}|}{(1-\gamma)^2}(C_1 + T_1)$$

$$+ 2\gamma \cdot \frac{\log|\mathcal{A}|}{(1-\gamma)^2}(C_2 + 2C_1 T_1 + T_2) + \frac{\log|\mathcal{A}|}{(1-\gamma)^3}(C_1 + T_1)^2$$

$\square$

By definition of smoothness, the "soft performative value function" $\tilde{V}_\pi^\pi$ is Lipschitz smooth with Lipschitz constant $L_\lambda$ where $L_\lambda \triangleq L + \beta_\lambda$. Once again, we can choose $C_1, C_2, T_1, T_2$ according to Corollary 1 for simplification to get the order $\beta_\lambda = \mathcal{O}\left(\frac{\log|\mathcal{A}|}{(1-\gamma)^3}\psi_{\max}^2\right)$. Thus, the final bound for $L_\lambda$ as

$$L_\lambda = \mathcal{O}\left(\max\{L, \lambda\beta_\lambda\}\right) = \mathcal{O}\left(\max\left\{\frac{\gamma R_{\max}|\mathcal{A}|}{(1-\gamma)^2}, \frac{\lambda\log|\mathcal{A}|\psi_{\max}^2}{(1-\gamma)^3}\right\}\right). \tag{31}$$

## F  DERIVATION OF PERFORMATIVE POLICY GRADIENTS

**Theorem 2** (Performative Policy Gradient Theorem). *The gradient of the performative value function w.r.t $\boldsymbol{\theta}$ is as follows:*

*(a) For the unregularised objective,*

$$\nabla_{\boldsymbol{\theta}} V_{\boldsymbol{\pi_\theta}}^{\boldsymbol{\pi_\theta}}(\tau) = \mathbb{E}_{\tau \sim \mathbb{P}_{\boldsymbol{\pi_\theta}}^{\boldsymbol{\pi_\theta}}} \left[ \sum_{t=0}^{\infty} \gamma^t \left( A_{\boldsymbol{\pi_\theta}}^{\boldsymbol{\pi_\theta}}(s_t, a_t) \left( \nabla_{\boldsymbol{\theta}} \log \boldsymbol{\pi_\theta}(a_t \mid s_t) + \nabla_{\boldsymbol{\theta}} \log P_{\boldsymbol{\pi_\theta}}(s_{t+1}|s_t, a_t) \right) + \nabla_{\boldsymbol{\theta}} r_{\boldsymbol{\pi_\theta}}(s_t, a_t) \right) \right]. \quad (32)$$

*(b) For the entropy-regularised objective, we define the soft advantage, soft Q, and soft value functions with respect to the soft rewards $\tilde{r}_{\boldsymbol{\pi_\theta}}$ satisfying $\tilde{A}_{\boldsymbol{\pi_\theta}}^{\boldsymbol{\pi_\theta}}(s, a) = \tilde{Q}_{\boldsymbol{\pi_\theta}}^{\boldsymbol{\pi_\theta}}(s, a) - \tilde{V}_{\boldsymbol{\pi_\theta}}^{\boldsymbol{\pi_\theta}}(s)$ that further yields*

$$\nabla_{\boldsymbol{\theta}} \tilde{V}_{\boldsymbol{\pi_\theta}}^{\boldsymbol{\pi_\theta}}(\tau) = \mathbb{E}_{\tau \sim \mathbb{P}_{\boldsymbol{\pi_\theta}}^{\boldsymbol{\pi_\theta}}} \left[ \sum_{t=0}^{\infty} \gamma^t \left( \tilde{A}_{\boldsymbol{\pi_\theta}}^{\boldsymbol{\pi_\theta}}(s_t, a_t) \left( \nabla_{\boldsymbol{\theta}} \log \boldsymbol{\pi_\theta}(a_t \mid s_t) + \nabla_{\boldsymbol{\theta}} \log P_{\boldsymbol{\pi_\theta}}(s_{t+1}|s_t, a_t) \right) + \nabla_{\boldsymbol{\theta}} \tilde{r}_{\boldsymbol{\pi_\theta}}(s_t, a_t|\boldsymbol{\theta}) \right) \right]. \quad (33)$$

*Proof of Theorem 2.* We prove each part of this theorem separately.

*Proof of part (a).* First, we derive explicit closed form gradient for unregularised performative value function.

**Step 1.** Given a trajectory $\tau = \{s_0, a_0, \ldots, s_t, a_t, \ldots\}$, let us denote the unregularised objective function as

$$f_{\boldsymbol{\theta}}(\tau) = \sum_{t=0}^{\infty} \gamma^t r_{\boldsymbol{\pi_\theta}}(s_t, a_t)$$

Thus,

$$\nabla_{\boldsymbol{\theta}} V_{\boldsymbol{\pi_\theta}}^{\boldsymbol{\pi_\theta}}(\tau) = \nabla_{\boldsymbol{\theta}} \mathbb{E}_{\tau \sim \mathbb{P}_{\boldsymbol{\pi_\theta}}^{\boldsymbol{\pi_\theta}}}[f_{\boldsymbol{\theta}}(\tau)] = \nabla_{\boldsymbol{\theta}} \sum_{\tau} \mathbb{P}_{\boldsymbol{\pi_\theta}}^{\boldsymbol{\pi_\theta}}(\tau) f_{\boldsymbol{\theta}}(\tau)$$

$$= \sum_{\tau} \nabla_{\boldsymbol{\theta}} (\mathbb{P}_{\boldsymbol{\pi_\theta}}^{\boldsymbol{\pi_\theta}}(\tau) f_{\boldsymbol{\theta}}(\tau))$$

$$= \sum_{\tau} (\nabla_{\boldsymbol{\theta}} \mathbb{P}_{\boldsymbol{\pi_\theta}}^{\boldsymbol{\pi_\theta}}(\tau)) f_{\boldsymbol{\theta}}(\tau) + \sum_{\tau} \mathbb{P}_{\boldsymbol{\pi_\theta}}^{\boldsymbol{\pi_\theta}}(\tau) (\nabla_{\boldsymbol{\theta}} f_{\boldsymbol{\theta}}(\tau))$$

$$\overset{(a)}{=} \sum_{\tau} \mathbb{P}_{\boldsymbol{\pi_\theta}}^{\boldsymbol{\pi_\theta}}(\tau) (\nabla_{\boldsymbol{\theta}} \log \mathbb{P}_{\boldsymbol{\pi_\theta}}^{\boldsymbol{\pi_\theta}}(\tau)) f_{\boldsymbol{\theta}}(\tau) + \mathbb{E}_{\tau \sim \mathbb{P}_{\boldsymbol{\pi_\theta}}^{\boldsymbol{\pi_\theta}}}[\nabla_{\boldsymbol{\theta}} f_{\boldsymbol{\theta}}(\tau)]$$

$$= \mathbb{E}_{\tau \sim \mathbb{P}_{\boldsymbol{\pi_\theta}}^{\boldsymbol{\pi_\theta}}} \left[ (\nabla_{\boldsymbol{\theta}} \log \mathbb{P}_{\boldsymbol{\pi_\theta}}^{\boldsymbol{\pi_\theta}}(\tau)) f_{\boldsymbol{\theta}}(\tau) \right] + \mathbb{E}_{\tau \sim \mathbb{P}_{\boldsymbol{\pi_\theta}}^{\boldsymbol{\pi_\theta}}}[\nabla_{\boldsymbol{\theta}} f_{\boldsymbol{\theta}}(\tau)].$$

$(a)$ holds since $\nabla_{\boldsymbol{\theta}} \log \mathbb{P}_{\boldsymbol{\pi_\theta}}^{\boldsymbol{\pi_\theta}}(\tau) = \frac{\nabla_{\boldsymbol{\theta}} \mathbb{P}_{\boldsymbol{\pi_\theta}}^{\boldsymbol{\pi_\theta}}(\tau)}{\mathbb{P}_{\boldsymbol{\pi_\theta}}^{\boldsymbol{\pi_\theta}}(\tau)}$.

**Step 2.** Given the initial state distribution $\boldsymbol{\rho}$, we further have

$$\log \mathbb{P}_{\boldsymbol{\pi_\theta}}^{\boldsymbol{\pi_\theta}}(\tau) = \log \boldsymbol{\rho}(s_0) + \sum_{t=0}^{\infty} \log \boldsymbol{\pi_\theta}(a_t \mid s_t) + \sum_{t=0}^{\infty} \log \mathbf{P}_{\boldsymbol{\pi_\theta}}(s_{t+1}|s_t, a_t)$$

Taking the gradient with respect to $\boldsymbol{\theta}$, we obtain

$$\nabla_{\boldsymbol{\theta}} \log \mathbb{P}_{\boldsymbol{\pi_\theta}}^{\boldsymbol{\pi_\theta}}(\tau) = \sum_{t=0}^{\infty} \nabla_{\boldsymbol{\theta}} \log \boldsymbol{\pi_\theta}(a_t \mid s_t) + \sum_{t=0}^{\infty} \nabla_{\boldsymbol{\theta}} \log \mathbf{P}_{\boldsymbol{\pi_\theta}}(s_{t+1}|s_t, a_t)$$

**Step 3.** Now, by substituting the value of $\nabla_{\boldsymbol{\theta}} \log(\mathbf{P}_{\boldsymbol{\pi_\theta}}^{\boldsymbol{\pi_\theta}})$ in $\nabla_{\boldsymbol{\theta}} V_{\boldsymbol{\pi_\theta}}^{\boldsymbol{\pi_\theta}}(\tau)$, we get,

$$\nabla_{\boldsymbol{\theta}} V_{\boldsymbol{\pi_\theta}}^{\boldsymbol{\pi_\theta}}(\tau) = \nabla_{\boldsymbol{\theta}} \mathbb{E}_{\tau \sim \mathbb{P}_{\boldsymbol{\pi_\theta}}^{\boldsymbol{\pi_\theta}}}[f_{\boldsymbol{\theta}}(\tau)] = \mathbb{E}_{\tau \sim \mathbb{P}_{\boldsymbol{\pi_\theta}}^{\boldsymbol{\pi_\theta}}} \left[ \left( \sum_{t=0}^{\infty} \nabla_{\boldsymbol{\theta}} \log \boldsymbol{\pi_\theta}(a_t \mid s_t) \right) \cdot \left( \sum_{t=0}^{\infty} \gamma^t r_{\boldsymbol{\pi_\theta}}(s_t, a_t) \right) \right]$$

$$+ \mathbb{E}_{\tau \sim \mathbb{P}_{\boldsymbol{\pi_\theta}}^{\boldsymbol{\pi_\theta}}} \left[ \left( \sum_{t=1}^{\infty} \nabla_{\boldsymbol{\theta}} \log \mathbf{P}_{\boldsymbol{\pi_\theta}}(s_t|s_{t-1}, a_{t-1}) \right) \cdot \left( \sum_{t=0}^{\infty} \gamma^t r_{\boldsymbol{\pi_\theta}}(s_t, a_t) \right) \right]$$

$$+ \mathbb{E}_{\tau \sim \mathbb{P}_{\pi_{\theta}}^{\pi_{\theta}}} \left[ \sum_{t=0}^{\infty} \gamma^t \nabla_{\theta} r_{\pi_{\theta}}(s_t, a_t) \right]$$

$$= \mathbb{E}_{\tau \sim \mathbb{P}_{\pi_{\theta}}^{\pi_{\theta}}} \left[ \sum_{t=0}^{\infty} \gamma^t A_{\pi_{\theta}}^{\pi_{\theta}}(s_t, a_t) \nabla_{\theta} \log \pi_{\theta}(a_t \mid s_t) \right]$$

$$+ \mathbb{E}_{\tau \sim \mathbb{P}_{\pi_{\theta}}^{\pi_{\theta}}} \left[ \sum_{t=1}^{\infty} \gamma^t A_{\pi_{\theta}}^{\pi_{\theta}}(s_t, a_t) \nabla_{\theta} \log \mathbf{P}_{\pi_{\theta}}(s_t | s_{t-1}, a_{t-1}) \right]$$

$$+ \mathbb{E}_{\tau \sim \mathbb{P}_{\pi_{\theta}}^{\pi_{\theta}}} \left[ \sum_{t=0}^{\infty} \gamma^t \nabla_{\theta} r_{\pi_{\theta}}(s_t, a_t) \right] .$$

The last equality is due to the definition of advantage function

$$A_{\pi_{\theta}}^{\pi_{\theta}}(s_t, a_t) \triangleq \sum_{i=t}^{\infty} \gamma^{t-i} r_{\pi_{\theta}}(s_i, \pi_{\theta}(s_i)) - \mathbb{E}_{\substack{s_{t'+1} \sim \mathbf{P}_{\pi_{\theta}}^{\pi_{\theta}}(\cdot | s_{t'}, a_{t'}) \\ \forall t' \in [t, \infty)}} \left[ \sum_{i=t}^{\infty} \gamma^{t-i} r_{\pi_{\theta}}(s_i, \pi_{\theta}(s_i)) \right] \triangleq Q_{\pi_{\theta}}^{\pi_{\theta}}(s_t) - V_{\pi_{\theta}}^{\pi_{\theta}}(s_t)$$

as in classical policy gradient theorem. Hence, we conclude the proof for part (a) of the theorem.

*Proof of part (b).* Now, we derive explicit gradient form for entropy-regularised value function.

Let us define the soft reward as $\tilde{r}_{\pi_{\theta}}(s_t, a_t) \triangleq r_{\pi_{\theta}}(s_t, a_t) - \lambda \log \pi_{\theta}(a_t | s_t)$. Again, we start by defining regularised objective function

$$\tilde{f}_{\theta}(\tau) = \sum_{t=0}^{\infty} \gamma^t \tilde{r}_{\pi_{\theta}}(s_t, a_t)$$

Following the same steps as that of *Part (a)*, we get

$$\nabla_{\theta} \tilde{V}_{\pi_{\theta}}^{\pi_{\theta}}(\tau) = \nabla_{\theta} \mathbb{E}_{\tau \sim \mathbb{P}_{\pi_{\theta}}^{\pi_{\theta}}}[\tilde{f}_{\theta}(\tau)] = \mathbb{E}_{\tau \sim \mathbb{P}_{\pi_{\theta}}^{\pi_{\theta}}} \left[ \sum_{t=0}^{\infty} \gamma^t \tilde{A}_{\pi_{\theta}}^{\pi_{\theta}}(s_t, a_t) \nabla_{\theta} \log \pi_{\theta}(a_t \mid s_t) \right]$$

$$+ \mathbb{E}_{\tau \sim \mathbb{P}_{\pi_{\theta}}^{\pi_{\theta}}} \left[ \sum_{t=1}^{\infty} \gamma^t \tilde{A}_{\pi_{\theta}}^{\pi_{\theta}}(s_t, a_t) \nabla_{\theta} \log \mathbf{P}_{\pi_{\theta}}(s_t | s_{t-1}, a_{t-1}) \right]$$

$$+ \mathbb{E}_{\tau \sim \mathbb{P}_{\pi_{\theta}}^{\pi_{\theta}}} \left[ \sum_{t=0}^{\infty} \gamma^t \nabla_{\theta} \tilde{r}_{\pi_{\theta}}(s_t, a_t) \right] .$$

$$= \mathbb{E}_{\tau \sim \mathbb{P}_{\pi_{\theta}}^{\pi_{\theta}}} \left[ \sum_{t=0}^{\infty} \gamma^t \tilde{A}_{\pi_{\theta}}^{\pi_{\theta}}(s_t, a_t) \nabla_{\theta} \log \pi_{\theta}(a_t \mid s_t) \right]$$

$$+ \mathbb{E}_{\tau \sim \mathbb{P}_{\pi_{\theta}}^{\pi_{\theta}}} \left[ \sum_{t=1}^{\infty} \gamma^t \tilde{A}_{\pi_{\theta}}^{\pi_{\theta}}(s_t, a_t) \nabla_{\theta} \log \mathbf{P}_{\pi_{\theta}}(s_t | s_{t-1}, a_{t-1}) \right]$$

$$+ \mathbb{E}_{\tau \sim \mathbb{P}_{\pi_{\theta}}^{\pi_{\theta}}} \left[ \sum_{t=0}^{\infty} \gamma^t \nabla_{\theta} r_{\pi_{\theta}}(s_t, a_t) \right] - \lambda \mathbb{E}_{\tau \sim \mathbb{P}_{\pi_{\theta}}^{\pi_{\theta}}} \left[ \sum_{t=0}^{\infty} \gamma^t \nabla_{\theta} \log \pi_{\theta}(a_t | s_t) \right]$$

Here,

$$\tilde{A}_{\pi_{\theta}}^{\pi_{\theta}}(s_t, a_t) \triangleq \sum_{i=t}^{\infty} \gamma^{t-i} \tilde{r}_{\pi_{\theta}}(s_i, \pi_{\theta}(s_i)) - \mathbb{E}_{\substack{s_{t'+1} \sim \mathbf{P}_{\pi_{\theta}}^{\pi_{\theta}}(\cdot | s_{t'}, a_{t'}) \\ \forall t' \in [t, \infty)}} \left[ \sum_{i=t}^{\infty} \gamma^{t-i} \tilde{r}_{\pi_{\theta}}(s_i, \pi_{\theta}(s_i)) \right] \triangleq \tilde{Q}_{\pi_{\theta}}^{\pi_{\theta}}(s_t, a_t) - \tilde{V}_{\pi_{\theta}}^{\pi_{\theta}}(s_t)$$

denotes the advantage function with soft rewards, or in brief, the soft advantage function. Hence, we conclude proof of part (b). □

# G  CONVERGENCE OF PePG : PROOFS OF SECTION 4

## G.1  PROOFS FOR UNREGULARISED VALUE FUNCTION

**Lemma 6** (Performative Policy Gradient for Softmax PeMDPs). *Given softmax PeMDPs defined by* (13)*, for all* $(s, a, s') \in (\mathcal{S}, \mathcal{A}, \mathcal{S})$*, derivative of the performative value function w.r.t* $\boldsymbol{\theta}_{s,a}$ *satisfies:*

$$\frac{\partial V_{\pi_{\boldsymbol{\theta}}}^{\pi_{\boldsymbol{\theta}}}(\rho)}{\partial \boldsymbol{\theta}_{s,a}} \geq \frac{1}{1-\gamma} d_{\pi_{\boldsymbol{\theta}}}^{\pi_{\boldsymbol{\theta}}}(s, a|\rho) \left(A_{\pi_{\boldsymbol{\theta}}}^{\pi_{\boldsymbol{\theta}}}(s, a) + \xi\right). \tag{34}$$

*Proof.* First, we note that

$$\frac{\partial}{\partial \boldsymbol{\theta}_{s',a'}} \log \pi_{\boldsymbol{\theta}}(a|s) = \mathbb{1}[s = s', a = a'] - \pi_{\boldsymbol{\theta}}(a'|s)\mathbb{1}[s = s']$$

$$\frac{\partial}{\partial \boldsymbol{\theta}_{s',a'}} \log \mathbf{P}_{\pi_{\boldsymbol{\theta}}}(s''|s, a) = \psi(s'')\mathbb{1}[s = s', a = a'] \left(1 - \mathbf{P}_{\pi_{\boldsymbol{\theta}}}(s''|s, a)\right)$$

$$\frac{\partial}{\partial \boldsymbol{\theta}_{s',a'}} r_{\pi_{\boldsymbol{\theta}}}(s, a) = \xi \mathbb{1}[s = s', a = a']. \tag{35}$$

In this proof, we further substitute the expressions of individual gradients in Equation (35) into Equation (10).

Therefore, for a given initial state distribution $\boldsymbol{\rho}$, we get

$$\frac{\partial}{\partial \boldsymbol{\theta}_{s,a}} V_{\pi_{\boldsymbol{\theta}}}^{\pi_{\boldsymbol{\theta}}}(\boldsymbol{\rho}) = \mathbb{E}_{\tau \sim \mathbb{P}_{\pi_{\boldsymbol{\theta}}}^{\pi_{\boldsymbol{\theta}}}} \Bigg[ \sum_{t=0}^{\infty} \gamma^t \Big( A_{\pi_{\boldsymbol{\theta}}}^{\pi_{\boldsymbol{\theta}}}(s_t, a_t) \frac{\partial}{\partial \boldsymbol{\theta}_{s,a}} \log \pi_{\boldsymbol{\theta}}(a_t \mid s_t)$$

$$+ A_{\pi_{\boldsymbol{\theta}}}^{\pi_{\boldsymbol{\theta}}}(s_t, a_t) \frac{\partial}{\partial \boldsymbol{\theta}_{s,a}} \log P_{\pi_{\boldsymbol{\theta}}}(s_{t+1}|s_t, a_t)$$

$$+ \frac{\partial}{\partial \boldsymbol{\theta}_{s,a}} r_{\pi_{\boldsymbol{\theta}}}(s_t, a_t) \Big) \Bigg]$$

$$= \mathbb{E}_{\tau \sim \mathbb{P}_{\pi_{\boldsymbol{\theta}}}^{\pi_{\boldsymbol{\theta}}}} \Bigg[ \sum_{t=0}^{\infty} \gamma^t \Big( A_{\pi_{\boldsymbol{\theta}}}^{\pi_{\boldsymbol{\theta}}}(s_t, a_t) \left(\mathbb{1}[s_t = s, a_t = a] - \pi_{\boldsymbol{\theta}}(a|s)\mathbb{1}[s_t = s]\right)$$

$$+ A_{\pi_{\boldsymbol{\theta}}}^{\pi_{\boldsymbol{\theta}}}(s_t, a_t)\psi(s_{t+1})\mathbb{1}[s_t = s, a_t = a]\left(1 - \mathbf{P}_{\pi_{\boldsymbol{\theta}}}(s_{t+1}|s, a)\right)$$

$$+ \xi\mathbb{1}[s_t = s, a_t = a]\Big) \Bigg]$$

$$\underset{(a)}{\geq} \mathbb{E}_{\tau \sim \mathbb{P}_{\pi_{\boldsymbol{\theta}}}^{\pi_{\boldsymbol{\theta}}}} \Bigg[ \sum_{t=0}^{\infty} \gamma^t A_{\pi_{\boldsymbol{\theta}}}^{\pi_{\boldsymbol{\theta}}}(s_t, a_t)\mathbb{1}[s_t = s, a_t = a] \Bigg] - \mathbb{E}_{\tau \sim \mathbb{P}_{\pi_{\boldsymbol{\theta}}}^{\pi_{\boldsymbol{\theta}}}} \Bigg[ \sum_{t=0}^{\infty} \gamma^t \pi_{\boldsymbol{\theta}}(a|s)\mathbb{1}[s_t = s]A_{\pi_{\boldsymbol{\theta}}}^{\pi_{\boldsymbol{\theta}}}(s_t, a_t) \Bigg]$$

$$+ \mathbb{E}_{\tau \sim \mathbb{P}_{\pi_{\boldsymbol{\theta}}}^{\pi_{\boldsymbol{\theta}}}} \Bigg[ \sum_{t=0}^{\infty} \gamma^t \xi\mathbb{1}[s_t = s, a_t = a] \Bigg]$$

$$\underset{(b)}{=} \frac{1}{1-\gamma} d_{\pi_{\boldsymbol{\theta}}, \rho}^{\pi_{\boldsymbol{\theta}}}(s, a)A_{\pi_{\boldsymbol{\theta}}}^{\pi_{\boldsymbol{\theta}}}(s, a) + \frac{1}{1-\gamma}\xi d_{\pi_{\boldsymbol{\theta}}, \rho}^{\pi_{\boldsymbol{\theta}}}(s, a)$$

(a) is due to the fact that $1 - \mathbf{P}_{\pi_{\boldsymbol{\theta}}}(s, a) \geq 0$ for all $s, a$. (b) is due to $\mathbb{E}_{\tau \sim \mathbb{P}_{\pi_{\boldsymbol{\theta}}}^{\pi_{\boldsymbol{\theta}}}} \left[ \sum_{t=0}^{\infty} \gamma^t \pi_{\boldsymbol{\theta}}(a|s)\mathbb{1}[s_t = s]A_{\pi_{\boldsymbol{\theta}}}^{\pi_{\boldsymbol{\theta}}}(s_t, a_t) \right] = 0$.

$\square$

**Lemma 3.** *Performative Gradient Domination for Softmax PeMDPs Let us consider PeMDPs defined in* (13)*.*

*(a) For unregularised value function,*

$$V_{\pi_o^\star}^{\pi_o^\star}(\rho) - V_{\pi_{\boldsymbol{\theta}}}^{\pi_{\boldsymbol{\theta}}}(\rho) \leq \sqrt{|\mathcal{S}||\mathcal{A}|} \left\| \frac{\boldsymbol{d}_{\pi_{\boldsymbol{\theta}}, \rho}^{\pi_o^\star}}{\boldsymbol{d}_{\pi_{\boldsymbol{\theta}}, \nu}^{\pi_{\boldsymbol{\theta}}}} \right\|_{\infty} \|\nabla_{\boldsymbol{\theta}} V_{\pi_{\boldsymbol{\theta}}}^{\pi_{\boldsymbol{\theta}}}(\nu)\|_2 + \frac{R_{\max}}{1-\gamma}(1 + \frac{2\gamma}{1-\gamma}\psi_{\max}). \tag{36}$$

*Proof of Lemma 3– Part (a).* This proof is divided into two parts. In the first part we bound the expected advantage term from Lemma 2 with the norm of the gradient of value function. During this step, we need to express the expected advantage

as a linear combination of the advantage itself and the occupancy measure over all states and actions like in equation (34). The expectation however is taken w.r.t the occupancy measure $\boldsymbol{d}_{\boldsymbol{\pi_\theta}}^{\boldsymbol{\pi_o^\star}}$, thus we need to perform a change of measure which introduces a coverage term as shown below. In the second step we directly use the bound of rewards and transitions obtained from their Lipchitzness in lemma 2. We know by Lemma 1 that

$$V_{\boldsymbol{\pi_o^\star}}^{\boldsymbol{\pi_o^\star}}(\rho) - V_{\boldsymbol{\pi_\theta}}^{\boldsymbol{\pi_\theta}}(\rho) = \frac{1}{1-\gamma}\mathbb{E}_{(s,a)\sim \boldsymbol{d}_{\boldsymbol{\pi_\theta},\rho}^{\boldsymbol{\pi_o^\star}}(\cdot|\rho)}[A_{\boldsymbol{\pi_\theta}}^{\boldsymbol{\pi_\theta}}(s,a)]$$
$$+ \frac{1}{1-\gamma}\mathbb{E}_{(s,a)\sim \boldsymbol{d}_{\boldsymbol{\pi_\theta},\rho}^{\boldsymbol{\pi_o^\star}}}[(r_{\boldsymbol{\pi_o^\star}}(s,a) - r_{\boldsymbol{\pi_\theta}}(s,a)) + \gamma(\mathbf{P}_{\boldsymbol{\pi_o^\star}}(\cdot|s,a) - \mathbf{P}_{\boldsymbol{\pi_\theta}}(\cdot|s,a))^\top V_{\boldsymbol{\pi_o^\star}}^{\boldsymbol{\pi_o^\star}}(\cdot)].$$

**Step 1: Upper bounding Term 1.**

$$\mathbb{E}_{(s,a)\sim \boldsymbol{d}_{\boldsymbol{\pi_\theta},\rho}^{\boldsymbol{\pi_o^\star}}}[A_{\boldsymbol{\pi_\theta}}^{\boldsymbol{\pi_\theta}}(s,a)] = \sum_{s,a} \boldsymbol{d}_{\boldsymbol{\pi_o^\star}}^{\boldsymbol{\pi_o^\star}}(s,a|\rho) A_{\boldsymbol{\pi_\theta}}^{\boldsymbol{\pi_\theta}}(s,a) = \sum_{s,a} \frac{\boldsymbol{d}_{\boldsymbol{\pi_\theta}}^{\boldsymbol{\pi_o^\star}}(s,a|\rho)}{\boldsymbol{d}_{\boldsymbol{\pi_\theta}}^{\boldsymbol{\pi_\theta}}(s,a|\nu)} \boldsymbol{d}_{\boldsymbol{\pi_\theta}}^{\boldsymbol{\pi_\theta}}(s,a|\nu) A_{\boldsymbol{\pi_\theta}}^{\boldsymbol{\pi_\theta}}(s,a)$$
$$\leq \left\| \frac{\boldsymbol{d}_{\boldsymbol{\pi_\theta},\rho}^{\boldsymbol{\pi_o^\star}}}{\boldsymbol{d}_{\boldsymbol{\pi_\theta},\nu}^{\boldsymbol{\pi_\theta}}} \right\|_\infty \sum_{s,a} \boldsymbol{d}_{\boldsymbol{\pi_\theta}}^{\boldsymbol{\pi_\theta}}(s,a|\nu) A_{\boldsymbol{\pi_\theta}}^{\boldsymbol{\pi_\theta}}(s,a) \tag{37}$$

Now, we leverage the gradient of softmax performative MDPs to obtain

$$\sum_{s,a} \boldsymbol{d}_{\boldsymbol{\pi_\theta}}^{\boldsymbol{\pi_\theta}}(s,a|\nu) A_{\boldsymbol{\pi_\theta}}^{\boldsymbol{\pi_\theta}}(s,a) \leq (1-\gamma)\sum_{s,a} \frac{\partial V_{\boldsymbol{\pi_\theta}}^{\boldsymbol{\pi_\theta}}(\nu)}{\partial \boldsymbol{\theta}_{s,a}} - \xi$$
$$= (1-\gamma)\mathbf{1}^\top \nabla_{\boldsymbol{\theta}} V_{\boldsymbol{\pi_\theta}}^{\boldsymbol{\pi_\theta}}(\nu) - \xi$$
$$\leq (1-\gamma)\sqrt{|\mathcal{S}||\mathcal{A}|}\|\nabla_{\boldsymbol{\theta}} V_{\boldsymbol{\pi_\theta}}^{\boldsymbol{\pi_\theta}}(\nu)\|_2 - \xi$$

The last inequality is obtained by applying Cauchy-Schwarz inequality.

Now, substituting the above result back in Equation (37), we get

$$\frac{1}{1-\gamma}\mathbb{E}_{(s,a)\sim \boldsymbol{d}_{\boldsymbol{\pi_\theta},\rho}^{\boldsymbol{\pi_o^\star}}(\cdot|s_0)}[A_{\boldsymbol{\pi_\theta}}^{\boldsymbol{\pi_\theta}}(s,a)] \leq \sqrt{|\mathcal{S}||\mathcal{A}|}\left\| \frac{\boldsymbol{d}_{\boldsymbol{\pi_\theta},\rho}^{\boldsymbol{\pi_o^\star}}}{\boldsymbol{d}_{\boldsymbol{\pi_\theta},\nu}^{\boldsymbol{\pi_\theta}}} \right\|_\infty \|\nabla_{\boldsymbol{\theta}} V_{\boldsymbol{\pi_\theta}}^{\boldsymbol{\pi_\theta}}(\nu)\|_2 - \left\| \frac{\boldsymbol{d}_{\boldsymbol{\pi_\theta},\rho}^{\boldsymbol{\pi_o^\star}}}{\boldsymbol{d}_{\boldsymbol{\pi_\theta},\nu}^{\boldsymbol{\pi_\theta}}} \right\|_\infty \frac{\xi}{1-\gamma} \tag{38}$$

**Step 2: Upper bounding Term 2.** For softmax rewards and transitions, we further obtain from Lemma 2,

$$\text{Term 2} = \frac{1}{1-\gamma}\mathbb{E}_{(s,a)\sim \boldsymbol{d}_{\boldsymbol{\pi_\theta},\rho}^{\boldsymbol{\pi_o^\star}}}\left[(r_{\boldsymbol{\pi_o^\star}}(s,a) - r_{\boldsymbol{\pi_\theta}}(s,a)) + \gamma(\mathbf{P}_{\boldsymbol{\pi_o^\star}}(\cdot|s,a) - \mathbf{P}_{\boldsymbol{\pi_\theta}}(\cdot|s,a))^\top V_{\boldsymbol{\pi_o^\star}}^{\boldsymbol{\pi_o^\star}}(\cdot)\right]$$
$$\leq \frac{1}{1-\gamma}(\xi + \frac{\gamma}{1-\gamma}R_{\max}\psi_{\max})\|\boldsymbol{\pi}_o^\star(\cdot|s_0) - \boldsymbol{\pi_\theta}(\cdot|s_0)\|_1 \tag{39}$$
$$\leq \frac{2}{1-\gamma}(\xi + \frac{\gamma}{1-\gamma}R_{\max}\psi_{\max}). \tag{40}$$

**Step 3:** Now, if we use Equation (38) and (40) together, we get

$$V_{\boldsymbol{\pi_o^\star}}^{\boldsymbol{\pi_o^\star}}(\rho) - V_{\boldsymbol{\pi_\theta}}^{\boldsymbol{\pi_\theta}}(\rho) \leq \sqrt{|\mathcal{S}||\mathcal{A}|}\left\| \frac{\boldsymbol{d}_{\boldsymbol{\pi_\theta},\rho}^{\boldsymbol{\pi_o^\star}}}{\boldsymbol{d}_{\boldsymbol{\pi_\theta},\nu}^{\boldsymbol{\pi_\theta}}} \right\|_\infty \|\nabla_{\boldsymbol{\theta}} V_{\boldsymbol{\pi_\theta}}^{\boldsymbol{\pi_\theta}}(\nu)\|_2 + \left(2 - \left\| \frac{\boldsymbol{d}_{\boldsymbol{\pi_\theta},\rho}^{\boldsymbol{\pi_o^\star}}}{\boldsymbol{d}_{\boldsymbol{\pi_\theta},\nu}^{\boldsymbol{\pi_\theta}}} \right\|_\infty\right)\frac{\xi}{1-\gamma} + + \frac{2\gamma}{(1-\gamma)^2}R_{\max}\psi_{\max}$$
$$\leq \sqrt{|\mathcal{S}||\mathcal{A}|}\left\| \frac{\boldsymbol{d}_{\boldsymbol{\pi_\theta},\rho}^{\boldsymbol{\pi_o^\star}}}{\boldsymbol{d}_{\boldsymbol{\pi_\theta},\nu}^{\boldsymbol{\pi_\theta}}} \right\|_\infty \|\nabla_{\boldsymbol{\theta}} V_{\boldsymbol{\pi_\theta}}^{\boldsymbol{\pi_\theta}}(\nu)\|_2 + \frac{R_{\max}}{1-\gamma} + \frac{2\gamma}{(1-\gamma)^2}R_{\max}\psi_{\max}$$
$$= \sqrt{|\mathcal{S}||\mathcal{A}|}\left\| \frac{\boldsymbol{d}_{\boldsymbol{\pi_\theta},\rho}^{\boldsymbol{\pi_o^\star}}}{\boldsymbol{d}_{\boldsymbol{\pi_\theta},\nu}^{\boldsymbol{\pi_\theta}}} \right\|_\infty \|\nabla_{\boldsymbol{\theta}} V_{\boldsymbol{\pi_\theta}}^{\boldsymbol{\pi_\theta}}(\nu)\|_2 + \frac{R_{\max}}{1-\gamma}\left(1 + \frac{2\gamma}{1-\gamma}\psi_{\max}\right)$$

The last inequality is true since $\left\| \frac{\boldsymbol{d}_{\boldsymbol{\pi_\theta},\rho}^{\boldsymbol{\pi_o^\star}}}{\boldsymbol{d}_{\boldsymbol{\pi_\theta},\nu}^{\boldsymbol{\pi_\theta}}} \right\|_\infty \geq 1$ (Lemma 9) and $\xi \leq R_{\max}$.

**Theorem 3** (Convergence of PePG in softmax PeMDPs – Part (a))**.** *Let* $\mathrm{Cov} \triangleq \max_{\boldsymbol{\theta}, \nu} \left\| \frac{d^{\boldsymbol{\pi}_o^\star}_{\boldsymbol{\pi}_{\boldsymbol{\theta}}, \rho}}{d^{\boldsymbol{\pi}_{\boldsymbol{\theta}}}_{\boldsymbol{\pi}_{\boldsymbol{\theta}}, \nu}} \right\|_\infty$ . *The gradient ascent algorithm on* $V^{\boldsymbol{\pi}_{\boldsymbol{\theta}}}_{\boldsymbol{\pi}_{\boldsymbol{\theta}}}(\rho)$ *(Equation* (9)*) with step size* $\eta = \Omega(\min\{\frac{(1-\gamma)^2}{\gamma|\mathcal{A}|}, \frac{(1-\gamma)^3}{\gamma^2}\})$ *satisfies, for all distributions* $\rho \in \Delta(\mathcal{S})$.

*(a) For unregularised case,*

$$
\min_{t < T} \left\{ V^{\boldsymbol{\pi}_o^\star}_{\boldsymbol{\pi}_o^\star}(\rho) - V^{\boldsymbol{\pi}_{\boldsymbol{\theta}_t}}_{\boldsymbol{\pi}_{\boldsymbol{\theta}_t}}(\rho) \right\} \leq \epsilon \text{ when } T = \Omega \left( \frac{|\mathcal{S}||\mathcal{A}|}{\epsilon^2} \max \left\{ \frac{\gamma R_{\max} |\mathcal{A}|}{(1-\gamma)^3}, \frac{\gamma^2}{(1-\gamma)^4} \right\} \right), \text{ and } \epsilon = \Omega \left( \frac{1}{1-\gamma} \right).
$$

*Proof of Theorem 3– Part (a).* We proceed with this proof by dividing it in four steps. In the first step, we use the smoothness of the value function to prove an upper bound for the minimum squared gradient norm of the value over time which is a constant times $1/T$ . In the second step, we derive a lower bound on the norm of gradient of value function using Lemma 3. In the final two steps, we combine the bounds obtained from the first two steps to derive lower bounds for $T$ and $\epsilon$, i.e. the error threshold.

**Step 1:** As $V^{\boldsymbol{\pi}_{\boldsymbol{\theta}}}_{\boldsymbol{\pi}_{\boldsymbol{\theta}}}$ is $L$-smooth (Lemma 4), it satisfies

$$
\left| V^{\boldsymbol{\pi}_{\boldsymbol{\theta}}}_{\boldsymbol{\pi}_{\boldsymbol{\theta}}}(\rho) - V^{\boldsymbol{\pi}'_{\boldsymbol{\theta}}}_{\boldsymbol{\pi}'_{\boldsymbol{\theta}}}(\rho) - \langle \nabla_{\boldsymbol{\theta}} V^{\boldsymbol{\pi}_{\boldsymbol{\theta}}}_{\boldsymbol{\pi}_{\boldsymbol{\theta}}}(\rho), \boldsymbol{\theta} - \boldsymbol{\theta}' \rangle \right| \leq \frac{L}{2} \|\boldsymbol{\theta} - \boldsymbol{\theta}'\|^2
$$

Thus, taking $\boldsymbol{\theta}$ as $\boldsymbol{\theta}_{t+1}$ and $\boldsymbol{\theta}'$ as $\boldsymbol{\theta}_t$ and using the gradient ascent expression (Equation (9)) yields

$$
\left| V^{\boldsymbol{\pi}^{(t+1)}_{\boldsymbol{\theta}}}_{\boldsymbol{\pi}^{(t+1)}_{\boldsymbol{\theta}}}(\rho) - V^{\boldsymbol{\pi}^{(t)}_{\boldsymbol{\theta}}}_{\boldsymbol{\pi}^{(t)}_{\boldsymbol{\theta}}}(\rho) - \eta \|\nabla_{\boldsymbol{\theta}} V^{\boldsymbol{\pi}^{(t)}_{\boldsymbol{\theta}}}_{\boldsymbol{\pi}^{(t)}_{\boldsymbol{\theta}}}(\rho)\|^2 \right| \leq \frac{L}{2} \|\boldsymbol{\theta}_{t+1} - \boldsymbol{\theta}_t\|^2
$$

$$
\implies \qquad V^{\boldsymbol{\pi}^{(t+1)}_{\boldsymbol{\theta}}}_{\boldsymbol{\pi}^{(t+1)}_{\boldsymbol{\theta}}}(\rho) - V^{\boldsymbol{\pi}^{(t)}_{\boldsymbol{\theta}}}_{\boldsymbol{\pi}^{(t)}_{\boldsymbol{\theta}}}(\rho) \geq \eta \|\nabla_{\boldsymbol{\theta}} V^{\boldsymbol{\pi}^{(t)}_{\boldsymbol{\theta}}}_{\boldsymbol{\pi}^{(t)}_{\boldsymbol{\theta}}}(\rho)\|^2 - \frac{L}{2} \|\boldsymbol{\theta}_{t+1} - \boldsymbol{\theta}_t\|^2
$$

This further implies that

$$
V^{\boldsymbol{\pi}^{(t+1)}_{\boldsymbol{\theta}}}_{\boldsymbol{\pi}^{(t+1)}_{\boldsymbol{\theta}}}(\rho) - V^{\boldsymbol{\pi}_o^\star}_{\boldsymbol{\pi}_o^\star}(\rho) \geq V^{\boldsymbol{\pi}^{(t)}_{\boldsymbol{\theta}}}_{\boldsymbol{\pi}^{(t)}_{\boldsymbol{\theta}}}(\rho) - V^{\boldsymbol{\pi}_o^\star}_{\boldsymbol{\pi}_o^\star}(\rho) + \eta \|\nabla_{\boldsymbol{\theta}} V^{\boldsymbol{\pi}^{(t)}_{\boldsymbol{\theta}}}_{\boldsymbol{\pi}^{(t)}_{\boldsymbol{\theta}}}(\rho)\|^2 - \frac{L}{2} \|\boldsymbol{\theta}_{t+1} - \boldsymbol{\theta}_t\|^2
$$

$$
= V^{\boldsymbol{\pi}^{(t)}_{\boldsymbol{\theta}}}_{\boldsymbol{\pi}^{(t)}_{\boldsymbol{\theta}}}(\rho) - V^{\boldsymbol{\pi}_o^\star}_{\boldsymbol{\pi}_o^\star}(\rho) + \eta(1 - \frac{L\eta}{2}) \|\nabla V^{\boldsymbol{\pi}^{(t)}_{\boldsymbol{\theta}}}_{\boldsymbol{\pi}^{(t)}_{\boldsymbol{\theta}}}(\rho)\|^2 \tag{41}
$$

The last equality is due to Equation (9).

Now, telescoping Equation (41) leads to

$$
\eta(1 - \frac{L\eta}{2}) \sum_{t=0}^{T-1} \|\nabla V^{\boldsymbol{\pi}^{(t)}_{\boldsymbol{\theta}}}_{\boldsymbol{\pi}^{(t)}_{\boldsymbol{\theta}}}(\rho)\|^2 \leq \left( V^{\boldsymbol{\pi}_o^\star}_{\boldsymbol{\pi}_o^\star}(\rho) - V^{\boldsymbol{\pi}^0_{\boldsymbol{\theta}}}_{\boldsymbol{\pi}^0_{\boldsymbol{\theta}}}(\rho) \right) - \left( V^{\boldsymbol{\pi}_o^\star}_{\boldsymbol{\pi}_o^\star}(\rho) - V^{\boldsymbol{\pi}^T_{\boldsymbol{\theta}}}_{\boldsymbol{\pi}^T_{\boldsymbol{\theta}}}(\rho) \right) \tag{42}
$$

$$
\leq \left( V^{\boldsymbol{\pi}_o^\star}_{\boldsymbol{\pi}_o^\star}(\rho) - V^{\boldsymbol{\pi}^0_{\boldsymbol{\theta}}}_{\boldsymbol{\pi}^0_{\boldsymbol{\theta}}}(\rho) \right) \tag{43}
$$

Since $\sum_{t=0}^{T-1} \|\nabla V^{\boldsymbol{\pi}^{(t)}_{\boldsymbol{\theta}}}_{\boldsymbol{\pi}^{(t)}_{\boldsymbol{\theta}}}(\rho)\|^2 \geq T \min_{t \in [T-1]} \|\nabla V^{\boldsymbol{\pi}^{(t)}_{\boldsymbol{\theta}}}_{\boldsymbol{\pi}^{(t)}_{\boldsymbol{\theta}}}(\rho)\|^2$, we obtain

$$
\min_{t \in [T-1]} \|\nabla V^{\boldsymbol{\pi}^{(t)}_{\boldsymbol{\theta}}}_{\boldsymbol{\pi}^{(t)}_{\boldsymbol{\theta}}}(\rho)\|^2 \leq \frac{1}{T\eta \left(1 - \frac{L\eta}{2}\right)} \left( V^{\boldsymbol{\pi}_o^\star}_{\boldsymbol{\pi}_o^\star}(\rho) - V^{\boldsymbol{\pi}^0_{\boldsymbol{\theta}}}_{\boldsymbol{\pi}^0_{\boldsymbol{\theta}}}(\rho) \right) \leq \frac{R_{\max}}{T\eta \left(1 - \frac{L\eta}{2}\right)(1-\gamma)} .
$$

The last inequality comes from $V^{\boldsymbol{\pi}_o^\star}_{\boldsymbol{\pi}_o^\star}(\rho) \leq \frac{R_{\max}}{1-\gamma}$ (Assumption 1).

**Step 2:** We derive from Equation (15) that

$$
(V^{\boldsymbol{\pi}_o^\star}_{\boldsymbol{\pi}_o^\star}(\rho) - V^{\boldsymbol{\pi}_{\boldsymbol{\theta}}}_{\boldsymbol{\pi}_{\boldsymbol{\theta}}}(\rho))^2 \leq \left( \sqrt{|\mathcal{S}||\mathcal{A}|} \left\| \frac{d^{\boldsymbol{\pi}_o^\star}_{\boldsymbol{\pi}_{\boldsymbol{\theta}}, \rho}}{d^{\boldsymbol{\pi}_{\boldsymbol{\theta}}}_{\boldsymbol{\pi}_{\boldsymbol{\theta}}, \nu}} \right\|_\infty \|\nabla_{\boldsymbol{\theta}} V^{\boldsymbol{\pi}_{\boldsymbol{\theta}}}_{\boldsymbol{\pi}_{\boldsymbol{\theta}}}(\nu)\|_2 + \frac{2R_{\max}}{1-\gamma} \left( \frac{1}{2} + \frac{\gamma}{1-\gamma} \psi_{\max} \right) \right)^2
$$

$$
\leq 2|\mathcal{S}||\mathcal{A}| \left\| \frac{d^{\boldsymbol{\pi}_o^\star}_{\boldsymbol{\pi}_{\boldsymbol{\theta}}, \rho}}{d^{\boldsymbol{\pi}_{\boldsymbol{\theta}}}_{\boldsymbol{\pi}_{\boldsymbol{\theta}}, \nu}} \right\|_\infty^2 \|\nabla_{\boldsymbol{\theta}} V^{\boldsymbol{\pi}_{\boldsymbol{\theta}}}_{\boldsymbol{\pi}_{\boldsymbol{\theta}}}(\nu)\|_2^2 + \frac{8R_{\max}^2}{(1-\gamma)^2} \left( \frac{1}{2} + \frac{\gamma}{1-\gamma} \psi_{\max} \right)^2 .
$$

Thus, we further get

$$\min_{t\in[T]}(V^{\pi^\star_o}_{\pi_\theta}(\rho) - V^{\pi^{(t)}_\theta}_{\pi^{(t)}_\theta}(\rho))^2 \le 2|\mathcal{S}||\mathcal{A}| \min_{t\in[T]} \left\| \frac{d^{\pi^\star_o}_{\pi^{(t)}_\theta,\rho}}{d^{\pi^{(t)}_\theta}_{\pi^{(t)}_\theta,\nu}} \right\|^2_\infty \|\nabla_\theta V^{\pi^{(t)}_\theta}_{\pi^{(t)}_\theta}(\nu)\|^2_2 + \frac{8R^2_{\max}}{(1-\gamma)^2}\left(\frac{1}{2} + \frac{\gamma}{1-\gamma}\psi_{\max}\right)^2$$

$$\le 2|\mathcal{S}||\mathcal{A}|\mathrm{Cov}^2 \min_{t\in[T]} \|\nabla_\theta V^{\pi^{(t)}_\theta}_{\pi^{(t)}_\theta}(\nu)\|^2_2 + \frac{8R^2_{\max}}{(1-\gamma)^2}\left(\frac{1}{2} + \frac{\gamma}{1-\gamma}\psi_{\max}\right)^2$$

$$\le 2|\mathcal{S}||\mathcal{A}|\mathrm{Cov}^2 \frac{R_{\max}}{T\eta\left(1 - \frac{L\eta}{2}\right)(1-\gamma)} + \frac{8R^2_{\max}}{(1-\gamma)^2}\left(\frac{1}{2} + \frac{\gamma}{1-\gamma}\psi_{\max}\right)^2 .$$

**Step 3:** Now, we set

$$\min_{t\in[T]}(V^{\pi^\star_o}_{\pi_\theta}(\rho) - V^{\pi^{(t)}_\theta}_{\pi^{(t)}_\theta}(\rho))^2 \le 2|\mathcal{S}||\mathcal{A}|\mathrm{Cov}^2 \frac{R_{\max}}{T\eta\left(1 - \frac{L\eta}{2}\right)(1-\gamma)} + \frac{8R^2_{\max}}{(1-\gamma)^2}\left(\frac{1}{2} + \frac{\gamma}{1-\gamma}\psi_{\max}\right)^2$$

$$\le \left( \sqrt{2|\mathcal{S}||\mathcal{A}|\frac{R_{\max}}{T\eta\left(1 - \frac{L\eta}{2}\right)(1-\gamma)}}\mathrm{Cov} + \frac{2\sqrt{2}R_{\max}}{(1-\gamma)}(\frac{1}{2} + \frac{\gamma}{1-\gamma}\psi_{\max}) \right)^2$$

$$\le \left( \epsilon + \frac{\sqrt{2}R_{\max}}{(1-\gamma)}\left(1 + \frac{2\gamma}{1-\gamma}\psi_{\max}\right) \right)^2 ,$$

and solve for $T$ to get

$$T \ge \frac{2|\mathcal{S}||\mathcal{A}|\mathrm{Cov}^2 R_{\max}}{\eta(1 - \frac{L\eta}{2})(1-\gamma)\epsilon^2} \tag{44}$$

Choosing $\eta = \frac{1}{L}$, we get the final expression

$$T \ge \frac{4L|\mathcal{S}||\mathcal{A}|\mathrm{Cov}^2 R_{\max}}{\epsilon^2(1-\gamma)} . \tag{45}$$

for any $\epsilon > 0$ and the smoothness constant $L = \mathcal{O}\left( \max\left\{ \frac{\gamma R_{\max}|\mathcal{A}|}{(1-\gamma)^2}, \frac{\gamma^2}{(1-\gamma)^3} \right\} \right)$.

Hence, we conclude that for $T = \Omega\left( \frac{|\mathcal{S}||\mathcal{A}|}{\epsilon^2} \max\left\{ \frac{\gamma R_{\max}|\mathcal{A}|}{(1-\gamma)^3}, \frac{\gamma^2}{(1-\gamma)^4} \right\} \right)$ and $\psi_{\max} = \mathcal{O}(\frac{1-\gamma}{\gamma})$,

$$\min_{t\in[T]}(V^{\pi^\star_o}_{\pi^\star_o}(\rho) - V^{\pi^{(t)}_\theta}_{\pi^{(t)}_\theta}(\rho)) \le \epsilon + \mathcal{O}\left( \frac{1}{1-\gamma} \right) .$$

$\square$

### G.2 PROOFS FOR ENTROPY-REGULARISED OR SOFT VALUE FUNCTION

**Definition 6.** *The discounted state occupancy measure $d_{\pi'}^{\pi}(s|s_0)$ induced by a policy $\pi$ and an MDP environment defined by $\pi'$ is defined as*

$$d_{\pi'}^{\pi}(s|s_0) \triangleq \sum_{a \in \mathcal{A}} d_{\pi'}^{\pi}(s, a|s_0) = (1 - \gamma) \sum_{a \in \mathcal{A}} \mathbb{E}_{\tau \sim \mathbb{P}_{\pi'}^{\pi}} \Big[ \sum_{t=0}^{\infty} \gamma^t \mathbb{1}\{s_t = s, a_t = a\} \Big].$$

**Lemma 7** (Regularized Performative Policy Difference: Generic Upper Bound)**.** *Under Assumption 1, the sub-optimality gap of a policy $\pi_\theta$ is*

$$\tilde{V}_{\pi_o^\star}^{\pi_o^\star}(s_0) - \tilde{V}_{\pi_\theta}^{\pi_\theta}(s_0) \leq \frac{1}{1 - \gamma} \mathbb{E}_{(s,a) \sim d_{\pi_\theta}^{\pi_o^\star}(\cdot|s_0)} [\tilde{A}_{\pi_\theta}^{\pi_\theta}(s, a)]$$

$$+ \frac{2}{1 - \gamma} \Big( \xi + \frac{\gamma}{1 - \gamma} \psi_{\max}(R_{\max} + \lambda \log |\mathcal{A}|) \Big)$$

$$- \frac{\lambda}{1 - \gamma} \sum_s d_{\pi_\theta}^{\pi_o^\star}(s|s_0) D_{\mathrm{KL}} (\pi_o^\star(\cdot|s) \| \pi_\theta(\cdot|s)) \tag{46}$$

*Proof.* This lemma follows the same sketch as Lemma 2 with an exception in the way the soft rewards are handled. The difference in the soft rewards equals the difference of the original rewards with a lagrange dependent term. This term is the expected KL divergence over the state visitation distribution. Lemma 1 for regularized rewards reduces to,

$$\tilde{V}_{\pi}^{\pi}(s_0) - \tilde{V}_{\pi'}^{\pi'}(s_0) = \frac{1}{1 - \gamma} \mathbb{E}_{(s,a) \sim d_{\pi'}^{\pi}(\cdot|s_0)} [\tilde{A}_{\pi'}^{\pi'}(s, a)]$$

$$+ \frac{1}{1 - \gamma} \mathbb{E}_{(s,a) \sim d_{\pi'}^{\pi}(\cdot|s_0)} \Big( [\tilde{r}_{\pi}(s, a) - \tilde{r}_{\pi'}(s, a)] + \gamma (\mathbf{P}_\pi - \mathbf{P}_{\pi'})^\top \tilde{V}_{\pi}^{\pi}(s_0) \Big). \tag{47}$$

Therefore,

$$\tilde{r}_{\pi_o^\star}(s, a) - \tilde{r}_{\pi_\theta}(s, a) = r_{\pi_o^\star}(s, a) - r_{\pi_\theta}(s, a) + \lambda \Big( \log \pi_\theta(a|s) - \log \pi_o^\star(a|s) \Big)$$

Therefore, we can write (47) in the following way,

$$\tilde{V}_{\pi_o^\star}^{\pi_o^\star}(s_0) - \tilde{V}_{\pi_\theta}^{\pi_\theta}(s_0) = \frac{1}{1 - \gamma} \mathbb{E}_{(s,a) \sim d_{\pi_\theta}^{\pi_o^\star}(\cdot|s_0)} [\tilde{A}_{\pi_\theta}^{\pi_\theta}(s, a)]$$

$$+ \frac{1}{1 - \gamma} \mathbb{E}_{(s,a) \sim d_{\pi_\theta}^{\pi_o^\star}(\cdot|s_0)} \Big( [\tilde{r}_{\pi_o^\star}(s, a) - \tilde{r}_{\pi_\theta}(s, a)] + \gamma (\mathbf{P}_{\pi_o^\star} - \mathbf{P}_{\pi_\theta})^\top \tilde{V}_{\pi_o^\star}^{\pi_o^\star}(s_0) \Big).$$

$$+ \frac{\lambda}{1 - \gamma} \sum_{s,a} [\log \pi_\theta(a|s) - \log \pi_o^\star(a|s)] d_{\pi_\theta}^{\pi_o^\star}(s, a|s_0)$$

$$= \frac{1}{1 - \gamma} \mathbb{E}_{(s,a) \sim d_{\pi_\theta}^{\pi_o^\star}(\cdot|s_0)} [\tilde{A}_{\pi_\theta}^{\pi_\theta}(s, a)]$$

$$+ \frac{1}{1 - \gamma} \mathbb{E}_{(s,a) \sim d_{\pi_\theta}^{\pi_o^\star}(\cdot|s_0)} \Big( [\tilde{r}_{\pi_o^\star}(s, a) - \tilde{r}_{\pi_\theta}(s, a)] + \gamma (\mathbf{P}_{\pi_o^\star} - \mathbf{P}_{\pi_\theta})^\top \tilde{V}_{\pi_o^\star}^{\pi_o^\star}(s_0) \Big)$$

$$+ \frac{\lambda}{1 - \gamma} \sum_{s,a} d_{\pi_\theta}^{\pi_o^\star}(s|s_0) \pi_o^\star(a|s) [\log \pi_\theta(a|s) - \log \pi_o^\star(a|s)]$$

$$\overset{=}{_{(a)}} \frac{1}{1 - \gamma} \mathbb{E}_{(s,a) \sim d_{\pi_\theta}^{\pi_o^\star}(\cdot|s_0)} [\tilde{A}_{\pi_\theta}^{\pi_\theta}(s, a)]$$

$$+ \frac{1}{1 - \gamma} \mathbb{E}_{(s,a) \sim d_{\pi_\theta}^{\pi_o^\star}(\cdot|s_0)} \Big( [\tilde{r}_{\pi_o^\star}(s, a) - \tilde{r}_{\pi_\theta}(s, a)] + \gamma (\mathbf{P}_{\pi_o^\star} - \mathbf{P}_{\pi_\theta})^\top \tilde{V}_{\pi_o^\star}^{\pi_o^\star}(s_0) \Big)$$

$$- \frac{\lambda}{1 - \gamma} \sum_s d_{\pi_\theta}^{\pi_o^\star}(s|s_0) D_{\mathrm{KL}} (\pi_o^\star(\cdot|s) \| \pi_\theta(\cdot|s))$$

$$\underset{\text{Holder's ineq.}}{\leq} \frac{1}{1 - \gamma} \mathbb{E}_{(s,a) \sim d_{\pi_\theta}^{\pi_o^\star}(\cdot|s_0)} [\tilde{A}_{\pi_\theta}^{\pi_\theta}(s, a)]$$

$$+ \frac{1}{1-\gamma} \mathbb{E}_{(s,a)\sim d_{\pi_\theta}^{\pi_o^\star}(\cdot|s_0)} \left( [\tilde{r}_{\pi_o^\star}(s,a) - \tilde{r}_{\pi_\theta}(s,a)] + \gamma \|\mathbf{P}_{\pi_o^\star} - \mathbf{P}_{\pi_\theta}\|_1 \|\tilde{V}_{\pi_o^\star}^{\pi_o^\star}(s_0)\|_\infty \right)$$

$$- \frac{\lambda}{1-\gamma} \sum_s d_{\pi_\theta}^{\pi_o^\star}(s|s_0) D_{\mathrm{KL}}\left(\pi_o^\star(\cdot|s) \parallel \pi_\theta(\cdot|s)\right)$$

$$\underset{(b)}{\leq} \frac{1}{1-\gamma} \mathbb{E}_{(s,a)\sim d_{\pi_\theta}^{\pi_o^\star}(\cdot|s_0)} [\tilde{A}_{\pi_\theta}^{\pi_\theta}(s,a)]$$

$$+ \frac{1}{1-\gamma} \mathbb{E}_{(s,a)\sim d_{\pi_\theta}^{\pi_o^\star}(\cdot|s_0)} \left( [\tilde{r}_{\pi_o^\star}(s,a) - \tilde{r}_{\pi_\theta}(s,a)] + \gamma \|\mathbf{P}_{\pi_o^\star} - \mathbf{P}_{\pi_\theta}\|_1 \frac{R_{\max} + \lambda \log|\mathcal{A}|}{1-\gamma} \right)$$

$$- \frac{\lambda}{1-\gamma} \sum_s d_{\pi_\theta}^{\pi_o^\star}(s|s_0) D_{\mathrm{KL}}\left(\pi_o^\star(\cdot|s) \parallel \pi_\theta(\cdot|s)\right)$$

$$\underset{\text{Lipschitz } r \,\&\, \mathbf{P}}{\leq} \frac{1}{1-\gamma} \mathbb{E}_{(s,a)\sim d_{\pi_\theta}^{\pi_o^\star}(\cdot|s_0)} [\tilde{A}_{\pi_\theta}^{\pi_\theta}(s,a)]$$

$$+ \frac{1}{1-\gamma} \mathbb{E}_{(s,a)\sim d_{\pi_\theta}^{\pi_o^\star}(\cdot|s_0)} \left( L_r + L_\mathbf{P} \frac{\gamma(R_{\max} + \lambda \log|\mathcal{A}|)}{1-\gamma} \right) \|\pi_o^\star - \pi_\theta\|_1$$

$$- \frac{\lambda}{1-\gamma} \sum_s d_{\pi_\theta}^{\pi_o^\star}(s|s_0) D_{\mathrm{KL}}\left(\pi_o^\star(\cdot|s) \parallel \pi_\theta(\cdot|s)\right)$$

$$\underset{(c)}{\leq} \frac{1}{1-\gamma} \mathbb{E}_{(s,a)\sim d_{\pi_\theta}^{\pi_o^\star}(\cdot|s_0)} [\tilde{A}_{\pi_\theta}^{\pi_\theta}(s,a)]$$

$$+ \frac{2}{1-\gamma} \mathbb{E}_{(s,a)\sim d_{\pi_\theta}^{\pi_o^\star}(\cdot|s_0)} \left( L_r + L_\mathbf{P} \frac{\gamma(R_{\max} + \lambda \log|\mathcal{A}|)}{1-\gamma} \right)$$

$$- \frac{\lambda}{1-\gamma} \sum_s d_{\pi_\theta}^{\pi_o^\star}(s|s_0) D_{\mathrm{KL}}\left(\pi_o^\star(\cdot|s) \parallel \pi_\theta(\cdot|s)\right)$$

The equality (a) holds since,

$$\mathbb{E}_{a\sim\pi_o^\star(\cdot|s)}[\log \pi_\theta(a|s) - \log \pi_o^\star(a|s)] = -D_{\mathrm{KL}}\left(\pi_o^\star(\cdot|s) \parallel \pi_\theta(\cdot|s)\right)$$

The inequality (b) holds due to the result of Mei et al. (2020), i.e.

$$\|\tilde{V}_{\pi_o^\star}^{\pi_o^\star}\|_\infty \leq \frac{R_{\max} + \lambda \log|\mathcal{A}|}{1-\gamma} \tag{48}$$

Finally, (c) is due to the fact that $\|\pi_o^\star - \pi_\theta\|_1 \leq 2$.

$\square$

**Lemma 8** (Regularized Performative Policy gradient for softmax policies and softmax MDPs)**.** *For a class of PeMDPs $\mathcal{M} \triangleq (\mathcal{S}, \mathcal{A}, \pi, \mathbf{P}_\pi, r_\pi, \theta, \rho)$ consider softmax parametrization for policy $\pi_\theta \in \Delta (\theta \in \Theta)$ and transition dynamics $\mathbf{P}_{\pi_\theta}$ and linear parametrization for reward $r_{\pi_\theta}$. For all $(s, a, s') \in (\mathcal{S}, \mathcal{A}, \mathcal{S})$, derivative of the expected return w.r.t $\theta_{s,a}$ satisfies:*

$$\frac{\partial \tilde{V}_{\pi_\theta}^{\pi_\theta}(\rho)}{\partial \theta_{s,a}} \geq \frac{1}{1-\gamma} d_{\pi_\theta}^{\pi_\theta}(s,a|\rho) \left( \tilde{A}_{\pi_\theta}^{\pi_\theta}(s,a) + \xi \right) - \frac{\lambda}{1-\gamma}(1 + \log|\mathcal{A}|). \tag{49}$$

*Proof.* This proof follows the same sketch as the proof of Theorem 3. However, we get two additional $\lambda$-dependent terms– (a) one from the log policy term in the soft advantage, and (b) the other from the log policy term in the soft rewards. We then simplify these terms to obtain the final expression.

First, let us note that

$$\frac{\partial}{\partial \theta_{s',a'}} \log \pi_\theta(a|s) = \mathbb{1}[s = s', a = a'] - \pi_\theta(a'|s)\mathbb{1}[s = s']$$

$$\frac{\partial}{\partial \theta_{s',a'}} \log \mathbf{P}_{\pi_\theta}(s''|s,a) = \psi(s'')\mathbb{1}[s = s', a = a'] (1 - \mathbf{P}_{\pi_\theta}(s''|s,a))$$

$$\frac{\partial}{\partial \theta_{s',a'}} r_{\pi_\theta}(s,a) = \xi \mathbb{1}[s = s', a = a']. \tag{50}$$

Now, we get from Theorem 2,

$$\frac{\partial}{\partial \boldsymbol{\theta}_{s,a}} \tilde{V}^{\boldsymbol{\pi}_{\boldsymbol{\theta}}}_{\boldsymbol{\pi}_{\boldsymbol{\theta}}}(\boldsymbol{\rho}) = \mathbb{E}_{\tau \sim \mathbb{P}^{\boldsymbol{\pi}_{\boldsymbol{\theta}}}_{\boldsymbol{\pi}_{\boldsymbol{\theta}}}} \Big[ \sum_{t=0}^{\infty} \gamma^t \Big( \tilde{A}^{\boldsymbol{\pi}_{\boldsymbol{\theta}}}_{\boldsymbol{\pi}_{\boldsymbol{\theta}}}(s_t, a_t) \frac{\partial}{\partial \boldsymbol{\theta}_{s,a}} \log \boldsymbol{\pi}_{\boldsymbol{\theta}}(a_t \mid s_t)$$

$$+ \tilde{A}^{\boldsymbol{\pi}_{\boldsymbol{\theta}}}_{\boldsymbol{\pi}_{\boldsymbol{\theta}}}(s_t, a_t) \frac{\partial}{\partial \boldsymbol{\theta}_{s,a}} \log P_{\boldsymbol{\pi}_{\boldsymbol{\theta}}}(s_{t+1}|s_t, a_t)$$

$$+ \frac{\partial}{\partial \boldsymbol{\theta}_{s,a}} r_{\boldsymbol{\pi}_{\boldsymbol{\theta}}}(s_t, a_t) - \lambda \frac{\partial}{\partial \boldsymbol{\theta}_{s,a}} \log \boldsymbol{\pi}_{\boldsymbol{\theta}}(a_t \mid s_t) \Big) \Big]$$

$$= \mathbb{E}_{\tau \sim \mathbb{P}^{\boldsymbol{\pi}_{\boldsymbol{\theta}}}_{\boldsymbol{\pi}_{\boldsymbol{\theta}}}} \Big[ \sum_{t=0}^{\infty} \gamma^t \Big( \tilde{A}^{\boldsymbol{\pi}_{\boldsymbol{\theta}}}_{\boldsymbol{\pi}_{\boldsymbol{\theta}}}(s_t, a_t) \left( \mathbb{1}[s_t = s, a_t = a] - \boldsymbol{\pi}_{\boldsymbol{\theta}}(a|s)\mathbb{1}[s_t = s] \right)$$

$$+ \tilde{A}^{\boldsymbol{\pi}_{\boldsymbol{\theta}}}_{\boldsymbol{\pi}_{\boldsymbol{\theta}}}(s_t, a_t) \psi(s_{t+1}) \mathbb{1}[s_t = s, a_t = a] \left( 1 - \mathbf{P}_{\boldsymbol{\pi}_{\boldsymbol{\theta}}}(s_{t+1}|s, a) \right)$$

$$+ \xi \mathbb{1}[s_t = s, a_t = a] - \lambda \mathbb{1}[s_t = s, a_t = a] + \lambda \boldsymbol{\pi}_{\boldsymbol{\theta}}(a|s)\mathbb{1}[s_t = s] \Big) \Big]$$

$$\underset{(a)}{\geq} \mathbb{E}_{\tau \sim \mathbb{P}^{\boldsymbol{\pi}_{\boldsymbol{\theta}}}_{\boldsymbol{\pi}_{\boldsymbol{\theta}}}} \Big[ \sum_{t=0}^{\infty} \gamma^t \tilde{A}^{\boldsymbol{\pi}_{\boldsymbol{\theta}}}_{\boldsymbol{\pi}_{\boldsymbol{\theta}}}(s_t, a_t) \mathbb{1}[s_t = s, a_t = a] \Big] - \mathbb{E}_{\tau \sim \mathbb{P}^{\boldsymbol{\pi}_{\boldsymbol{\theta}}}_{\boldsymbol{\pi}_{\boldsymbol{\theta}}}} \Big[ \sum_{t=0}^{\infty} \gamma^t \boldsymbol{\pi}_{\boldsymbol{\theta}}(a|s)\mathbb{1}[s_t = s] \tilde{A}^{\boldsymbol{\pi}_{\boldsymbol{\theta}}}_{\boldsymbol{\pi}_{\boldsymbol{\theta}}}(s_t, a_t) \Big]$$

$$+ \mathbb{E}_{\tau \sim \mathbb{P}^{\boldsymbol{\pi}_{\boldsymbol{\theta}}}_{\boldsymbol{\pi}_{\boldsymbol{\theta}}}} \Big[ \sum_{t=0}^{\infty} \gamma^t \xi \mathbb{1}[s_t = s, a_t = a] \Big] - \lambda \mathbb{E}_{\tau \sim \mathbb{P}^{\boldsymbol{\pi}_{\boldsymbol{\theta}}}_{\boldsymbol{\pi}_{\boldsymbol{\theta}}}} \Big[ \sum_{t=0}^{\infty} \gamma^t \mathbb{1}[s_t = s, a_t = a] \Big] + \lambda \mathbb{E}_{\tau \sim \mathbb{P}^{\boldsymbol{\pi}_{\boldsymbol{\theta}}}_{\boldsymbol{\pi}_{\boldsymbol{\theta}}}} \Big[ \sum_{t=0}^{\infty} \gamma^t \boldsymbol{\pi}_{\boldsymbol{\theta}}(a_t|s_t)\mathbb{1}[s_t = s] \Big]$$

$$= \frac{1}{1-\gamma} d^{\boldsymbol{\pi}_{\boldsymbol{\theta}}}_{\boldsymbol{\pi}_{\boldsymbol{\theta}},\boldsymbol{\rho}}(s, a) \tilde{A}^{\boldsymbol{\pi}_{\boldsymbol{\theta}}}_{\boldsymbol{\pi}_{\boldsymbol{\theta}}}(s, a) + \lambda \mathbb{E}_{\tau \sim \mathbb{P}^{\boldsymbol{\pi}_{\boldsymbol{\theta}}}_{\boldsymbol{\pi}_{\boldsymbol{\theta}}}} \Big[ \sum_{t=0}^{\infty} \gamma^t \boldsymbol{\pi}_{\boldsymbol{\theta}}(a_t|s_t) \log \boldsymbol{\pi}_{\boldsymbol{\theta}}(a_t|s_t)\mathbb{1}[s_t = s] \Big]$$

$$+ \frac{1}{1-\gamma} \xi d^{\boldsymbol{\pi}_{\boldsymbol{\theta}}}_{\boldsymbol{\pi}_{\boldsymbol{\theta}},\boldsymbol{\rho}}(s, a) - \frac{\lambda}{1-\gamma} d^{\boldsymbol{\pi}_{\boldsymbol{\theta}}}_{\boldsymbol{\pi}_{\boldsymbol{\theta}},\boldsymbol{\rho}}(s, a|s_0) + \lambda \mathbb{E}_{\tau \sim \mathbb{P}^{\boldsymbol{\pi}_{\boldsymbol{\theta}}}_{\boldsymbol{\pi}_{\boldsymbol{\theta}}}} \Big[ \sum_{t=0}^{\infty} \gamma^t \boldsymbol{\pi}_{\boldsymbol{\theta}}(a_t|s_t) \sum_a \mathbb{1}[s_t = s, a_t = a] \Big]$$

$$= \frac{1}{1-\gamma} d^{\boldsymbol{\pi}_{\boldsymbol{\theta}}}_{\boldsymbol{\pi}_{\boldsymbol{\theta}},\boldsymbol{\rho}}(s, a) \tilde{A}^{\boldsymbol{\pi}_{\boldsymbol{\theta}}}_{\boldsymbol{\pi}_{\boldsymbol{\theta}}}(s, a) + \lambda \mathbb{E}_{\tau \sim \mathbb{P}^{\boldsymbol{\pi}_{\boldsymbol{\theta}}}_{\boldsymbol{\pi}_{\boldsymbol{\theta}}}} \Big[ \sum_{t=0}^{\infty} \gamma^t \boldsymbol{\pi}_{\boldsymbol{\theta}}(a_t|s_t) \log \boldsymbol{\pi}_{\boldsymbol{\theta}}(a_t|s_t) \sum_a \mathbb{1}[s_t = s, a_t = a] \Big]$$

$$+ \frac{1}{1-\gamma} \xi d^{\boldsymbol{\pi}_{\boldsymbol{\theta}}}_{\boldsymbol{\pi}_{\boldsymbol{\theta}},\boldsymbol{\rho}}(s, a) - \frac{\lambda}{1-\gamma} d^{\boldsymbol{\pi}_{\boldsymbol{\theta}}}_{\boldsymbol{\pi}_{\boldsymbol{\theta}},\boldsymbol{\rho}}(s, a) + \lambda \mathbb{E}_{\tau \sim \mathbb{P}^{\boldsymbol{\pi}_{\boldsymbol{\theta}}}_{\boldsymbol{\pi}_{\boldsymbol{\theta}}}} \Big[ \sum_a \boldsymbol{\pi}_{\boldsymbol{\theta}}(a|s) \sum_{t=0}^{\infty} \gamma^t \mathbb{1}[s_t = s, a_t = a] \Big]$$

$$= \frac{1}{1-\gamma} d^{\boldsymbol{\pi}_{\boldsymbol{\theta}}}_{\boldsymbol{\pi}_{\boldsymbol{\theta}},\boldsymbol{\rho}}(s, a) \tilde{A}^{\boldsymbol{\pi}_{\boldsymbol{\theta}}}_{\boldsymbol{\pi}_{\boldsymbol{\theta}}}(s, a) + \lambda \mathbb{E}_{\tau \sim \mathbb{P}^{\boldsymbol{\pi}_{\boldsymbol{\theta}}}_{\boldsymbol{\pi}_{\boldsymbol{\theta}}}} \Big[ \sum_a \boldsymbol{\pi}_{\boldsymbol{\theta}}(a|s) \log \boldsymbol{\pi}_{\boldsymbol{\theta}}(a|s) \sum_{t=0}^{\infty} \gamma^t \mathbb{1}[s_t = s, a_t = a] \Big]$$

$$+ \frac{1}{1-\gamma} \xi d^{\boldsymbol{\pi}_{\boldsymbol{\theta}}}_{\boldsymbol{\pi}_{\boldsymbol{\theta}},\boldsymbol{\rho}}(s, a) - \frac{\lambda}{1-\gamma} d^{\boldsymbol{\pi}_{\boldsymbol{\theta}}}_{\boldsymbol{\pi}_{\boldsymbol{\theta}},\boldsymbol{\rho}}(s, a) + \frac{\lambda}{1-\gamma} \sum_a d^{\boldsymbol{\pi}_{\boldsymbol{\theta}}}_{\boldsymbol{\pi}_{\boldsymbol{\theta}},\boldsymbol{\rho}}(s, a) \boldsymbol{\pi}_{\boldsymbol{\theta}}(a|s)$$

$$\geq \frac{1}{1-\gamma} d^{\boldsymbol{\pi}_{\boldsymbol{\theta}}}_{\boldsymbol{\pi}_{\boldsymbol{\theta}},\boldsymbol{\rho}}(s, a) \left( \tilde{A}^{\boldsymbol{\pi}_{\boldsymbol{\theta}}}_{\boldsymbol{\pi}_{\boldsymbol{\theta}}}(s, a) + \xi \right) - \frac{\lambda}{1-\gamma} d^{\boldsymbol{\pi}_{\boldsymbol{\theta}}}_{\boldsymbol{\pi}_{\boldsymbol{\theta}},\boldsymbol{\rho}}(s, a) - \frac{\lambda}{1-\gamma} \sum_a d^{\boldsymbol{\pi}_{\boldsymbol{\theta}}}_{\boldsymbol{\pi}_{\boldsymbol{\theta}},\boldsymbol{\rho}}(s, a) \boldsymbol{\pi}_{\boldsymbol{\theta}}(a|s) \log \frac{1}{\boldsymbol{\pi}_{\boldsymbol{\theta}}(a|s)}$$

$$\underset{(b)}{\geq} \frac{1}{1-\gamma} d^{\boldsymbol{\pi}_{\boldsymbol{\theta}}}_{\boldsymbol{\pi}_{\boldsymbol{\theta}},\boldsymbol{\rho}}(s, a) \left( \tilde{A}^{\boldsymbol{\pi}_{\boldsymbol{\theta}}}_{\boldsymbol{\pi}_{\boldsymbol{\theta}}}(s, a) + \xi \right) - \frac{\lambda}{1-\gamma} d^{\boldsymbol{\pi}_{\boldsymbol{\theta}}}_{\boldsymbol{\pi}_{\boldsymbol{\theta}},\boldsymbol{\rho}}(s, a)(1 + \log |\mathcal{A}|) .$$

(b) holds from the following:

$$- \sum_a d^{\boldsymbol{\pi}_{\boldsymbol{\theta}}}_{\boldsymbol{\pi}_{\boldsymbol{\theta}}}(s, a|s_0) \log \boldsymbol{\pi}_{\boldsymbol{\theta}}(a|s) = d^{\boldsymbol{\pi}_{\boldsymbol{\theta}}}_{\boldsymbol{\pi}_{\boldsymbol{\theta}}}(s|s_0) \Big( - \sum_a \boldsymbol{\pi}_{\boldsymbol{\theta}}(a|s) \log \boldsymbol{\pi}_{\boldsymbol{\theta}}(a|s) \Big)$$

$$\underset{(c)}{\leq} d^{\boldsymbol{\pi}_{\boldsymbol{\theta}}}_{\boldsymbol{\pi}_{\boldsymbol{\theta}}}(s|s_0) \log |\mathcal{A}| \leq \log |\mathcal{A}|$$

and (c) holds as entropy is upper bounded by $\log |\mathcal{A}|$ (Cover & Thomas, 2006, Theorem 2.6.4).

$\square$

**Lemma 3** (Regularized Performative Gradient Domination: Part(b) of Lemma 3)**.** *For regularized PeMDPs the following inequality holds:*

$$\tilde{V}^{\boldsymbol{\pi}^\star_o}_{\boldsymbol{\pi}^\star_o}(\rho) - \tilde{V}^{\boldsymbol{\pi}_{\boldsymbol{\theta}}}_{\boldsymbol{\pi}_{\boldsymbol{\theta}}}(\rho)$$

$$\leq \sqrt{|\mathcal{S}||\mathcal{A}|}\left\|\frac{d_{\pi_\theta,\rho}^{\pi_o^\star}}{d_{\pi_\theta,\nu}^{\pi_\theta}}\right\|_\infty \|\nabla_\theta \tilde{V}_{\pi_\theta}^{\pi_\theta}(\nu)\|_2 + \frac{R_{\max}}{1-\gamma}\left(1 + \frac{2\gamma}{1-\gamma}\psi_{\max}\left(1 + \frac{\lambda}{R_{\max}}\log|\mathcal{A}|\right)\right) + \frac{\lambda}{1-\gamma}(1+\log|\mathcal{A}|). \quad (51)$$

*Proof.* **Step 1.** First, we observe that

$$-D_{\mathrm{KL}}\left(\pi_o^\star(\cdot|s) \,\|\, \pi_\theta(\cdot|s)\right) \leq -\sum_{a \in \mathcal{A}} \pi_o^\star(a|s)\log\pi_o^\star(a|s) \leq \log|\mathcal{A}|$$

Hence, we get

$$-\sum_s d_{\pi_\theta}^{\pi_o^\star}(s|s_0)D_{\mathrm{KL}}\left(\pi_o^\star(\cdot|s) \,\|\, \pi_\theta(\cdot|s)\right) \leq \log|\mathcal{A}| \quad (52)$$

**Step 2.** Using Lemma 8 and applying Cauchy-Schwarz inequality, we get

$$\sum_{s,a} d_{\pi_\theta}^{\pi_\theta}(s,a)\tilde{A}_{\pi_\theta}^{\pi_\theta}(s,a) \leq \sqrt{|\mathcal{S}||\mathcal{A}|}(1-\gamma)\|\nabla_\theta \tilde{V}_{\pi_\theta}^{\pi_\theta}(\nu)\|_2 - \xi + \lambda(\log|\mathcal{A}|+1) \quad (53)$$

**Step 3.** Now, substituting Equation (52) and (53) in Equation (46), we finally get

$$\tilde{V}_{\pi_o^\star}^{\pi_o^\star}(\rho) - \tilde{V}_{\pi_\theta}^{\pi_\theta}(\rho) \leq \sqrt{|\mathcal{S}||\mathcal{A}|}\left\|\frac{d_{\pi_\theta,\rho}^{\pi_o^\star}}{d_{\pi_\theta,\nu}^{\pi_\theta}}\right\|_\infty \|\nabla_\theta \tilde{V}_{\pi_\theta}^{\pi_\theta}(\nu)\|_2 - \left\|\frac{d_{\pi_\theta,\rho}^{\pi_o^\star}}{d_{\pi_\theta,\nu}^{\pi_\theta}}\right\|_\infty \frac{\xi}{1-\gamma} + \frac{\lambda}{1-\gamma}(\log|\mathcal{A}|+1)$$

$$+ \frac{2}{1-\gamma}\left(L_r + \frac{\gamma}{1-\gamma}L_{\mathbf{P}}(R_{\max} + \lambda\log|\mathcal{A}|)\right)$$

$$\overset{(a)}{=} \sqrt{|\mathcal{S}||\mathcal{A}|}\left\|\frac{d_{\pi_\theta,\rho}^{\pi_o^\star}}{d_{\pi_\theta,\nu}^{\pi_\theta}}\right\|_\infty \|\nabla_\theta \tilde{V}_{\pi_\theta}^{\pi_\theta}(\nu)\|_2 - \left\|\frac{d_{\pi_\theta,\rho}^{\pi_o^\star}}{d_{\pi_\theta,\nu}^{\pi_\theta}}\right\|_\infty \frac{\xi}{1-\gamma} + \frac{\lambda}{1-\gamma}(1+\log|\mathcal{A}|)$$

$$+ \frac{2}{1-\gamma}\left(\xi + \frac{\gamma}{1-\gamma}\psi_{\max}(R_{\max} + \lambda\log|\mathcal{A}|)\right)$$

$$\overset{(b)}{\leq} \sqrt{|\mathcal{S}||\mathcal{A}|}\left\|\frac{d_{\pi_\theta,\rho}^{\pi_o^\star}}{d_{\pi_\theta,\nu}^{\pi_\theta}}\right\|_\infty \|\nabla_\theta \tilde{V}_{\pi_\theta}^{\pi_\theta}(\nu)\|_2 + \frac{\xi}{1-\gamma}$$

$$+ \frac{2\gamma}{1-\gamma}\psi_{\max}(R_{\max} + \lambda\log|\mathcal{A}|) + \frac{\lambda}{1-\gamma}(1+\log|\mathcal{A}|)$$

$$= \sqrt{|\mathcal{S}||\mathcal{A}|}\left\|\frac{d_{\pi_\theta,\rho}^{\pi_o^\star}}{d_{\pi_\theta,\nu}^{\pi_\theta}}\right\|_\infty \|\nabla_\theta \tilde{V}_{\pi_\theta}^{\pi_\theta}(\nu)\|_2$$

$$+ \frac{2}{1-\gamma}\left(\frac{\xi}{2} + \frac{\gamma}{1-\gamma}\psi_{\max}(R_{\max} + \lambda\log|\mathcal{A}|)\right) + \frac{\lambda}{1-\gamma}(1+\log|\mathcal{A}|)$$

$$\leq \sqrt{|\mathcal{S}||\mathcal{A}|}\left\|\frac{d_{\pi_\theta,\rho}^{\pi_o^\star}}{d_{\pi_\theta,\nu}^{\pi_\theta}}\right\|_\infty \|\nabla_\theta \tilde{V}_{\pi_\theta}^{\pi_\theta}(\nu)\|_2$$

$$+ \frac{R_{\max}}{1-\gamma}\left(1 + \frac{2\gamma}{1-\gamma}\psi_{\max}\left(1 + \frac{\lambda}{R_{\max}}\log|\mathcal{A}|\right)\right) + \frac{\lambda}{1-\gamma}(1+\log|\mathcal{A}|)$$

In (a), we substitute the values of $L_r$ and $L_{\mathbf{P}}$ for softmax PeMDPs, and in (b), we use $\left\|\frac{d_{\pi_\theta,\rho}^{\pi_o^\star}}{d_{\pi_\theta,\nu}^{\pi_\theta}}\right\|_\infty \geq 1$ (Lemma 9).

$\square$

**Theorem 3** (Convergence of PePG in softmax PeMDPs – Part (b)). *Let* $\mathrm{Cov} \triangleq \max_{\theta,\nu}\left\|\frac{d_{\pi_\theta,\rho}^{\pi_o^\star}}{d_{\pi_\theta,\nu}^{\pi_\theta}}\right\|_\infty$. *The gradient ascent algorithm on* $V_{\pi_\theta}^{\pi_\theta}(\rho)$ *(Equation (9)) with step size* $\eta = \Omega\left(\frac{(1-\gamma)^2}{\gamma|\mathcal{A}|}\right)$ *satisfies, for all distributions* $\rho \in \Delta(\mathcal{S})$.

*(b) For entropy regularised case, if we set* $\lambda = \frac{(1-\gamma)R_{\max}}{1+\log|\mathcal{A}|}$, *we get*

$$\min_{t<T}\left\{\tilde{V}_{\pi_o^\star}^{\pi_o^\star}(\rho) - \tilde{V}_{\pi_\theta^{(t)}}^{\pi_\theta^{(t)}}(\rho)\right\} \leq \epsilon \text{ when } T = \Omega\left(\frac{R_{\max}|\mathcal{S}||\mathcal{A}|^2}{\epsilon^2(1-\gamma)^3}\mathrm{Cov}^2\right), \text{ and } \epsilon = \Omega\left(\frac{1}{1-\gamma}\right).$$

*Proof.* This proof follows similar steps as part (a) of Theorem 3 with two additional changes: (i) We have a $\lambda$, i.e. regularisation coefficient, dependent term due to the entropy regulariser. (ii) The maximum value of the soft value function is $\frac{R_{\max} + \lambda \log |\mathcal{A}|}{1 - \gamma}$ instead of $\frac{R_{\max}}{1 - \gamma}$ for the unregularised value function.

**Step 1:** From Equation (31), we observe that the soft-value function $\tilde{V}^{\pi_\theta}_{\pi_\theta}$ is $L_\lambda$-smooth.

Thus, following the Step 1 of Theorem 3, we get

$$
\min_{t \in [T-1]} \|\nabla \tilde{V}^{\pi^{(t)}_\theta}_{\pi^{(t)}_\theta}(\rho)\|^2 \leq \frac{1}{T\eta \left(1 - \frac{L_\lambda \eta}{2}\right)} \left( \tilde{V}^{\pi^\star_o}_{\pi^\star_o}(\rho) - \tilde{V}^{\pi^0_\theta}_{\pi^0_\theta}(\rho) \right)
$$

$$
\leq \frac{R_{\max} + \lambda \log |\mathcal{A}|}{T\eta \left(1 - \frac{L_\lambda \eta}{2}\right)(1 - \gamma)}.
\tag{54}
$$

The last inequality is true due to the fact that $\tilde{V}^{\pi^\star_o}_{\pi^\star_o}(\rho) - \tilde{V}^{\pi^{(0)}_\theta}_{\pi^{(0)}_\theta}(\rho) \leq \tilde{V}^{\pi^\star_o}_{\pi^\star_o}(\rho) \leq \frac{R_{\max} + \lambda \log |\mathcal{A}|}{1 - \gamma}$.

**Step 2:** Now, from Part (b) of Lemma 3, we obtain that

$$
\min_{t \in [T-1]} \left( \tilde{V}^{\pi^\star_o}_{\pi^\star_o}(\rho) - \tilde{V}^{\pi^{(t)}_\theta}_{\pi^{(t)}_\theta}(\rho) \right)^2
$$

$$
\leq \min_{t \in [T-1]} \left( \sqrt{|\mathcal{S}||\mathcal{A}|} \left\| \frac{d^{\pi^\star_o}_{\pi^{(t)}_\theta, \rho}}{d^{\pi^{(t)}_\theta}_{\pi^{(t)}_\theta, \nu}} \right\|_\infty \|\nabla_\theta \tilde{V}^{\pi^{(t)}_\theta}_{\pi^{(t)}_\theta}(\nu)\|_2 + \frac{R_{\max}}{1 - \gamma} \left( 1 + \frac{2\gamma}{1-\gamma} \psi_{\max} \left( 1 + \frac{\lambda}{R_{\max}} \log |\mathcal{A}| \right) \right) + \frac{\lambda}{1 - \gamma}(1 + \log |\mathcal{A}|) \right)^2
$$

$$
\leq 2|\mathcal{S}||\mathcal{A}| \min_{t \in [T-1]} \left\| \frac{d^{\pi^\star_o}_{\pi^{(t)}_\theta, \rho}}{d^{\pi^{(t)}_\theta}_{\pi^{(t)}_\theta, \nu}} \right\|_\infty^2 \|\nabla_\theta \tilde{V}^{\pi^{(t)}_\theta}_{\pi^{(t)}_\theta}(\nu)\|_2^2 + 2 \left( \frac{R_{\max}}{1 - \gamma} \left( 1 + \frac{2\gamma}{1-\gamma} \psi_{\max} \left( 1 + \frac{\lambda}{R_{\max}} \log |\mathcal{A}| \right) \right) + \frac{\lambda}{1 - \gamma}(1 + \log |\mathcal{A}|) \right)^2
$$

$$
\leq \frac{2|\mathcal{S}||\mathcal{A}| \mathrm{Cov}^2 (R_{\max} + \lambda \log |\mathcal{A}|)}{T\eta \left(1 - \frac{L_\lambda \eta}{2}\right)(1 - \gamma)} + 2 \left( \frac{R_{\max}}{1 - \gamma} \left( 1 + \frac{2\gamma}{1-\gamma} \psi_{\max} \left( 1 + \frac{\lambda}{R_{\max}} \log |\mathcal{A}| \right) \right) + \frac{\lambda}{1 - \gamma}(1 + \log |\mathcal{A}|) \right)^2.
$$

The last inequality is due to the upper bound on the minimum gradient norm as in Equation (54) and by definition of the coverage parameter $\mathrm{Cov}$.

Thus, we conclude that

$$
\min_{t \in [T-1]} \tilde{V}^{\pi^\star_o}_{\pi^\star_o}(\rho) - \tilde{V}^{\pi^{(t)}_\theta}_{\pi^{(t)}_\theta}(\rho)
$$

$$
\leq \sqrt{\frac{2|\mathcal{S}||\mathcal{A}| \mathrm{Cov}^2 (R_{\max} + \lambda \log |\mathcal{A}|)}{T\eta \left(1 - \frac{L_\lambda \eta}{2}\right)(1 - \gamma)}} + \sqrt{2} \left( \frac{R_{\max}}{1 - \gamma} \left( 1 + \frac{2\gamma}{1-\gamma} \psi_{\max} \left( 1 + \frac{\lambda}{R_{\max}} \log |\mathcal{A}| \right) \right) + \frac{\lambda}{1 - \gamma}(1 + \log |\mathcal{A}|) \right).
$$

$$
\tag{55}
$$

**Step 4:** Now, by setting the $T$-dependent term in Equation (55) to $\epsilon$, we get $T \geq \frac{2|\mathcal{S}||\mathcal{A}| \mathrm{Cov}^2 (R_{\max} + \lambda \log |\mathcal{A}|)}{\eta \left(1 - \frac{L_\lambda \eta}{2}\right)(1 - \gamma)\epsilon^2}$.

Choosing $\eta = \frac{1}{L_\lambda}$, $\lambda = \frac{(1-\gamma)R_{\max}}{(1 + \log |\mathcal{A}|)}$, and $\psi_{\max} = \mathcal{O}(\frac{1-\gamma}{\gamma})$, we get the final expression $T \geq \frac{8|\mathcal{S}||\mathcal{A}| \mathrm{Cov}^2 L_\lambda R_{\max}}{(1-\gamma)\epsilon^2}$, and

$$
\min_{t \in [T-1]} \tilde{V}^{\pi^\star_o}_{\pi^\star_o}(\rho) - \tilde{V}^{\pi^{(t)}_\theta}_{\pi^{(t)}_\theta}(\rho) \leq \epsilon + \mathcal{O}(\frac{1}{1 - \gamma}).
$$

Finally, noting that $L_\lambda = \mathcal{O} \left( \max \left\{ \frac{\gamma R_{\max} |\mathcal{A}| \psi_{\max}^2}{(1-\gamma)^2}, \frac{R_{\max} \psi_{\max}^2}{(1-\gamma)^2} \right\} \right)$, we get

$$
T = \Omega \left( \frac{|\mathcal{S}||\mathcal{A}|}{\epsilon^2 (1 - \gamma)^3} \max\{1, \gamma|\mathcal{A}|\} \right).
$$

$\square$

# H   ABLATION STUDY ON ENTROPY REGULARISATION

Figure 3: Ablation study for PePG  for different values of regularised $\lambda$ with 20 random seeds, each for 100 iterations

We conducted an ablation study across four entropy regularization strengths ($\lambda \in \{0.01, 0.5, 1, 2\}$ to determine the optimale balance between exploration and convergence stability in RegPePG. The results demonstrate that $\lambda = 2$ achieves the highest final performance ( 0.05), while smaller values ($\lambda \leq 1$) converge to similar suboptimal levels around $-0.01$ to 0, indicating that stronger entropy regularization enables more effective exploration of the policy space in performative settings.

## I    TECHNICAL LEMMAS

**Lemma 9** (Lower Bound of Coverage). *For any $\boldsymbol{\pi}, \boldsymbol{\pi}' \in \Pi(\Theta)$, the following non-trivial lower bound holds,*

$$\left\| \frac{\boldsymbol{d_{\pi'}}}{\boldsymbol{d_\pi}} \right\|_\infty \geq 1$$

*Proof.*

$$\left\| \frac{\boldsymbol{d_{\pi'}}}{\boldsymbol{d_\pi}} \right\|_\infty = \max_{s,a} \frac{\boldsymbol{d_{\pi'}}(s,a)}{\boldsymbol{d_\pi}(s,a)} \geq \frac{1}{\sum_{s,a} w_{s,a}} \sum_{s,a} \frac{\boldsymbol{d_{\pi'}}(s,a)}{\boldsymbol{d_\pi}(s,a)} \cdot w_{s,a}$$

Choose $w_{s,a} = \boldsymbol{d_\pi}(s,a)$ Hence, we get,

$$\max_{s,a} \frac{\boldsymbol{d_{\pi'}}(s,a)}{\boldsymbol{d_\pi}(s,a)} \geq \frac{\sum_{s,a} \boldsymbol{d_{\pi'}}(s,a)}{\sum_{s,a} \boldsymbol{d_\pi}(s,a)} = 1$$

The last equality holds from the fact that the state-action occupancy measure is a distribution over $\mathcal{S} \times \mathcal{A}$. Hence, $\sum_{s,a} \boldsymbol{d_{\pi'}}(s,a) = \sum_{s,a} \boldsymbol{d_\pi}(s,a)$ $\qquad\square$

**Lemma 10.** *The discounted state occupancy measure*

$$\boldsymbol{d_{\pi'}^\pi}(s|s_0) \triangleq (1-\gamma) \, \mathbb{E}_{\tau \sim \mathbb{P}_{\pi'}^\pi} \left[ \sum_{t=0}^\infty \gamma^t \mathbb{1}\{s_t = s\} \right]$$

*is a probability mass function over the state-space $\mathcal{S}$.*

*Proof.* For each fixed $s$ the integrand $\sum_{t=0}^\infty \gamma^t \mathbb{1}\{s_t = s\} \geq 0$, hence $\boldsymbol{d_{\pi'}^\pi}(s|s_0) \geq 0$.

To check normalization, we sum over all states and use Tonelli/Fubini (permitted because the summand is non-negative) to exchange sums and expectation:

$$\sum_{s \in \mathcal{S}} \boldsymbol{d_{\pi'}^\pi}(s|s_0) = (1-\gamma) \, \mathbb{E}_{\tau \sim \mathbb{P}_{\pi'}^\pi(\cdot|s_0)} \left[ \sum_{t=0}^\infty \gamma^t \sum_{s \in \mathcal{S}} \mathbb{1}\{s_t = s\} \right] = (1-\gamma) \, \mathbb{E}_{\tau \sim \mathbb{P}_{\pi'}^\pi(\cdot|s_0)} \left[ \sum_{t=0}^\infty \gamma^t \cdot 1 \right] = (1-\gamma) \sum_{t=0}^\infty \gamma^t = 1.$$

Therefore $\rho$ is a probability mass function on $\mathcal{S}$. $\qquad\square$

A very similar argument holds for the discounted state-action occupancy measure $\boldsymbol{d_{\pi'}^\pi}(s,a|s_0)$ as well.

