# OpenReview forum: "Performative Policy Gradient: Ascent to Optimality in Performative Reinforcement Learning"
_ICLR.cc/2026/Conference — ICLR 2026 Conference Desk Rejected Submission_

### Official Review · Reviewer_UpXT · 2025-11-01

**Soundness:** 3
**Presentation:** 3
**Contribution:** 3
**Rating:** 4
**Confidence:** 3

**Summary:**

This paper studies performative reinforcement learning, where an agent's deployed policy influences the environment's reward and transition dynamics. This work applies policy gradient methods in this performative RL setting. It provides the performative RL version of the performance difference lemma and the policy gradient theorem. It provides a rigorous convergence analysis in the tabular setting. Experiments are also provided.

**Strengths:**

- This paper successfully applies policy gradients in the performative RL setting and provides theoretical analysis. The derivation of the Performative Performance Difference Lemma (Lemma 1) and the Performative Policy Gradient Theorem (Theorem 2) provides general-purpose tools for future research. Theorem 3 provides an analysis of the rate of convergence.
- The authors argue that the prevailing focus in PeRL on achieving performative stability is insufficient, as such policies can be suboptimal. Instead, this work aims to achieve performative optimality, finding a policy that maximizes the expected return within the environment it induces.

**Weaknesses:**

- The Performative Policy Gradient Theorem (Theorem 2) and the resulting gradient estimator (Equation 10) explicitly include the gradient terms (for policy, transition, and reward). To compute these terms, one must know the exact, differentiable functional form that maps the policy parameters to the environment's transition and reward functions. For instance, in the specific case of the softmax PeMDP (Equation 11), one must have access to the feature map $\psi(\cdot)$ and the reward coefficient $\xi$. This makes PePG fundamentally different from, and less generally applicable than, standard policy gradient methods.
- The application of policy gradients in performative RL also seems not to be theoretically challenging. The proof structure follows quite consistently with the seminal works.

**Questions:**

How is your Cov different from that in the standard RL setting? Would it be unbounded?
Typos: Line 97 and Line 480 have a typo on (1-\gamma). Typo also appears on line 1415.

---

> ### Author Response · Authors · 2025-11-21
>
> We thank the reviewer for taking their valuable time and effort put into reviewing our paper. We first respond to the weaknesses mentioned and move on to answer the question raised.
>
>
> **Weakness 1 : Applicability of PePG**
>
> To prove convergence of policy gradients, we need to assume a parametrization of the policy and the PeMDPs. One can assume the policy parametrisation and keep the rest vague with the Lipschitzness assumption. We consciously avoided this because: (a) we wanted to show explicit gradient computations involved in performative gradient, (b) we wanted to establish a clean relation between parametrisation of the environment with the smoothness of value functions (In Appendix E, Corollary 1  shows that the Lipschitz constants $L_r$ and $L_p$ reduce to $\xi$ and $\psi_{\max} = \max_s \psi(s)$ under our chosen parametrization of the reward and transition functions), (c) we wanted to give an algorithm that handles something more applicable than only direct parametrisation of policies in the existing literature [4,5]. These reasons have driven our choice while still generalising the direct parametrisation studied in literature.
>
> While implementing PePG in practice, one can estimate the performative parameters $\xi$ and $\psi_{\max}$ empirically and then run the PePG update using the estimated values.
>
> **Weakness 2: Challenges and Non-triviality of Convergence Analysis**
>
> Following [6,7], we understand that the proof structure of policy gradient analysis in RL follows the convergence analysis of gradient ascent for non-convex functions at a high level.
> Specifically, it involves proving three steps: establishing smoothness of the value function, showing gradient domination over the value difference (or performance difference), and subsequently deriving convergence guarantees.
> This is true for almost any setting in classical MDPs and we show for the first time that the same structure can be reused with careful choices even for proving convergence of performative policy gradient.
> The additional challenge is that performative policy gradient includes advantage function aligned gradient of log transition probabilities and gradient of reward functions.
> We show that the three steps of the proofs can be done rigorously and successfully for any softmax (and similarly exponential family) transitions and linear rewards, and softmax policies with anad without entropy regularisation, which are ubiquitous in RL literature.
> But this requires substantial new analysis.
> In general our proof technique indicates that the convergence of performative policy gradient would hold for any smooth transition and reward function with bounded expectations.
> Though this unified treatment might make the convergence analysis look traditional but incorporating these aspects with careful choices was absent in the performative RL literature, and we fill that gap in this work.
>
> After our submission, we found that [8] has recently tried to design an optimality-seeking algorithm with zeroth-order performative policy gradients and proved the convergence of performative policy gradient but they could not extend the traditional analysis and followed a different path with additional assumptions to obtain $O(\epsilon^{-4}\log(1/\epsilon))$ sample complexity only for the entropy regularised case.
> In contrast, our analysis is much intuitive-- it goes through computing the classic first order gradients, and yields $O(\epsilon^{-2})$ sample complexity.
> This reinforces the non-triviality of our convergence analysis.
>
> **Question**
>
> 1. In the classical RL literature, the coverage coefficient measures the maximum change of measure between the occupancy induced by a current
> policy $\pi_\theta$ and that induced by an optimal policy $\pi^\star$. In our performative RL setting, the term is analogous but since we do not have a fixed MDP, here we define the ratio for the MDP induced by the current policy $\pi_\theta$. Then, we take maximum over this ratio over all possible current policies.
> 2. As in classical RL ([1,2]), we prove that the lower bound of this coverage term is $1$ but no non-trivial upper bound can in general be established. Even in classical RL, we only know bounds on the coverage more discrete state-action MDPs and some variants of it ([3]). Hence, we leave it explicitly as a multiplicative factor (Theorem 3).

---

> ### Author Response · Authors · 2025-11-21
> **Rebuttal to Reviewer UpXT continued**
>
> **References**
>
> [1] Mei, J., Xiao, C., Szepesvari, C. and Schuurmans, D., 2020, November. On the global convergence rates of softmax policy gradient methods. In International conference on machine learning (pp. 6820-6829). PMLR.
>
> [2] Agarwal, A., Kakade, S.M., Lee, J.D. and Mahajan, G., 2021. On the theory of policy gradient methods: Optimality, approximation, and distribution shift. Journal of Machine Learning Research, 22(98), pp.1-76.
>
> [3] Xie, T., Foster, D.J., Bai, Y., Jiang, N., \& Kakade, S.M. (2022). The Role of Coverage in Online Reinforcement Learning.
>
> [4] Mandal, D., Triantafyllou, S. and Radanovic, G., 2023, July. Performative reinforcement learning. In International Conference on Machine Learning (pp. 23642-23680). PMLR.
>
> [5] Rank, B., Triantafyllou, S., Mandal, D. and Radanovic, G., 2024. Performative reinforcement learning in gradually shifting environments. arXiv preprint arXiv:2402.09838.
>
> [6] Yuan, R., Du, S.S., Gower, R.M., Lazaric, A. and Xiao, L., 2022. Linear convergence of natural policy gradient methods with log-linear policies.
>
> [7] Yuan, R., Gower, R.M. and Lazaric, A., 2022, May. A general sample complexity analysis of vanilla policy gradient. In International Conference on Artificial Intelligence and Statistics (pp. 3332-3380). PMLR.
>
> [8] Chen, Z. and Huang, H., 2025. Achieve Performatively Optimal Policy for Performative Reinforcement Learning.

---

### Official Review · Reviewer_D3Xq · 2025-11-01

**Soundness:** 1
**Presentation:** 2
**Contribution:** 2
**Rating:** 2
**Confidence:** 4

**Summary:**

The paper studies the performative reinforcement learning (PRL) setting, where the transition kernel and reward function depend on the deployed policy. In this setting, the important solution concepts are performatively optimal points (PO) and performatively stable points (PS). The algorithms in the PRL literature find PS rather than PO points. The authors propose the first PRL algorithm that converges to PO rather than converge to PS as the algorithms in the literature do. The main contributions of the paper are to derive performance difference lemma for the PRL setting and to derive the policy gradient theorem for the performative and entropy-regularized performative settings. They propose the performative policy gradient algorithm (PePG), which is the standard policy gradient (PG) algorithm with the gradient estimation replaced by the performative counterparts. They analyze the convergence of this algorithm in the softmax policy class for softmax PeMDPs and showcase the difference between algorithms in the experiments. A summary of the theoretical contributions are as follows:
- Lemma 1: Performance difference lemma in performative setting
- Lemma 2: A bound on the performative shift around PO following from the performative performance difference lemma
- Theorem 2: Policy gradient derivation for performative and entropy-regularized performative settings
- Lemma 3: Gradient domination in softmax PeMDPs for the unregularized and entropy-regularized value function
- Theorem 3: Convergence of PePG in softmax MDPs

Note: softmax PeMDPs refer to the setting where there exists a feature map $\psi: \mathcal{S} \to [0, \psi_{\max}]$ such that the transition kernel satisfies $P_{\pi_\theta}(s' | s, a) \propto \exp (\theta_{s, a} \psi(s'))$ and the rewards are linear in policy parameters $r_{\pi_{\theta}}(s, a) = \mathcal{P}_{-[R_{\max}, R_{\max}]}[\xi \theta_{s, a}]$.

**Strengths:**

- The focus in the PRL literature is on finding PS. The question in this paper of whether we can find performatively optimal points is important.
- The results are new in PRL literature although the techniques does appear to be standard in RL/PRL literature.
- The first two chapters are well-written and enjoyable to read. I comment on the rest further on the weaknesses and questions sections.
- The authors contribute the first algorithm that finds $\epsilon$-PO in the softmax PeMDPs.
- The experiments look promising for PePG.

**Weaknesses:**

The analysis is only for softmax PeMDPs. It would be helpful to understand how restrictive this class of problems is. It would make the paper clearer to compare and contrast this class with other common classes and possibly give some examples of what can be represented in this class or not. I also find references to some related works on softmax PeMDPs neccessary.
- It is also unclear why the authors avoid the sensitivity assumption and opt for the softmax PeMDP assumption. The softmax PeMDP assumption seems to be much restrictive. It looks like it is a theoretical convenience as there is no discussion about this choice. In addition, the statement "This is the only assumption needed through the paper" (line 241) looks deceptive due to the softmax PeMDP assumption.
- The algorithm only works for softmax PeMDPs as the gradient estimation uses the specific structure of softmax PeMDPs, which appears to be a very strong assumption necessary for gradient estimation.
- Theorem 3 does not work for $\epsilon \le (\star)$ where the $(\star)$ is the term in line 1623 for the unregularized case and the term in line 1886 for the regularized case. It is not obvious whether this is a proof artifact or the true behavior. Given Theorem 3 and the experiments in [1], this might not be a proof artifact but the true behavior. The authors should consider adding a related discussion.
- The presentation of the results could be improved. It is not obvious what is assumed for results. The authors claim that only bounded rewards is assumed. However, in Theorem 2, the gradient of reward and transition functions are taken w.r.t. the policy. What ensures the differentiability? If it is the softmax PeMDP setting, then this theorem are misplaced in the Section 3 and should be in Section 4 (or the softmax PeMDPs) should be introduced and assumed earlier.
- The implication for Lemma 2 could be stated clearer, possibly by connecting the results to the later sections.
- There are no values that quantifies the performative strength. This implies the results do not depend on any such constant and it is not obvious how one should interpret the results in the sense of performativity.
- The experiments discussion are not sufficient. I elaborate on this in questions section.

Minor issues:
- The authors note that "propose the first provably converging and computationally efficient PG algorithm." While this is true for the standard PRL problem, [1] proves that the standard policy gradient ascent and natural policy gradient algorithms converge in the general setting of performative Markov potential games. The authors should consider mentioning this in the paper.
- For the theoretical results, it would be helpful to refer to the related proofs in the appendix.
- How are the sample-based estimator part (between lines 1323-1329) related to the proof? If it is not related, the authors could consider placing it to somewhere else.
- Line 323: kernesls -> kernels
- Line 1919: optimale -> optimal
- The latex equations in y-axes of Figure 2 are not rendered.
- The sentence "As we develop a PG-type algorithm, it will be interesting to see how much can we reduce variance (Wu et al., 2018; Papini et al., 2018) while achieving optimality." (line 482) could be more clear.

[1] Sahitaj, R., Sasnauskas, P., Yalın, Y., Mandal, D. and Radanovic, G., 2025, April. Independent Learning in Performative Markov Potential Games. In _International Conference on Artificial Intelligence and Statistics_ (pp. 3304-3312). PMLR.

**Questions:**

- The authors note that the only assumption needed is the bounded rewards. However, in this case, what guarantees the existence of the performative gradients? Doesn't one at least need to assume the differentiability of $P_{\pi}$ and $r_{\pi}$ w.r.t. the policy? Doesn't gradient estimation require some further assumptions such as boundedness in this case?
- In the PRL literature, the strength of the performative effects are quantified through the sensitivity assumption and the related constants, e.g., $\omega_r$, $\omega_p$ in [1] and  $\varepsilon_r$, $\varepsilon_p$ in [2]. However, the authors refrain from having such an assumption (although softmax MDP assumption seems to be stronger). In this case, how does one quantify the performative strength in this setting? It looks like it depends on $\xi$ and $\psi$ but it is not obvious how. Moreover, the regret bounds does not depend on the performative strength so it is unclear how the performance changes w.r.t. the performative strength.
- Similar to the previous question, it is not clear how one should pick the performative strength for the regularization term. In the papers referenced in Table 1, the regularization strength depends on the performative strength. Although there is an ablation study in Appendix H, the discussion is insufficient. Could you elaborate on how to pick the term?
- Could you also compare the standard PGA and NPG methods in this setting (e.g. as described in [1])? They also provide meaningful baselines as the other baselines in the work is repeated retraining algorithms.
- Why is policy stability measured w.r.t. the consecutive occupancy measures instead of policies? How can one conclude policy stability from this difference? Why is this better than measuring the difference between consecutive policies?
- What are the stopping criteria for the experiments? Are they run for fixed 100 iterations? The distance between consecutive occupancy measures seems to have converged but the value function keeps increasing on average for PePG and Reg PePG. What is the optimal policy and value? How suboptimal are the policies in the graphs? What does the theory suggests about the value function gap in this case? How many iterations are necessary to achieve a PO? Could you also plot the consecutive policy differences as well as the occupancy differences?
- What do the authors mean by "However, this stability comes at the cost of solution quality, as MDRR becomes trapped in a suboptimal point" (line 468)? The average difference between consecutive occupancy measures are the same as PePG algorithm. Could you elaborate on how one can interpret the same average behavior as "prioritizing finding any stable point over finding an optimal solution" (line 468) for the baseline MDRR algorithm and as "actively exploring for better solutions" (line 469) for the PePG algorithm. In addition, could you elaborate on why one would not want policy stability in this case?

[1] Sahitaj, R., Sasnauskas, P., Yalın, Y., Mandal, D. and Radanovic, G., 2025, April. Independent Learning in Performative Markov Potential Games. In _International Conference on Artificial Intelligence and Statistics_ (pp. 3304-3312). PMLR.

[2] Mandal, D., Triantafyllou, S. and Radanovic, G., 2023, July. Performative reinforcement learning. In _International Conference on Machine Learning_ (pp. 23642-23680). PMLR.

---

> ### Author Response · Authors · 2025-11-21
>
> Thank you for the time and effort put into reviewing our manuscript. We will respond to both the weaknesses and questions raised by the reviewer one by one below.
>
>
>
> **Weakness 1: Softmax PeMDP is a restriction**
>
> 1. To prove convergence of policy gradients, we need to assume a parametrization of the policy and the PeMDPs. One can assume the policy parametrisation and keep the rest vague with the Lipschitzness assumption. We consciously avoided this because: (a) we wanted to show explicit gradient computations involved in performative gradient, (b) we wanted to establish a clean relation between parametrisation of the environment with the smoothness of value functions, (c) we wanted to give an algorithm that handles something more applicable than only direct parametrisation of policies in the existing literature. These reasons have driven our choice while still generalising the direct parametrisation studied in literature.
>
> 2. Our proofs still work if we assume Lipschitzness like the existing literature but we cannot use or prove smoothness of value function any more. This is theoretically interesting but less informative and complex for practice.
>
> 3. Softmax policies are very standard in RL literature ([1,2,3]). If we additionally consider features of actions and states, we get log-linear policies, which are ubiquitous in practical policy gradient literature. For rigor and clarity, we avoided this further generalisation in a setup that was never studied as it might hide the insights yielded by performative RL wrt classical RL. As we have mentioned (line 483-484), it can be an important future work.
>
> 4. The softmax transitions are fair enough to consider for the first analysis as they represent exponential family of distributions. Exponential families are the most common parametric family studied in statistics and ML, and thus, we believed it to be the first distribution family to analyse. Softmax (log-linear) family is a canonical and representative choice for modeling discrete conditional distributions ([4]). In particular, softmax parametrisations are commonly employed for log-linear models of transitions in structured stochastic systems, and provide a tractable yet expressive way to capture performative feedback.
>
> 5. We also would like to differ on the remark *"The algorithm only works for softmax PeMDPs as the gradient estimation uses the specific structure of softmax PeMDPs"*. PePG works for any differentiable parametric transitions, rewards, and policies. We use softmax policies and PeMDPs to only prove the convergence result rigorously.
>
> 6. We agree that linear reward could be changed to something non-linear. But the proof will flow almost similarly (with just changed constants) if the function is differentiable, smooth, and has bounded gradients.
>
> We will add detailed discussion on this.
>
> **$\psi(\cdot)$, $\xi$ and performative constants**
>
> In Appendix E, Corollary 1 (line 1334) shows that the Lipschitz constants $L_r$ and $L_p$ reduce to $\xi$ and $\psi_{\max} = \max_s \psi(s)$ under our chosen parametrization of the reward and transition functions. Consequently, the final results do indeed depend on the performative strength
> of the environment.
>
> **Theorem 3**
>
> The $\mathcal{O}\left(\tfrac{1}{1-\gamma}\right)$ term is indeed not a proof artifact. Our result is directly analogous to relaxed weak gradient domination result i.e., Corollary 3.7 of [8], which establishes an additive term in convergence guarantees under relaxed weak gradient domination. Please refer to the discussion added to the manuscript under the implications of Theorem 3 for further clarification.
>
> **Assumption for Theorem 2.**
>
> We thank the reviewer for pointing this out, and we fully comply with the suggestion. The presentation in the current draft may indeed give the impression that Theorem 2 assumes only bounded rewards, whereas its proof implicitly relies
> on differentiability of the reward and transition operators with respect to the policy. This differentiability is guaranteed under the softmax PeMDP parametrisation introduced later in Section 4, and we acknowledge that this may create confusion regarding the logical order of assumptions.
>
> **Implication of Lemma 2**
>
> A relevant discussion clarifying the connection between Lemma 2 and the later results has been added to our manuscript.
>
> **Constants for performative strength**
>
> Refer to the discussion under *$\psi(\cdot)$, $\xi$ and performative constants*. For regularized PeMDPs, an analogous result holds for both the smoothness of the regulariser as well as the soft value function as demonstrated by equation (31) (in line 1370).

---

> > ### Author Response · Authors · 2025-11-21
> > **Rebuttal to Reviewer D3Xq continued**
> >
> > **Policy stability w.r.t occupancy**
> >
> > In the performative RL literature ([5,6,7]), policy stability and convergence are often measured via the distance between consecutive occupancy measures, since this literature leverages a one-to-one mapping between an occupancy measure and its corresponding policy. To maintain consistency with prior work and enable fair comparisons, we therefore also demonstrate policy stability using distances between consecutive occupancy measures rather than directly comparing policies. But our goal is never to show PePG's performance in stability sense but to contrast its behavior from stability-seeking algorithms.
> >
> >
> > **Stopping criteria for PePG.**
> > All experiments run for a fixed 100 iterations to ensure fair comparison across all methods. For PePG specifically, a practical stopping criterion would be when $|V_t - V_{t-1}| < \epsilon + \mathcal{O}(\frac{1}{1-\gamma})$   for some threshold $\epsilon > 0$,
> > indicating convergence of the value function following Theorem 3 in our manuscript. The 100 iteration window shows PePG's progression toward this optimal solution, though more iterations may be needed to fully converge depending on the desired $\epsilon$.
> >
> > **Why optimality?**
> >
> > Stability and optimality are two distinct goals in Performative RL. While literature focuses on stability till now, we chose to focus on optimality and develop a theory for it.
> >
> > We further show that they are connected by the Equation after Lemma 2 (lines 225-228 in first draft). The difference in performative value functions is further decomposed into the *performative shift term* and the *performance difference term*.
> > In stability-focused Performative RL algorithms, the optimization targets only the performative difference term to achieve a stable policy. For optimality, we have to minimise both of them simultaneously. This creates the main difference between stability-seeking algorithms like MDRR, and optimality-seeking algorithms like PePG. We will clarify this further.
> >
> >
> >
> > We hope, we have been able to respond to all the concerns raised by the reviewer. We are eager to discuss if there are further concerns/comments.
> >
> > **References**
> >
> > [1] Yuan, R., Gower, R.M. and Lazaric, A., 2022, May. A general sample complexity analysis of vanilla policy gradient. In International Conference on Artificial Intelligence and Statistics (pp. 3332-3380). PMLR.
> >
> > [2] Agarwal, A., Kakade, S.M., Lee, J.D. and Mahajan, G., 2021. On the theory of policy gradient methods: Optimality, approximation, and distribution shift. Journal of Machine Learning Research, 22(98), pp.1-76.
> >
> > [3] Mei, J., Xiao, C., Szepesvari, C. and Schuurmans, D., 2020, November. On the global convergence rates of softmax policy gradient methods. In International conference on machine learning (pp. 6820-6829). PMLR.
> >
> > [4] Xu, T., Zhu, H. and Paschalidis, I.C., 2021. Learning parametric policies and transition probability models of markov decision processes from data. European journal of control, 57, pp.68-75.
> >
> > [5] Mandal, D., Triantafyllou, S. and Radanovic, G., 2023, July. Performative reinforcement learning. In International Conference on Machine Learning (pp. 23642-23680). PMLR.
> >
> > [6] Rank, B., Triantafyllou, S., Mandal, D. and Radanovic, G., 2024. Performative reinforcement learning in gradually shifting environments.
> >
> > [7] Mandal, D. and Radanovic, G., 2024. Performative Reinforcement Learning with Linear Markov Decision Process.
> >
> > [8] Yuan, R., Gower, R.M. and Lazaric, A., 2022, May. A general sample complexity analysis of vanilla policy gradient. In International Conference on Artificial Intelligence and Statistics (pp. 3332-3380). PMLR.

---

> > > ### Author Response · Authors · 2025-11-28
> > >
> > > Dear Reviewer D3Xq,
> > >
> > > We appreciate your effort towards reviewing our paper. The deadline for the discussion period is coming to an end soon, so we request you let us know if our response has been satisfactory. We have tried our best to address all the concerns raised. We have already revised the current draft to improve and to comply the remarks. We are eager to discuss further if there are any further question/comments/suggestions.
> > >
> > > Regards,
> > >
> > > Authors of Submission 20746

---

### Official Review · Reviewer_izdx · 2025-11-01

**Soundness:** 2
**Presentation:** 2
**Contribution:** 2
**Rating:** 6
**Confidence:** 2

**Summary:**

In this paper, the author prove the performative counterparts of the performance difference lemma and the policy gradient theorem in RL. Based on these, they introduce the Performative Policy Gradient algorithm (PePG). The proposed method is evlauated on standard performative RL environments and outperforms s standard policy gradient algorithms and the existing performative RL algorithms aiming for stability.

**Strengths:**

Strong theorical anylisis and formulation.

Sufficient baseline and ablations.

**Weaknesses:**

1. Incorrect statement: At introduction, it says "Some examples of successful and popular policy gradient methods include TRPO, PPO, DDPG, SAC ". But DDPG and SAC are Q-learning algorithms not policy gradient methods.

2. Misleading plots: The left plot in Figure 2 marked as "Value Function Evolution". Shouldn't evaluation average return be the accurate plot here?

**Questions:**

Since evaluation is limited to one environment, would the results generalize to environments with different mechanisms?

---

> ### Author Response · Authors · 2025-11-21
>
> We thank the reviewer for their constructive feedback and for recognising the strong theoretical analysis and sufficient baselines in our work.
>
> **Incorrect statement:**
>
>
> Thank you for this observation. DDPG and SAC do use policy-gradient updates for their actors, but they are more accurately described as off-policy actor–critic methods that learn a Q-function critic and apply policy gradients through that critic. To avoid conflating them with pure on-policy policy-gradient methods such as TRPO and PPO, we revise the text to describe DDPG and SAC specifically as actor–critic algorithms.
>
>
>
> **"Value Function Evolution" Plot Label:**
>
> Thank you for raising this point—we appreciate the opportunity to clarify. The distinction here is between training and evaluation:
>
> - During training: PePG uses sampled trajectories to estimate gradients and update the policy (as described in Algorithm 2, Line 4).
>
> - For evaluation (the plot): We compute the exact value function $V^\pi_\pi$ for each learned policy by solving for the occupancy measure $d^\pi_\pi (s,a)$ across all state-action pairs, then calculating $V^\pi_\pi = \sum_{s,a} d^\pi_\pi (s,a) \cdot r_\pi (s,a)$. This gives us the precise expected return without sampling variance.
>
>
> So "Value Function Evolution" accurately describes that we're plotting the true performance of each policy. However, we agree that "Average Return" or "Expected Return" might be clearer terminology. We add this clarification in the caption to make this distinction explicit.
>
> **Generalization to Other Environments:**
>
> This is a valuable point. To our knowledge, the gridworld environment is the only standard benchmark currently available for performative RL that allows direct comparison with prior work ([1,2]). Our theoretical results (Theorem 3) hold for all softmax PeMDPs satisfying our assumptions - specifically, softmax transitions/policies and bounded rewards, which apply broadly beyond this specific environment. If you are aware of other performative RL environments that would strengthen our empirical evaluation, we would be very interested to learn about them and consider including additional experiments.
>
> We hope we have been able to answer to all the concerns raised. We look forward to further discussion during the review period.
>
>
> *References:*
>
> [1] Mandal, D., Triantafyllou, S. and Radanovic, G., 2023, July. Performative reinforcement learning. In International Conference on Machine Learning (pp. 23642-23680). PMLR.
>
> [2] Rank, B., Triantafyllou, S., Mandal, D. and Radanovic, G., 2024. Performative reinforcement learning in gradually shifting environments.

---

### Note · Program_Chairs · 2026-01-05
**Submission Desk Rejected by Program Chairs**

This paper's template violates ICLR's formatting requirements (small margins) and must be desk rejected.